# MERLIN: MULTI-VIEW REPRESENTATION LEARNING FOR ROBUST MULTIVARIATE TIME SERIES FORECASTING WITH UNFIXED MISSING RATES

## ABSTRACT

Multivariate Time Series Forecasting (MTSF) aims to predict the future values of multiple interrelated time series and support decision-making. While deep learning models have attracted much attention in MTSF for their powerful spatial-temporal encoding capabilities, they frequently encounter the challenge of missing data resulting from numerous malfunctioning data collectors in practice. In this case, existing models only rely on sparse observation, making it difficult to fully mine the semantics of MTS, which leads to a decline in their forecasting performance. Furthermore, the unfixed missing rates across different samples in reality pose robustness challenges. To address these issues, we propose Multi-View Representation Learning (Merlin) based on offline knowledge distillation and multi-view contrastive learning, which aims to help existing models achieve semantic alignment between sparse observations with different missing rates and complete observations, and enhance their robustness. On the one hand, we introduce offline knowledge distillation where a teacher model guides a student model in learning how to mine semantics from sparse observations similar to those obtainable from complete observations. On the other hand, we construct positive and negative data pairs using sparse observations with different missing rates. Then, we use multi-view contrastive learning to help the student model align semantics across sparse observations with different missing rates, thereby further enhancing its robustness. In this way, Merlin can fully enhance the robustness of existing forecasting models to MTS with unfixed missing rates and achieves high-precision MTSF with sparse observations. Experiments on four real-world datasets validate our motivation and demonstrate the superiority and practicability of Merlin.

## 1 INTRODUCTION

Multivariate Time Series Forecasting (MTSF) is widely used in practice, such as transportation (Wang et al., 2023), environment (Tan et al., 2022) and weather (Xu et al., 2021). Deep learning-based models, such as Spatial-Temporal Graph Neural Networks (STGNNs) (Shao et al., 2022b) and Transformers (Yu et al., 2023b), are widely used due to their powerful semantic mining capabilities (Benidis et al., 2022). However, they need to fully mine semantics (Global and local information) from the complete MTS, and achieve accurate spatial-temporal forecasting (Zheng et al., 2020). In reality, due to factors such as natural disasters and component failures, data collectors can easily malfunction and fail to output data normally (Zheng et al., 2023). In this case, existing models only use sparse observations to predict future values, which limits their performances (Cini et al., 2022). To illustrate, we evaluate the performance of several models (Liu et al., 2023; Shao et al., 2022a; Zhou et al., 2023) under different missing rates on the METR-LA dataset and PEMS04 dataset. As shown in Figure 1(a) and Figure 1(b), the forecasting errors (Mean Absolute Error) of the above forecasting models increase significantly as the missing rate increases.

To mitigate the adverse effects of incomplete MTS data, we must delve deeper into two questions: **how do missing values lead to the performance degradation of these models, and how can this issue be mitigated as much as possible?** By rethinking the characteristics of this task, we believe

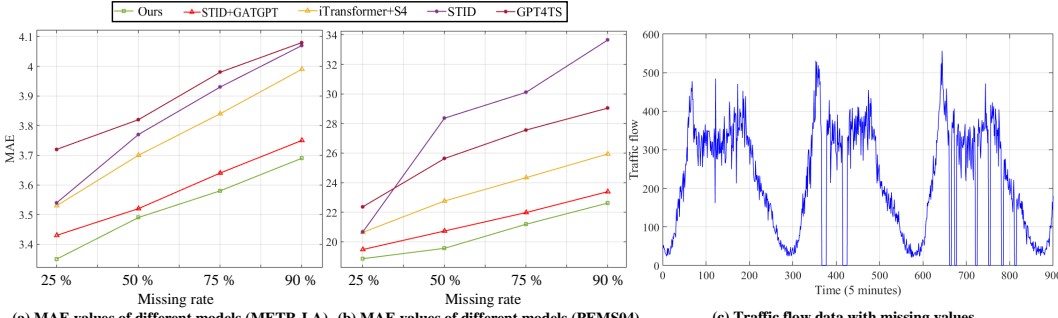

(a) MAE values of different models (METR-LA)  (b) MAE values of different models (PEMS04)    (c) Traffic flow data with missing values.

Figure 1: Examples of MTSF with sparse observations. (a) MAE values of different models on METR-LA. (b) MAE values of different models on PEMS04. As the missing rate increases, the forecasting errors of several models increase significantly. (c) Missing values disrupt the global information in time series (such as periodicity), and introduce error local information (such as sudden changes). Furthermore, the missing rate of time series changes over time.

that a large number of missing values[1] in historical observations can severely disrupt the semantics of MTS and affect the robustness of forecasting models. Specifically, as shown in Figure 1 (b), on the one hand, missing values disrupt the global information (such as periodicity) of time series and introduce error local information such as sudden changes (From normal to zero) and abnormal straight lines. If models forcibly capture these anomalies, they will mine incorrect semantics, leading to a decline in forecasting accuracy. On the other hand, since the distribution of missing values usually changes over time, the missing rates of time series at different time points are often unfixed. In this case, existing models (Lim et al., 2021; Li & Zhu, 2021; Tang et al., 2020) often need to be trained separately for different missing rates to ensure their performance, further limiting their practicability. These two phenomena lead to existing models having poor robustness in MTSF with sparse observations, resulting in a decline in their forecasting performance.

Based on the findings above, we believe that the core reason why existing forecasting models fail to achieve effective forecasting results in MTSF with sparse observations is that missing values inhibit their ability to accurately capture the semantics in sparse observations and limit their robustness. To solve the above problems, existing works use imputation methods to improve the performance of forecasting models and propose two-stage modeling approaches (Xu et al., 2023) or end-to-end modeling approaches (Tran et al., 2023) to improve their performance. However, these methods still face several challenges: (1) Existing imputation methods (Miao et al., 2021; Wu et al., 2023a) usually require reconstructing both missing and normal values, which can disrupt the local information of MTS and lead to error accumulation. (2) Existing imputation methods (Du et al., 2023; Zhou et al., 2023) need to train models separately for data with different missing rates to ensure the accuracy of data recovery. Since the missing rates in MTS are often unfixed at different time points in reality, existing imputation methods struggle to effectively recover time series with unfixed missing rates, which limits their robustness and practicality. Overall, imputation methods still fail to fully assist forecasting models in accurately mining semantics from sparse observations and addressing the issue of poor robustness. As shown in Appendix H, if imputation and forecasting models are not trained separately for each missing rate, their performance is limited.

To solve the above problems and realize robust multivariate time series forecasting with unfixed missing rates, we need to enhance the capability of existing forecasting models for semantic alignment, which includes two aspects: (1) enabling forecasting models to align the semantics between sparse observations and complete observations. (2) enabling forecasting models to align the semantics among sparse observations with different missing rates. To this end, we propose Multi-View Representation Learning (Merlin) by taking advantage of knowledge distillation and contrastive learning. On the one hand, knowledge distillation can transfer valuable knowledge from the teacher model to the student model, thereby constraining the modeling process of the student model and improving its performance (Dong et al., 2023). Considering that the model can mine more accurate semantics with complete data, we use the model trained with complete data as the teacher model.

---

[1]Missing values in most datasets, such as PEMS04 and METR-LA, are usually processed as zeros.

The student model, whose input features are sparse observations, has the same structure as the teacher model. In the training process, we transfer representations and forecasting results obtained by the teacher model as knowledge to the student model, aiming to make the student model produce representations and forecasting results that are as similar to them as possible. In this way, by constraining the student model's encoding process and forecasting process, it can learn how to align the semantics between sparse observations and complete observations, thereby enhancing the quality of the semantics mined by the student model. On the other hand, multi-view contrastive learning can help the model enhance the dissimilarity for negative data pairs and the similarity for positive data pairs, thereby achieving semantic alignment among positive data pairs (Zhang et al., 2024). To further achieve semantic alignment between samples with different missing rates and enhance the robustness of the student model, we treat samples from the same time point with different missing rates as positive data pairs, and samples from different time points as negative data pairs. In this way, multi-view contrastive learning strengthens the ability of the student model to mine and align the semantics of sparse observations with different missing rates. In this way, we only need to train one student model to adapt to samples with unfixed missing rates, significantly enhancing its robustness. Based on above methods, Merlin can effectively help existing forecasting models learn how to mine semantics from sparse observations, just as if using complete observations. Additionally, Merlin can enhance the ability of existing forecasting models to achieve semantic alignment between sparse observations with different missing rates, enabling them to achieve robust multivariate time series forecasting with unfixed missing rates. **The main contributions can be outlined as follows:**

- We believe that the main issue limiting the performance of existing forecasting models in MTSF with sparse observations is their poor robustness. On the one hand, missing values introduce error semantics to MTS. On the other hand, the missing rate of MTS changes over time, and existing models need to be trained separately for different missing rates.

- We believe that the key to achieving robust MTSF with unfixed missing rates is to help existing models achieve semantic alignment between sparse observations with different missing rates and complete observations. To this end, we propose Multi-View Representation Learning (Merlin), including knowledge distillation and contrastive learning.

- We design experiments on four real-world datasets. Results show that Merlin can enhance the performance of existing forecasting models more effectively than other imputation methods. Besides, through Merlin, forecasting models only need to be trained once to adapt to sparse observations with different missing rates.

## 2 RELATED WORK

### 2.1 SPATIAL-TEMPORAL FORECASTING METHODS

Classic STGNNs (Liu et al., 2021; Li et al., 2018; Wu et al., 2019) combine the Graph Convolutional Network (GCN) and sequence models to exploit spatial-temporal correlations. Besides, existing advanced STGNNs (Yi et al., 2023; Yu et al., 2024a) introduces graph learning technology to further improve the ability of modeling spatial correlations. Different from STGNNs, existing Transformers (Wu et al., 2023b; Zhang & Yan, 2022; Yu et al., 2023a) combine temporal attention and spatial attention, or their variants, to capture spatio-temporal information. Although STGNNs and Transformers have achieved extensive research, they often suffer from high complexity and limited scalability (Yu et al., 2024b). Currently, lightweight models based on Multi-Layer Perceptron (MLP) have gained widespread recognition. (Chen et al., 2023b) proposes TSMixer, which use all-MLP architecture to mine spatial-temporal correlations. (Shao et al., 2022a) analyze the core of modeling spatial-temporal correlations and propose an MLP framework based on the spatial-Temporal Identity (STID). In summary, a suitable MLP framework can achieve satisfactory results more efficiently than complex models. Considering that STID analyzes the characteristics of MTSF and has satisfactory performance on most datasets, it is selected as the backbone. Besides, we also evaluate the performance improvement of Merlin on other complex models.

### 2.2 KNOWLEDGE DISTILLATION

Knowledge distillation can transfer valuable knowledge from the teacher model to the student model to improve the student model's performance (Xu et al., 2022). Mainstream techniques include offline

knowledge distillation and online knowledge distillation. Among them, offline knowledge distillation offers advantages such as good stability, high flexibility, and a simplified training process. It improves the ability of the student model by continually guiding it to align with the teacher model (Yang et al., 2022). (Chattha et al., 2022) use knowledge distillation to enhance the ability of neural networks to mine samples. Experiments show that the proposed method can still achieve satisfactory results even if the sample size is reduced by 50%. (Monti et al., 2022) propose a trajectory forecasting model based on knowledge distillation and spatial-temporal Transformer, enabling the student model to perform well with only 25% of historical observations. In summary, knowledge distillation can help the student model achieve satisfactory forecasting results even when the effective information in input features is significantly reduced (Wang et al., 2021). Therefore, it can enhance the student model's capability to handle sparse observations.

### 2.3 CONTRASTIVE LEARNING

Multi-view contrastive learning enhances the model's ability to mine key information by aligning the semantics of the similar samples under different views (Hassani & Khasahmadi, 2020). (Woo et al., 2021) treat the seasonal and trend components of time series as different views and use contrastive learning to align the semantics of these different views. (Yue et al., 2022) propose the hierarchical contrastive learning method to help the model improve their ability to align the semantics of time series with different scales. (Liu & Chen, 2023) propose a self-supervised contrastive learning framework for time series representation learning, and make the forecasting model produce more reliable representations. (Dong et al., 2024) combine different masking ways with contrastive learning to mine semantics from time series. Experimental results show that contrastive learning aligns the semantics of different masked time series and enhances the reconstruction effect. Based on these references, it can be found that contrastive learning can enhance the model's ability to distinguish different samples and align the semantics between positive data pairs (Liu et al., 2022). Therefore, if we can effectively construct positive data pairs, contrastive learning can align the semantics of sparse observations with different missing rates and enhance the model's robustness.

## 3 METHODOLOGY

### 3.1 PRELIMINARIES

In this section, we introduce multivariate time series forecasting and multivariate time series forecasting with sparse observations. Some of the commonly used notations are presented in Table 1.

**Multivariate time series** (Chen et al., 2023a). It represents the data composed of multiple sequences that change over time, and can be defined through a tensor $X \in R^{N_v * N_L * N_c}$. $N_v$ is the number of sequences. $N_L$ is the number of time slices. $N_c$ is the number of features.

**Multivariate time series forecasting** (Chengqing et al., 2023). Given a historical observation tensor $X \in R^{N_v * N_H * N_c}$ from $N_H$ time slices in history, the model can predict the value $Y \in R^{N_v * N_L}$ of the nearest $N_L$ time slices in the future. $N_v$ is the number of sequences. $N_c$ is the number of features. The core goal of MTSF is to construct mapping function between input $X \in R^{N_v * N_H * N_c}$ and output $Y \in R^{N_v * N_L}$.

**Multivariate time series forecasting with sparse observations** (Sridevi et al., 2011). Compared with MTSF, the main difference of this task is that there are so much missing values in historical observations. In other words, we need to mask $M\%$ point randomly from the historical observation tensor $X \in R^{N_v * N_H * N_c}$. After the above processing, a new input feature $X_M \in R^{N_v * N_H * N_c}$ is obtained. The core goal of this task is to construct mapping function between input $X_M \in R^{N_v * N_H * N_c}$ and output $Y \in R^{N_v * N_L}$.

### 3.2 OVERALL FRAMEWORK

The overall framework of Merlin is shown in Figure 2. During the training phase, we utilize STID as the backbone and propose Merlin that combines offline knowledge distillation with multi-view contrastive learning to it. At this stage, the input features of the teacher model are complete historical observations. The input features of the student model are sparse observations. During the inference

Table 1: Frequently used notation.

| Notation | size | Definitions |
|---|---|---|
| $N_H$ | Constant | Length of historical observations |
| $N_L$ | Constant | Length of forecasting results |
| $N_s$ | Constant | Batch size |
| $N_v$ | Constant | Number of variables |
| $N_c$ | Constant | Number of features |
| $m$ | Constant | Number of missing rates |
| $X$ | $N_v * N_H * N_c$ | Complete historical observations |
| $X_M$ | $N_v * N_H * N_c$ | Sparse observations |
| $Y$ | $N_v * N_L$ | Forecasting results |
| FC | Functions | Fully connected layer |
| ReLU | Functions | Activation function ReLU |
| Mean | Functions | The mean of the Tensor |
| softmax | Functions | Activation function softmax |

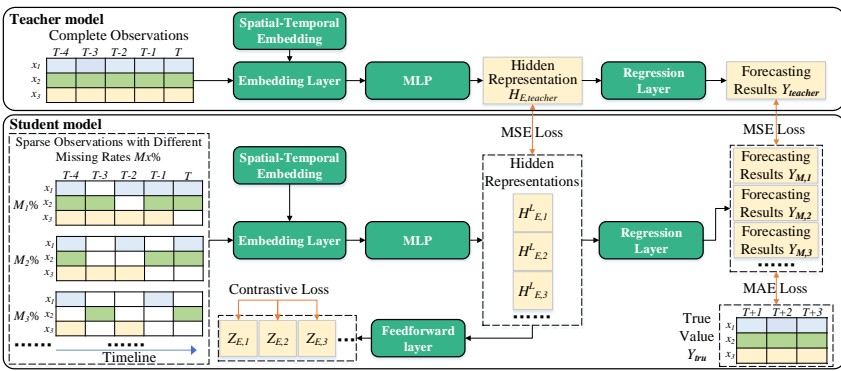

Figure 2: Overall framework of Merlin. During the training phase, the inputs of the teacher model and the student model are complete observations and sparse observations respectively. During the inference phase, only the student model is used for forecasting, whose inputs are sparse observations.

phase, we only use the student model for forecasting, whose input features are sparse observations with different missing rates. Next, we briefly describe the motivation for designing each component.

First, we explain the motivation for using STID as the backbone, which has the following advantages: (1) It introduces spatial-temporal identity embeddings to provide the model with additional information, effectively mitigating the damage of missing values. (2) It adopts a lightweight framework, which results in the model's computational complexity being only $O(N_H)$.

Then, we briefly introduce offline knowledge distillation, whose purpose is to enable STID to learn how to align the semantics between sparse and complete observations. We first train STID as a teacher model using complete observations. Then, when training the student model using sparse observations, we transfer the knowledge of the teacher model by using the representations and forecasting results generated by the teacher model. This helps the student model learn how to use sparse observations to generate representations and forecasting results similar to those generated by the teacher model. In this way, the student model can achieve semantic alignment between sparse observations and complete observations as much as possible.

Finally, we discuss the effects of multi-view contrastive learning. Although offline knowledge distillation helps STID learn how to align the semantics between sparse and complete observations, the student model still needs to improve its robustness to unfixed missing rates. Therefore, the student model needs to learn how to align the semantics between sparse observations under different missing rates. Therefore, we use sparse observations under different missing rates as positive data pairs and different samples within the same batch as negative data pairs. Through this method, the student model can utilize multi-view contrastive learning to enhance its robustness to sparse observations with different missing rates, without the need for retraining.

### 3.3 BACKBONE

In this section, we briefly introduce the basic structure of the backbone (STID), which is composed of a embeding layer, $L$ fully connected layer and a regression layer. A detailed description and definition of STID can be found in the reference (Shao et al., 2022a). The basic modeling process of STID is shown as follows:

Step I: First, the embedded layer based on a fully connected layer is used to transform the input feature $X$ into a high dimension hidden representation $H$:

$$H = \text{FC}(X), \tag{1}$$

where, $\text{FC}(\cdot)$ is the fully connected layer.

Step II: Then, the spatial-temporal identity embedding ($S_E$, $T_E^D$ and $T_E^W$) are passed to $H$ as additional inputs to improve the ability of the encoder to produce effective representations.

$$H_E = \text{Concat}(H, S_E, T_E^D, T_E^W), \tag{2}$$

where, $\text{Concat}(\cdot)$ means concatenate several tensors. Assuming $N_v$ time series and $N_H$ time slots in a day and $N_w = 7$ days in a week. $S_E \in R^{N_v * D}$ is the spatial identity embedding. $T_E^D \in R^{N_H * D}$ and $T_E^W \in R^{N_w * D}$ are the temporal embedding. $D$ is the embedding size.

Step III: The encoder based on $L$ layers of MLP with the residual connection is used to mine the above representation $Z$. The $l$-th MLP layer can be denoted as:

$$H_E^{l+1} = \text{FC}(\text{Relu}(\text{FC}(H_E^l))) + H_E^l, \tag{3}$$

where, $Relu(\cdot)$ is the activation function.

Step IV: Finally, based on the hidden representation $H_E^L$, the regression layer is used to obtain the forecasting results $Y$.

$$Y = \text{FC}(H_E^L), \tag{4}$$

In the following section, we will show how to use the hidden representation $H_E^L$ and forecasting result $Y$ for knowledge distillation and contrastive learning.

### 3.4 OFFLINE KNOWLEDGE DISTILLATION

In this paper, we use two STID models as the student model and the teacher model. The input features to the teacher model are the complete historical observations $X$. It produces the hidden representation $H_{E,Teacher}^L$ and the forecasting result $Y_{Teacher}$. The input features to the student model are the sparse observations $X_{M,1}$ to $X_{M,m}$. $m$ stands for the number of missing rates. It produces $m$ hidden representations $H_{E,1}^L$ to $H_{E,m}^L$ and $m$ forecasting results $Y_{M,1}$ to $Y_{M,m}$.

The offline knowledge distillation consists of two components: the hidden representation distillation and the forecasting result distillation. The hidden representation distillation refers to transferring the representations produced by the teacher model to the student model, aiming to minimize the mean squared error (MSE) between the representations produced by the student model and those produced by the teacher model. Its specific formula is shown as follows:

$$L_{HD} = \frac{1}{m}(\sum_{i=1}^{m} \text{Mean}((H_{E,Teacher}^L - H_{E,i}^L)^2)), \tag{5}$$

where, $Mean(\cdot)$ is the mean of the Tensor.

The process of forecasting result distillation involves transferring the forecasting results produced by the teacher model to the student model, with the objective of minimizing the MSE between the forecasting results produced by the student model and those produced by the teacher model. The specific formula is shown as follows:

$$L_{RD} = \frac{1}{m}(\sum_{i=1}^{m} \text{Mean}((Y_{Teacher} - Y_{M,i})^2)), \tag{6}$$

Based on $L_{HD}$ and $L_{RD}$, the teacher model can effectively guide the student model to use sparse observations to produce better representations and forecasting results. In this way, the student model can effectively achieve semantic alignment between sparse observations and complete observations, thereby enhancing its ability to mine key semantics from sparse observations.

### 3.5 MULTI-VIEW CONTRASTIVE LEARNING

Considering that the missing rates of historical observations in reality are not fixed, in order to further enhance the robustness of the student model and realize the semantic alignment of data with different missing rates, this paper proposes a multi-view contrastive learning method. We use historical observations with different missing rates at the same time point as positive data pairs, and use historical observations at different time point (other samples within a batch) as negative data pairs. For representations $H_{E,1}^L$ to $H_{E,m}^L$ encoded by historical observations with different missing rates, we employ a pairwise contrastive learning approach to achieve multi-view contrastive learning. The specific steps are given as follows:

Step I: Considering that appropriate dimension reduction can enhance the effectiveness of contrastive learning, a fully connected layer is used to decode the hidden representations $H_{E,1}^L$ to $H_{E,m}^L$, and get the representations $Z_{E,1}$ to $Z_{E,m}$ for Contrastive learning.

$$Z_{E,1} = \text{FC}(H_{E,1}^L), \tag{7}$$

Step II: Firstly, we use the $Z_{E,1}$ and $Z_{E,2}$ to obtain $2N_s$ samples. In $Z_{E,1}$ and $Z_{E,2}$, the corresponding two samples form a positive data pair, while the other samples are their negative data pairs. The contrast loss between any two samples $z_{E,i}$ and $z_{E,j}$ is shown as follows:

$$l_{i,j} = -\log\left(\frac{\exp(\text{sim}(z_{E,i}, z_{E,j})/\tau)}{\sum_{k=1 \& k \neq i}^{2N_s} \exp(\text{sim}(z_{E,i}, z_{E,k})/\tau)}\right), \tag{8}$$

where, $\exp(\cdot)$ is the exp function. $\text{sim}(\cdot)$ is the Cosine similarity. $N_s$ is the number of samples. $\tau$ is the temperature parameter.

Step III: Then, the contrastive loss between $Z_{E,1}$ and $Z_{E,2}$ can be obtained by the following formula:

$$L_{Z1,Z2} = \frac{1}{2N_s} \sum_{k=1}^{N_s} (l_{2k-1,2k} + l_{2k,2k-1}), \tag{9}$$

Step IV: Repeat the above steps and obtain the contrastive loss between $Z_{E,1}$ to $Z_{E,m}$ pairwise. The final multi-view contrastive learning loss is shown below:

$$L_{CL} = \frac{2}{m(m-1)}\left(\sum_{Zj=Zi}^{m} \sum_{Zi=1}^{m-1} L_{Zi,Zj}\right), \tag{10}$$

### 3.6 LOSS FUNCTION

To realize the supervised learning process, we also incorporate ground truth and L1 loss to train the student model. The formula is shown as follows (Challu et al., 2023):

$$\tag{11}$$

where, $Y_{tru}$ is the ground truth. $|\cdot|$ stands for absolute value.

Finally, we need to effectively combine all the above Loss functions. There are two main ways to integrate these Loss functions (Gou et al., 2023): multi-stage training or stacking all Loss functions. Considering the problem of information forgetting caused by multi-stage training, we use the method of adding all Loss functions. The formula is given as follows:

$$L_{Finally} = L_{Pre} + \beta(L_{HD} + L_{RD} + L_{CL}), \tag{12}$$

where, $\beta$ stands for the weight of the Loss. After completing the process of the training phase, the inference phase is performed by using only the student model. Besides, the input features are sparse observations with different missing rates.

# 4 EXPERIMENT AND ANALYSIS

## 4.1 EXPERIMENTAL DESIGN

**Datasets.** To comprehensively evaluate the validity of the proposed model, we select four real-world datasets from different domains: traffic speed (METR-LA), traffic flow (PEMS04), environment (China AQI), and meteorology (Global Wind). Detailed descriptions are provided in Appendix A.1.

**Baselines.** To comprehensively verify the performance of the proposed model, we select baselines from three perspectives: (1) We select three one-stage models that can handle missing values: GPT4TS (Zhou et al., 2023), MegaCRN (Jiang et al., 2023), and Corrformer (Wu et al., 2023b). (2) To demonstrate the improvement of Merlin on STID, we compare the STID+Merlin with the raw STID. Besides, we select four imputation methods and combine them with STID to create multiple two-stage models: STID+GATGPT (Chen et al., 2023d), STID+SPIN (Ivan et al., 2022), STID+GPT2 (Zhou et al., 2023) and STID+MAE (Li et al., 2023). (3) We combine several existing spatial-temporal forecasting models with imputation models, and obtain several two-stage models as baselines: iTransformer (Liu et al., 2023) + S4 (Gu et al., 2022), FourierGNN (Yi et al., 2023) + SPIN, DSformer (Yu et al., 2023a) + GATGPT, and TSMixer (Chen et al., 2023b) + GPT2 (Note: The previous method for each combination is the forecasting model.).

**Setting.** Hyperparametric analysis can be found in the Appendix B. Besides, we design the experiments from the following aspects: (1) According to ratios in (Shao et al., 2023), four datasets are uniformly divided into training sets, validation sets, and testing sets. (2) The history length and future length of all forecasting models are 12. All Metrics are calculated as the average of the 12-step forecasting results. More experiments on the history length and future length can be found in the Appendix G and Appendix B. (3) We randomly assign mask points with ratios of 25%, 50%, 75%, and 90%. The value of the masked point is uniformly set according to related works (Chen et al., 2023c). Experiments are repeated with 5 different random seeds for each model. The final metrics are calculated as the mean value of repeated experiments. In addition, we provide the standard deviation of the forecasting results. (4) To prove the robustness of our model, we train it once, using samples with multiple missing rates. In other words, the student model is trained simultaneously using data with missing rates of 25%, 50%, 75%, and 90%. For other baselines, we train them using two ways and report the best results: one is training a separate model for each missing rate, and the other is training a single model using samples with multiple missing rates (Shan et al., 2023). In the process of training imputation models and the teacher model, the raw data is used.

**Metrics.** In order to comprehensively evaluate the forecasting performance of our model and other baselines, three classical metrics are used, including MAE (Mean Absolute Error), RMSE (Root Mean Square Error) and MAPE (Mean Absolute Percentage Error) (Liu et al., 2020).

## 4.2 MAIN RESULTS

Table 2 shows the performance comparison results of all baselines and the proposed model on all datasets. Based on the experimental results, we can draw the following conclusions: (1) Compared with other two-stage models, the forecasting errors of all single-stage models are larger. The main reason is that existing single-stage models are easily affected by missing values, leading them to mine incorrect semantic. (2) Compared with other imputation methods, Merlin can improve the forecasting performance of STID more effectively. The main reason is that Merlin effectively combines the advantages of multi-view contrastive learning and offline knowledge distillation, which can significantly enhance the robustness of STID in modeling sparse observations and improve the capacity of STID to mine the semantics from data. (3) STID+Merlin can work better than all baselines in all cases. Firstly, we select the high-performance STID as our backbone model, which introduces temporal and spatial embeddings to provide additional semantic information for the model, helping to mitigate the impact of missing values. Secondly, we introduce offline knowledge distillation to instruct STID on how to align the semantics between sparse observations and complete observations, thereby enhancing the model's ability to mine crucial information. Finally, we propose multi-view contrastive learning to achieve semantic alignment among sparse observations with different missing rates, further improving the robustness of STID. Therefore, STID+Merlin can achieve the best forecasting results on all datasets and all missing rates. In the next section, we will further evaluate Merlin's performance improvement effects on other backbone models.

Table 2: Performance comparison results of several models. The best results are shown in **bold**. The subscript represents the standard deviation of the forecasting results.

| Datasets | Models | Missing rate 25% | | | Missing rate 50% | | | Missing rate 75% | | | Missing rate 90% | | |
|---|---|---|---|---|---|---|---|---|---|---|---|---|---|
| | | MAE | MAPE | RMSE | MAE | MAPE | RMSE | MAE | MAPE | RMSE | MAE | MAPE | RMSE |
| METR-LA | Corrformer | $3.74_{\pm0.02}$ | $10.56_{\pm0.10}$ | $7.22_{\pm0.04}$ | $3.88_{\pm0.02}$ | $11.15_{\pm0.14}$ | $7.62_{\pm0.04}$ | $3.97_{\pm0.04}$ | $11.71_{\pm0.14}$ | $7.94_{\pm0.06}$ | $4.15_{\pm0.04}$ | $12.38_{\pm0.18}$ | $8.25_{\pm0.7}$ |
| | MegaCRN | $3.63_{\pm0.02}$ | $10.13_{\pm0.10}$ | $6.88_{\pm0.04}$ | $3.79_{\pm0.02}$ | $10.76_{\pm0.12}$ | $7.38_{\pm0.04}$ | $3.94_{\pm0.04}$ | $11.18_{\pm0.14}$ | $7.65_{\pm0.02}$ | $4.03_{\pm0.04}$ | $11.89_{\pm0.17}$ | $7.93_{\pm0.06}$ |
| | GPT4TS | $3.72_{\pm0.02}$ | $10.49_{\pm0.10}$ | $7.21_{\pm0.04}$ | $3.82_{\pm0.02}$ | $10.86_{\pm0.11}$ | $7.39_{\pm0.04}$ | $3.98_{\pm0.04}$ | $11.31_{\pm0.14}$ | $7.75_{\pm0.06}$ | $4.08_{\pm0.04}$ | $12.01_{\pm0.15}$ | $8.04_{\pm0.07}$ |
| | iTransformer+S4 | $3.53_{\pm0.02}$ | $9.43_{\pm0.10}$ | $6.74_{\pm0.04}$ | $3.70_{\pm0.02}$ | $10.31_{\pm0.12}$ | $6.97_{\pm0.04}$ | $3.84_{\pm0.02}$ | $10.91_{\pm0.13}$ | $7.42_{\pm0.04}$ | $3.99_{\pm0.04}$ | $11.44_{\pm0.14}$ | $7.86_{\pm0.06}$ |
| | FourierGNN+SPIN | $3.50_{\pm0.01}$ | $9.32_{\pm0.08}$ | $6.71_{\pm0.02}$ | $3.63_{\pm0.01}$ | $10.15_{\pm0.08}$ | $6.89_{\pm0.02}$ | $3.75_{\pm0.02}$ | $10.79_{\pm0.10}$ | $7.34_{\pm0.04}$ | $3.91_{\pm0.02}$ | $11.24_{\pm0.13}$ | $7.68_{\pm0.04}$ |
| | DSformer+GATGPT | $3.52_{\pm0.01}$ | $9.37_{\pm0.09}$ | $6.73_{\pm0.02}$ | $3.65_{\pm0.01}$ | $10.24_{\pm0.09}$ | $6.94_{\pm0.02}$ | $3.78_{\pm0.02}$ | $10.86_{\pm0.10}$ | $7.38_{\pm0.04}$ | $3.89_{\pm0.02}$ | $11.19_{\pm0.12}$ | $7.66_{\pm0.04}$ |
| | TSmixer+GPT2 | $3.48_{\pm0.01}$ | $9.29_{\pm0.08}$ | $6.69_{\pm0.02}$ | $3.62_{\pm0.01}$ | $9.97_{\pm0.09}$ | $6.85_{\pm0.02}$ | $3.71_{\pm0.02}$ | $10.48_{\pm0.10}$ | $7.25_{\pm0.04}$ | $3.85_{\pm0.02}$ | $11.14_{\pm0.12}$ | $7.65_{\pm0.04}$ |
| | STID (Raw) | $3.54_{\pm0.02}$ | $9.35_{\pm0.10}$ | $6.74_{\pm0.04}$ | $3.77_{\pm0.02}$ | $10.83_{\pm0.12}$ | $7.29_{\pm0.04}$ | $3.93_{\pm0.04}$ | $11.16_{\pm0.14}$ | $7.64_{\pm0.07}$ | $4.07_{\pm0.04}$ | $11.89_{\pm0.16}$ | $8.03_{\pm0.08}$ |
| | STID+SPIN | $3.44_{\pm0.01}$ | $9.27_{\pm0.07}$ | $6.65_{\pm0.02}$ | $3.54_{\pm0.01}$ | $9.36_{\pm0.08}$ | $6.75_{\pm0.02}$ | $3.67_{\pm0.02}$ | $10.44_{\pm0.12}$ | $7.05_{\pm0.04}$ | $3.79_{\pm0.02}$ | $10.92_{\pm0.13}$ | $7.41_{\pm0.04}$ |
| | STID+GPT2 | $3.49_{\pm0.01}$ | $9.31_{\pm0.08}$ | $6.68_{\pm0.02}$ | $3.59_{\pm0.01}$ | $9.44_{\pm0.09}$ | $6.79_{\pm0.02}$ | $3.68_{\pm0.02}$ | $10.46_{\pm0.10}$ | $7.09_{\pm0.04}$ | $3.77_{\pm0.02}$ | $10.84_{\pm0.12}$ | $7.35_{\pm0.04}$ |
| | STID+MAE | $3.50_{\pm0.02}$ | $9.34_{\pm0.10}$ | $6.70_{\pm0.04}$ | $3.60_{\pm0.02}$ | $9.52_{\pm0.07}$ | $6.82_{\pm0.04}$ | $3.70_{\pm0.02}$ | $10.51_{\pm0.08}$ | $7.12_{\pm0.04}$ | $3.78_{\pm0.02}$ | $10.86_{\pm0.08}$ | $7.37_{\pm0.04}$ |
| | STID+GATGPT | $3.43_{\pm0.01}$ | $9.25_{\pm0.07}$ | $6.64_{\pm0.02}$ | $3.52_{\pm0.01}$ | $9.33_{\pm0.09}$ | $6.71_{\pm0.02}$ | $3.64_{\pm0.02}$ | $10.07_{\pm0.10}$ | $6.93_{\pm0.04}$ | $3.75_{\pm0.02}$ | $10.76_{\pm0.13}$ | $7.31_{\pm0.04}$ |
| | STID+Merlin | $\mathbf{3.35}_{\pm0.01}$ | $\mathbf{9.21}_{\pm0.05}$ | $\mathbf{6.58}_{\pm0.02}$ | $\mathbf{3.49}_{\pm0.01}$ | $\mathbf{9.29}_{\pm0.05}$ | $\mathbf{6.65}_{\pm0.02}$ | $\mathbf{3.58}_{\pm0.02}$ | $\mathbf{9.56}_{\pm0.08}$ | $\mathbf{6.81}_{\pm0.04}$ | $\mathbf{3.69}_{\pm0.02}$ | $\mathbf{10.45}_{\pm0.10}$ | $\mathbf{7.06}_{\pm0.04}$ |
| PEMS04 | Corrformer | $23.65_{\pm0.21}$ | $16.24_{\pm0.15}$ | $37.71_{\pm0.26}$ | $27.38_{\pm0.23}$ | $18.29_{\pm0.18}$ | $41.83_{\pm0.27}$ | $30.46_{\pm0.23}$ | $21.54_{\pm0.20}$ | $46.07_{\pm0.29}$ | $33.12_{\pm0.25}$ | $24.06_{\pm0.22}$ | $50.95_{\pm0.30}$ |
| | MegaCRN | $21.95_{\pm0.18}$ | $14.82_{\pm0.13}$ | $34.06_{\pm0.22}$ | $24.43_{\pm0.20}$ | $17.15_{\pm0.14}$ | $39.48_{\pm0.24}$ | $26.09_{\pm0.22}$ | $18.49_{\pm0.17}$ | $41.18_{\pm0.25}$ | $28.29_{\pm0.24}$ | $19.91_{\pm0.20}$ | $42.81_{\pm0.26}$ |
| | GPT4TS | $22.37_{\pm0.20}$ | $14.97_{\pm0.14}$ | $35.62_{\pm0.24}$ | $25.63_{\pm0.21}$ | $18.04_{\pm0.15}$ | $39.74_{\pm0.25}$ | $27.56_{\pm0.23}$ | $19.21_{\pm0.18}$ | $42.95_{\pm0.27}$ | $29.04_{\pm0.23}$ | $20.18_{\pm0.19}$ | $44.31_{\pm0.29}$ |
| | iTransformer+S4 | $20.64_{\pm0.16}$ | $14.08_{\pm0.14}$ | $32.56_{\pm0.19}$ | $22.76_{\pm0.18}$ | $15.34_{\pm0.16}$ | $36.25_{\pm0.21}$ | $24.34_{\pm0.19}$ | $17.26_{\pm0.15}$ | $39.16_{\pm0.23}$ | $25.94_{\pm0.21}$ | $18.06_{\pm0.18}$ | $40.23_{\pm0.24}$ |
| | FourierGNN+SPIN | $20.06_{\pm0.14}$ | $13.75_{\pm0.11}$ | $32.13_{\pm0.16}$ | $21.54_{\pm0.15}$ | $14.57_{\pm0.12}$ | $33.92_{\pm0.14}$ | $22.65_{\pm0.18}$ | $15.89_{\pm0.16}$ | $35.64_{\pm0.21}$ | $24.03_{\pm0.19}$ | $16.72_{\pm0.16}$ | $38.15_{\pm0.22}$ |
| | DSformer+GATGPT | $20.38_{\pm0.15}$ | $13.87_{\pm0.13}$ | $32.35_{\pm0.19}$ | $21.98_{\pm0.16}$ | $14.89_{\pm0.13}$ | $34.14_{\pm0.20}$ | $22.71_{\pm0.18}$ | $15.74_{\pm0.15}$ | $34.57_{\pm0.23}$ | $24.26_{\pm0.20}$ | $16.56_{\pm0.17}$ | $39.10_{\pm0.24}$ |
| | TSmixer+GPT2 | $20.49_{\pm0.15}$ | $13.94_{\pm0.12}$ | $32.47_{\pm0.18}$ | $22.47_{\pm0.16}$ | $15.13_{\pm0.13}$ | $35.99_{\pm0.20}$ | $24.16_{\pm0.18}$ | $17.02_{\pm0.16}$ | $38.94_{\pm0.21}$ | $25.58_{\pm0.19}$ | $17.94_{\pm0.16}$ | $39.89_{\pm0.23}$ |
| | STID (Raw) | $20.67_{\pm0.19}$ | $14.11_{\pm0.14}$ | $32.68_{\pm0.23}$ | $28.36_{\pm0.21}$ | $19.25_{\pm0.17}$ | $43.44_{\pm0.25}$ | $30.11_{\pm0.22}$ | $21.38_{\pm0.18}$ | $45.91_{\pm0.26}$ | $33.65_{\pm0.25}$ | $24.27_{\pm0.23}$ | $51.47_{\pm0.31}$ |
| | STID+SPIN | $19.53_{\pm0.13}$ | $13.22_{\pm0.11}$ | $31.35_{\pm0.15}$ | $20.79_{\pm0.15}$ | $13.82_{\pm0.12}$ | $32.79_{\pm0.18}$ | $22.85_{\pm0.15}$ | $15.77_{\pm0.13}$ | $35.69_{\pm0.18}$ | $23.79_{\pm0.17}$ | $16.45_{\pm0.15}$ | $37.96_{\pm0.21}$ |
| | STID+GPT2 | $19.85_{\pm0.14}$ | $13.54_{\pm0.11}$ | $31.80_{\pm0.17}$ | $21.45_{\pm0.16}$ | $14.33_{\pm0.13}$ | $33.54_{\pm0.19}$ | $22.44_{\pm0.17}$ | $15.51_{\pm0.13}$ | $35.21_{\pm0.21}$ | $23.51_{\pm0.19}$ | $16.21_{\pm0.16}$ | $37.58_{\pm0.24}$ |
| | STID+MAE | $19.94_{\pm0.15}$ | $13.62_{\pm0.12}$ | $31.97_{\pm0.18}$ | $21.05_{\pm0.17}$ | $13.94_{\pm0.14}$ | $33.04_{\pm0.22}$ | $22.06_{\pm0.18}$ | $15.03_{\pm0.15}$ | $34.65_{\pm0.22}$ | $23.34_{\pm0.20}$ | $15.98_{\pm0.18}$ | $37.42_{\pm0.24}$ |
| | STID+GATGPT | $19.48_{\pm0.12}$ | $13.15_{\pm0.09}$ | $31.28_{\pm0.15}$ | $20.73_{\pm0.14}$ | $14.16_{\pm0.10}$ | $32.72_{\pm0.17}$ | $21.98_{\pm0.14}$ | $14.92_{\pm0.11}$ | $35.41_{\pm0.18}$ | $23.39_{\pm0.16}$ | $16.04_{\pm0.14}$ | $37.53_{\pm0.20}$ |
| | STID+Merlin | $\mathbf{18.86}_{\pm0.10}$ | $\mathbf{12.97}_{\pm0.07}$ | $\mathbf{30.67}_{\pm0.13}$ | $\mathbf{19.56}_{\pm0.11}$ | $\mathbf{13.29}_{\pm0.09}$ | $\mathbf{31.41}_{\pm0.15}$ | $\mathbf{21.19}_{\pm0.13}$ | $\mathbf{14.21}_{\pm0.11}$ | $\mathbf{33.38}_{\pm0.16}$ | $\mathbf{22.62}_{\pm0.13}$ | $\mathbf{15.49}_{\pm0.12}$ | $\mathbf{36.27}_{\pm0.17}$ |
| China AQI | Corrformer | $16.52_{\pm0.15}$ | $34.96_{\pm0.21}$ | $27.81_{\pm0.20}$ | $18.32_{\pm0.16}$ | $39.27_{\pm0.22}$ | $30.44_{\pm0.21}$ | $20.47_{\pm0.19}$ | $43.51_{\pm0.24}$ | $31.95_{\pm0.22}$ | $22.48_{\pm0.23}$ | $45.37_{\pm0.28}$ | $34.79_{\pm0.26}$ |
| | MegaCRN | $16.35_{\pm0.15}$ | $34.75_{\pm0.21}$ | $27.61_{\pm0.20}$ | $18.14_{\pm0.16}$ | $38.43_{\pm0.22}$ | $29.46_{\pm0.20}$ | $19.96_{\pm0.18}$ | $42.64_{\pm0.23}$ | $32.54_{\pm0.21}$ | $22.06_{\pm0.21}$ | $44.28_{\pm0.27}$ | $34.42_{\pm0.24}$ |
| | GPT4TS | $16.03_{\pm0.15}$ | $33.06_{\pm0.21}$ | $27.04_{\pm0.20}$ | $17.85_{\pm0.16}$ | $37.68_{\pm0.22}$ | $28.91_{\pm0.21}$ | $19.28_{\pm0.18}$ | $41.15_{\pm0.24}$ | $32.07_{\pm0.21}$ | $21.65_{\pm0.21}$ | $43.97_{\pm0.26}$ | $33.95_{\pm0.25}$ |
| | iTransformer+S4 | $15.49_{\pm0.13}$ | $32.06_{\pm0.19}$ | $25.57_{\pm0.17}$ | $16.79_{\pm0.15}$ | $35.76_{\pm0.21}$ | $27.84_{\pm0.19}$ | $18.44_{\pm0.17}$ | $39.76_{\pm0.22}$ | $30.68_{\pm0.21}$ | $21.32_{\pm0.20}$ | $43.62_{\pm0.25}$ | $33.68_{\pm0.23}$ |
| | FourierGNN+SPIN | $15.28_{\pm0.12}$ | $31.44_{\pm0.18}$ | $25.24_{\pm0.15}$ | $16.17_{\pm0.14}$ | $34.13_{\pm0.20}$ | $27.02_{\pm0.17}$ | $18.05_{\pm0.15}$ | $38.56_{\pm0.21}$ | $30.06_{\pm0.19}$ | $20.53_{\pm0.17}$ | $42.15_{\pm0.22}$ | $32.43_{\pm0.20}$ |
| | DSformer+GATGPT | $15.39_{\pm0.12}$ | $31.89_{\pm0.18}$ | $25.43_{\pm0.16}$ | $16.39_{\pm0.14}$ | $34.82_{\pm0.20}$ | $27.58_{\pm0.19}$ | $18.29_{\pm0.16}$ | $39.37_{\pm0.22}$ | $30.17_{\pm0.20}$ | $21.07_{\pm0.18}$ | $42.97_{\pm0.23}$ | $33.04_{\pm0.21}$ |
| | TSmixer+GPT2 | $15.45_{\pm0.12}$ | $32.04_{\pm0.18}$ | $25.59_{\pm0.16}$ | $16.43_{\pm0.14}$ | $34.73_{\pm0.20}$ | $27.65_{\pm0.18}$ | $18.33_{\pm0.16}$ | $39.85_{\pm0.22}$ | $30.23_{\pm0.20}$ | $21.25_{\pm0.18}$ | $43.54_{\pm0.23}$ | $33.59_{\pm0.21}$ |
| | STID (Raw) | $15.53_{\pm0.14}$ | $32.46_{\pm0.20}$ | $25.71_{\pm0.19}$ | $18.56_{\pm0.16}$ | $39.95_{\pm0.22}$ | $30.47_{\pm0.21}$ | $20.36_{\pm0.19}$ | $43.63_{\pm0.25}$ | $32.09_{\pm0.22}$ | $23.24_{\pm0.21}$ | $46.18_{\pm0.26}$ | $35.54_{\pm0.24}$ |
| | STID+SPIN | $14.98_{\pm0.09}$ | $30.25_{\pm0.16}$ | $25.06_{\pm0.13}$ | $15.67_{\pm0.13}$ | $32.25_{\pm0.19}$ | $25.98_{\pm0.17}$ | $17.43_{\pm0.15}$ | $37.65_{\pm0.21}$ | $28.89_{\pm0.19}$ | $19.94_{\pm0.17}$ | $41.78_{\pm0.18}$ | $32.16_{\pm0.20}$ |
| | STID+GPT2 | $15.12_{\pm0.12}$ | $30.89_{\pm0.17}$ | $25.15_{\pm0.15}$ | $15.89_{\pm0.14}$ | $32.84_{\pm0.20}$ | $26.74_{\pm0.18}$ | $17.35_{\pm0.15}$ | $37.22_{\pm0.20}$ | $28.72_{\pm0.18}$ | $19.50_{\pm0.16}$ | $41.26_{\pm0.22}$ | $31.73_{\pm0.19}$ |
| | STID+MAE | $15.22_{\pm0.12}$ | $31.06_{\pm0.18}$ | $25.19_{\pm0.16}$ | $15.94_{\pm0.14}$ | $32.76_{\pm0.20}$ | $26.97_{\pm0.18}$ | $17.29_{\pm0.14}$ | $37.05_{\pm0.19}$ | $28.42_{\pm0.17}$ | $19.23_{\pm0.15}$ | $40.53_{\pm0.21}$ | $31.59_{\pm0.18}$ |
| | STID+GATGPT | $15.07_{\pm0.10}$ | $30.53_{\pm0.16}$ | $25.11_{\pm0.14}$ | $15.75_{\pm0.12}$ | $32.65_{\pm0.18}$ | $26.61_{\pm0.16}$ | $17.26_{\pm0.13}$ | $36.94_{\pm0.19}$ | $29.15_{\pm0.17}$ | $19.19_{\pm0.16}$ | $40.56_{\pm0.20}$ | $31.36_{\pm0.18}$ |
| | STID+Merlin | $\mathbf{14.89}_{\pm0.08}$ | $\mathbf{29.97}_{\pm0.15}$ | $\mathbf{24.93}_{\pm0.12}$ | $\mathbf{15.39}_{\pm0.10}$ | $\mathbf{31.86}_{\pm0.16}$ | $\mathbf{25.46}_{\pm0.14}$ | $\mathbf{16.83}_{\pm0.11}$ | $\mathbf{36.30}_{\pm0.17}$ | $\mathbf{27.30}_{\pm0.15}$ | $\mathbf{18.68}_{\pm0.13}$ | $\mathbf{39.39}_{\pm0.19}$ | $\mathbf{30.31}_{\pm0.17}$ |
| Global Wind | Corrformer | $5.78_{\pm0.02}$ | $34.32_{\pm0.17}$ | $8.52_{\pm0.04}$ | $5.99_{\pm0.02}$ | $37.18_{\pm0.19}$ | $8.79_{\pm0.05}$ | $6.29_{\pm0.04}$ | $42.65_{\pm0.20}$ | $9.18_{\pm0.07}$ | $6.59_{\pm0.04}$ | $45.98_{\pm0.22}$ | $9.63_{\pm0.08}$ |
| | MegaCRN | $5.71_{\pm0.02}$ | $32.98_{\pm0.16}$ | $8.39_{\pm0.03}$ | $5.91_{\pm0.02}$ | $36.12_{\pm0.18}$ | $8.71_{\pm0.04}$ | $6.17_{\pm0.04}$ | $40.69_{\pm0.19}$ | $9.09_{\pm0.07}$ | $6.44_{\pm0.04}$ | $45.21_{\pm0.21}$ | $9.48_{\pm0.08}$ |
| | GPT4TS | $5.73_{\pm0.02}$ | $33.25_{\pm0.16}$ | $8.41_{\pm0.03}$ | $5.95_{\pm0.02}$ | $36.57_{\pm0.18}$ | $8.76_{\pm0.04}$ | $6.23_{\pm0.04}$ | $41.35_{\pm0.20}$ | $9.13_{\pm0.07}$ | $6.53_{\pm0.04}$ | $45.79_{\pm0.21}$ | $9.56_{\pm0.08}$ |
| | iTransformer+S4 | $5.62_{\pm0.01}$ | $32.66_{\pm0.15}$ | $8.30_{\pm0.02}$ | $5.86_{\pm0.02}$ | $35.12_{\pm0.17}$ | $8.67_{\pm0.04}$ | $6.10_{\pm0.02}$ | $39.45_{\pm0.18}$ | $8.94_{\pm0.05}$ | $6.32_{\pm0.04}$ | $43.61_{\pm0.20}$ | $9.24_{\pm0.07}$ |
| | FourierGNN+SPIN | $5.59_{\pm0.01}$ | $32.18_{\pm0.14}$ | $8.23_{\pm0.02}$ | $5.72_{\pm0.02}$ | $33.22_{\pm0.16}$ | $8.43_{\pm0.03}$ | $5.95_{\pm0.02}$ | $35.69_{\pm0.17}$ | $8.69_{\pm0.04}$ | $6.16_{\pm0.03}$ | $40.18_{\pm0.18}$ | $9.01_{\pm0.06}$ |
| | DSformer+GATGPT | $5.60_{\pm0.01}$ | $32.25_{\pm0.13}$ | $8.25_{\pm0.02}$ | $5.79_{\pm0.02}$ | $34.53_{\pm0.16}$ | $8.54_{\pm0.03}$ | $5.98_{\pm0.02}$ | $37.21_{\pm0.17}$ | $8.76_{\pm0.04}$ | $6.21_{\pm0.03}$ | $41.25_{\pm0.18}$ | $9.15_{\pm0.06}$ |
| | TSmixer+GPT2 | $5.61_{\pm0.01}$ | $32.58_{\pm0.14}$ | $8.28_{\pm0.02}$ | $5.83_{\pm0.02}$ | $34.94_{\pm0.16}$ | $8.62_{\pm0.03}$ | $6.09_{\pm0.02}$ | $38.52_{\pm0.17}$ | $8.91_{\pm0.04}$ | $6.31_{\pm0.03}$ | $43.57_{\pm0.18}$ | $9.22_{\pm0.06}$ |
| | STID (Raw) | $5.63_{\pm0.01}$ | $32.73_{\pm0.15}$ | $8.31_{\pm0.02}$ | $6.05_{\pm0.02}$ | $38.49_{\pm0.18}$ | $8.87_{\pm0.04}$ | $6.34_{\pm0.04}$ | $43.19_{\pm0.19}$ | $9.25_{\pm0.06}$ | $6.68_{\pm0.04}$ | $46.72_{\pm0.22}$ | $9.77_{\pm0.08}$ |
| | STID+SPIN | $5.53_{\pm0.01}$ | $31.15_{\pm0.11}$ | $7.93_{\pm0.02}$ | $5.64_{\pm0.01}$ | $32.78_{\pm0.14}$ | $8.33_{\pm0.02}$ | $5.97_{\pm0.04}$ | $36.71_{\pm0.17}$ | $8.74_{\pm0.04}$ | $6.22_{\pm0.03}$ | $41.45_{\pm0.18}$ | $9.11_{\pm0.07}$ |
| | STID+GPT2 | $5.57_{\pm0.01}$ | $32.01_{\pm0.12}$ | $7.99_{\pm0.02}$ | $5.69_{\pm0.02}$ | $33.09_{\pm0.15}$ | $8.39_{\pm0.03}$ | $5.89_{\pm0.02}$ | $35.90_{\pm0.17}$ | $8.65_{\pm0.04}$ | $6.15_{\pm0.03}$ | $40.05_{\pm0.18}$ | $9.08_{\pm0.06}$ |
| | STID+MAE | $5.58_{\pm0.01}$ | $32.06_{\pm0.13}$ | $8.04_{\pm0.02}$ | $5.71_{\pm0.02}$ | $33.25_{\pm0.15}$ | $8.43_{\pm0.04}$ | $5.86_{\pm0.02}$ | $35.46_{\pm0.16}$ | $8.62_{\pm0.04}$ | $6.11_{\pm0.02}$ | $39.45_{\pm0.17}$ | $9.02_{\pm0.05}$ |
| | STID+GATGPT | $5.55_{\pm0.01}$ | $31.75_{\pm0.11}$ | $7.98_{\pm0.02}$ | $5.68_{\pm0.01}$ | $32.45_{\pm0.13}$ | $8.37_{\pm0.02}$ | $5.85_{\pm0.02}$ | $35.08_{\pm0.16}$ | $8.57_{\pm0.04}$ | $6.13_{\pm0.02}$ | $39.84_{\pm0.17}$ | $9.05_{\pm0.05}$ |
| | STID+Merlin | $\mathbf{5.49}_{\pm0.01}$ | $\mathbf{30.54}_{\pm0.10}$ | $\mathbf{7.85}_{\pm0.02}$ | $\mathbf{5.57}_{\pm0.01}$ | $\mathbf{31.98}_{\pm0.12}$ | $\mathbf{8.01}_{\pm0.02}$ | $\mathbf{5.78}_{\pm0.01}$ | $\mathbf{34.19}_{\pm0.14}$ | $\mathbf{8.49}_{\pm0.02}$ | $\mathbf{6.02}_{\pm0.02}$ | $\mathbf{38.47}_{\pm0.16}$ | $\mathbf{8.84}_{\pm0.04}$ |

## 4.3 Transferability of Merlin

It can be found from the main results that Merlin can effectively improve the forecasting performance of STID in MTSF with sparse observations. To further validate the effectiveness and transferability of Merlin, we choose three other models (TSmixer, DSformer, and FourierGNN) as backbones and compare the performance of Merlin with other imputation methods (GATGPT, GPT2, MAE and SPIN). Table 3 shows the MAE values of Merlin and other imputation methods. Based on the results, we can draw the following conclusions: (1) Advanced one-stage models struggle to perform well in MTSF with sparse observations. Specifically, the presence of missing data makes it difficult for existing models to mine semantics from sparse observations, resulting in poor robustness. Therefore, existing forecasting models struggle to achieve satisfactory results. (2) Compared with SPIN, the generative imputation methods can achieve better forecasting results when the missing rate is higher. The main reason is that SPIN relies on local spatial-temporal information, which makes its performance limited at high missing rates. (3) Compared with other methods, Merlin can better restore the performance of all backbone models on all datasets. The experimental results fully prove the transfer ability and practical value of Merlin. Specifically, Merlin can help existing advanced models achieve semantic alignment between sparse observations and complete observations, thereby effectively enhancing the model's robustness and achieving better forecasting results.

## 4.4 Ablation Experiments

We conduct ablation experiments from the following perspectives: (1) **w/o HD**: We remove the hidden representation distillation. (2) **w/o RD**: We remove the forecasting result distillation. (3) **w/o KD**: We removed the teacher model and knowledge distillation. In this case, STID uses complete observations and sparse observations to construct contrastive learning. (4) **w/o CL**: We remove the multi-view contrastive learning. Figure 3 shows the results of the ablation experiment. Based on the experimental results, we can draw the following conclusions: (1) The forecasting result distillation

Table 3: MAE values of Merlin and other methods (The best results are shown in **bold**).

| Backbone | Methods | METR-LA | | | | PEMS04 | | | | China AQI | | | | Global Wind | | | |
|---|---|---|---|---|---|---|---|---|---|---|---|---|---|---|---|---|---|
| | | 25% | 50% | 75% | 90% | 25% | 50% | 75% | 90% | 25% | 50% | 75% | 90% | 25% | 50% | 75% | 90% |
| TSmixer | +Merlin | **3.44**±0.01 | **3.54**±0.01 | **3.66**±0.02 | **3.78**±0.02 | **19.53**±0.12 | **21.54**±0.13 | **22.39**±0.14 | **23.95**±0.15 | **15.18**±0.09 | **16.07**±0.11 | **17.94**±0.12 | **20.58**±0.14 | **5.55**±0.01 | **5.77**±0.01 | **5.96**±0.02 | **6.15**±0.02 |
| | +GATGPT | 3.46±0.01 | 3.59±0.01 | 3.69±0.02 | 3.81±0.02 | 19.97±0.13 | 21.85±0.14 | 23.06±0.16 | 24.47±0.17 | 15.22±0.10 | 16.38±0.12 | 18.25±0.14 | 20.97±0.16 | 5.58±0.01 | 5.79±0.01 | 6.03±0.02 | 6.22±0.02 |
| | +GPT2 | 3.48±0.01 | 3.62±0.01 | 3.71±0.02 | 3.85±0.02 | 20.49±0.15 | 22.47±0.16 | 24.16±0.18 | 25.58±0.19 | 15.45±0.12 | 16.43±0.14 | 18.33±0.16 | 21.25±0.18 | 5.61±0.01 | 5.83±0.02 | 6.09±0.02 | 6.31±0.03 |
| | +MAE | 3.53±0.02 | 3.65±0.02 | 3.73±0.02 | 3.84±0.02 | 20.67±0.16 | 22.58±0.17 | 24.23±0.18 | 25.37±0.20 | 15.52±0.14 | 16.52±0.15 | 18.27±0.16 | 21.03±0.17 | 5.64±0.01 | 5.92±0.02 | 6.11±0.02 | 6.28±0.02 |
| | +SPIN | 3.47±0.01 | 3.60±0.01 | 3.72±0.02 | 3.87±0.03 | 20.23±0.13 | 22.15±0.15 | 24.19±0.16 | 25.76±0.18 | 15.25±0.12 | 16.39±0.14 | 18.41±0.16 | 21.54±0.17 | 5.60±0.01 | 5.81±0.02 | 6.07±0.02 | 6.34±0.03 |
| | raw | 3.62±0.02 | 3.78±0.02 | 3.95±0.04 | 4.06±0.04 | 21.53±0.20 | 26.39±0.22 | 29.18±0.25 | 31.42±0.27 | 16.33±0.15 | 18.44±0.17 | 20.59±0.20 | 22.98±0.22 | 5.77±0.02 | 6.01±0.02 | 6.31±0.04 | 6.63±0.04 |
| DSformer | +Merlin | **3.49**±0.01 | **3.61**±0.01 | **3.70**±0.02 | **3.82**±0.02 | **20.17**±0.13 | **21.67**±0.13 | **22.08**±0.16 | **23.84**±0.17 | **15.23**±0.10 | **16.15**±0.12 | **18.07**±0.13 | **20.78**±0.15 | **5.54**±0.01 | **5.72**±0.01 | **5.87**±0.02 | **6.14**±0.02 |
| | +GATGPT | 3.52±0.01 | 3.65±0.01 | 3.78±0.02 | 3.89±0.02 | 20.38±0.15 | 21.98±0.16 | 22.71±0.18 | 24.26±0.20 | 15.39±0.12 | 16.72±0.14 | 18.76±0.16 | 21.35±0.18 | 5.60±0.01 | 5.79±0.02 | 5.98±0.02 | 6.21±0.03 |
| | +GPT2 | 3.56±0.01 | 3.69±0.01 | 3.83±0.02 | 3.97±0.02 | 20.79±0.16 | 22.59±0.18 | 23.78±0.19 | 25.14±0.21 | 15.54±0.21 | 16.82±0.15 | 18.84±0.17 | 21.46±0.19 | 5.62±0.01 | 5.82±0.02 | 6.04±0.03 | 6.25±0.04 |
| | +MAE | 3.57±0.02 | 3.71±0.02 | 3.85±0.02 | 3.95±0.02 | 20.94±0.16 | 22.67±0.17 | 23.84±0.19 | 24.98±0.21 | 15.87±0.14 | 16.91±0.16 | 18.90±0.17 | 21.39±0.19 | 5.68±0.01 | 5.89±0.02 | 6.05±0.03 | 6.23±0.03 |
| | +SPIN | 3.54±0.01 | 3.66±0.01 | 3.82±0.02 | 3.98±0.02 | 20.54±0.15 | 22.45±0.17 | 23.95±0.19 | 25.47±0.22 | 15.43±0.22 | 16.74±0.14 | 18.79±0.16 | 21.54±0.19 | 5.64±0.02 | 5.78±0.02 | 6.01±0.04 | 6.27±0.04 |
| | raw | 3.72±0.02 | 3.87±0.02 | 3.95±0.04 | 4.11±0.04 | 23.24±0.21 | 27.85±0.23 | 30.47±0.23 | 33.25±0.25 | 16.52±0.15 | 18.75±0.16 | 20.96±0.18 | 23.47±0.21 | 5.75±0.02 | 5.98±0.02 | 6.25±0.04 | 6.57±0.04 |
| FourierGNN | +Merlin | **3.45**±0.01 | **3.53**±0.01 | **3.65**±0.02 | **3.76**±0.02 | **19.32**±0.11 | **20.19**±0.12 | **21.76**±0.13 | **23.24**±0.15 | **15.04**±0.09 | **15.92**±0.11 | **17.67**±0.13 | **20.04**±0.14 | **5.52**±0.01 | **5.67**±0.01 | **5.88**±0.02 | **6.06**±0.02 |
| | +GATGPT | 3.48±0.01 | 3.57±0.01 | 3.68±0.02 | 3.79±0.02 | 19.76±0.13 | 20.86±0.15 | 22.13±0.15 | 23.51±0.17 | 15.19±0.11 | 16.15±0.12 | 17.97±0.14 | 20.37±0.15 | 5.56±0.01 | 5.69±0.01 | 5.91±0.02 | 6.10±0.02 |
| | +GPT2 | 3.53±0.01 | 3.61±0.01 | 3.72±0.02 | 3.84±0.02 | 19.97±0.14 | 21.61±0.16 | 22.58±0.18 | 23.91±0.20 | 15.37±0.12 | 16.25±0.14 | 18.12±0.16 | 20.51±0.18 | 5.58±0.01 | 5.71±0.02 | 5.94±0.02 | 6.14±0.03 |
| | +MAE | 3.55±0.02 | 3.66±0.02 | 3.75±0.02 | 3.83±0.02 | 20.08±0.15 | 21.73±0.17 | 22.70±0.18 | 23.87±0.19 | 15.42±0.13 | 16.31±0.15 | 18.15±0.16 | 20.49±0.17 | 5.61±0.01 | 5.73±0.02 | 5.93±0.02 | 6.12±0.03 |
| | +SPIN | 3.50±0.01 | 3.58±0.01 | 3.71±0.02 | 3.86±0.02 | 19.83±0.14 | 21.54±0.15 | 22.65±0.18 | 24.03±0.19 | 15.28±0.12 | 16.17±0.14 | 18.05±0.15 | 20.53±0.17 | 5.59±0.01 | 5.72±0.02 | 5.95±0.02 | 6.16±0.03 |
| | raw | 3.61±0.02 | 3.77±0.02 | 3.92±0.04 | 4.05±0.04 | 21.34±0.18 | 24.58±0.20 | 27.05±0.22 | 29.71±0.24 | 15.98±0.16 | 17.69±0.16 | 19.13±0.18 | 21.57±0.21 | 5.73±0.02 | 5.93±0.02 | 6.15±0.04 | 6.39±0.04 |

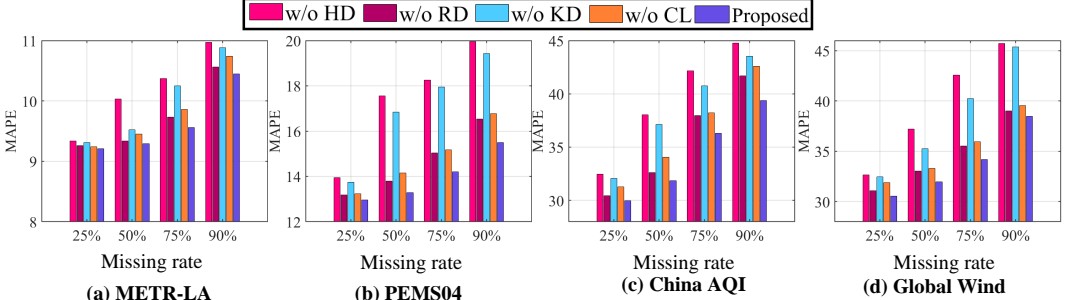

| (a) METR-LA | (b) PEMS04 | (c) China AQI | (d) Global Wind |
|---|---|---|---|

Figure 3: Results of ablation experiments. w/o HD represents the removal of hidden representation distillation. w/o RD stands for the removal of forecasting result distillation. w/o KD indicates the deletion of the teacher model and the offline knowledge distillation. w/o CL represents the removal of multi-view contrastive learning.

has the least effect on the results. The experimental results show that as long as the encoder can mine important semantics, the decoder can realize effective forecasting. (2) When the missing rate is large, the effect of multi-view contrastive learning increases significantly. The main reason is that the STID has the ability to mine semantics when the missing rate is low. (3) When STID does not use the teacher model and knowledge distillation, it can only use contrastive learning to help STID learn how to align the semantics between sparse observations and complete observations. In this case, without the guidance of teachers, it is difficult for STID to fully mine semantics from sparse observations. (4) After the hidden representation distillation is removed, the forecasting performance of STID decreases significantly. The main reason is that hidden representation distillation enables STID to learn how to make full use of sparse observations to obtain representations that can be obtained with complete observations, which is crucial for aligning the semantics between sparse observations and complete observations.

## 5 CONCLUSION

This paper considers the challenge of MTSF with unfixed missing rates from the perspective of robustness. Specifically, existing models face two challenges when modeling sparse observations: on the one hand, they must address the issue of missing values disrupting the semantics of MTS. On the other hand, they also need to face the challenge that the missing rate of MTS is unfixed at different time points in the real world. To this end, we propose Merlin based on offline knowledge distillation and multi-view contrastive learning. Merlin aims to assist existing models in effectively achieving semantic alignment between sparse observations with different missing rates and complete observations, thereby significantly enhancing their robustness. Extensive experiments show that the proposed model achieves satisfactory forecasting results on all datasets and settings. Additionally, Merlin can significantly improve the performance and robustness of existing forecasting models in MTSF with unfixed missing rates. In future work, we plan to investigate the effects of knowledge distillation when the teacher model and the student model utilize different network structures, such as large language models.

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

# A    IMPLEMENTATION DETAILS

## A.1    DATASETS

The basic statistics for these datasets are shown in Table 4. A brief introduction to these datasets is provided as follows:

- **METR-LA**[2]: It is a traffic speed dataset collected by loop-detectors located on the LA County road network, which contains data collected by 207 sensors from Mar 1st, 2012 to Jun 30th, 2012. Each time series is sampled at a 5-minute interval, totaling 34272 time slices.

- **PEMS04**[3]: It is a traffic flow dataset collected by CalTrans PeMS, which contains data collected by 307 sensors from January 1st, 2018, to February 28th, 2018. Each time series is sampled at a 5-minute interval, totaling 16992 time slices.

- **China AQI**[4]: It is an air quality dataset collected by environmental monitoring stations in China, which includes data from 1,300 air monitoring stations from January 2015 to December 2020. Each time series is sampled at a 1-hour interval, totaling 41,506 time slices.

- **Global Wind**[5]: It is derived from the global wind speed dataset of the National Oceanic and Atmospheric Administration (NOAA) National Center for Environmental Information (NCEI), which includes data from 2,908 meteorological monitoring stations from 1993 to 2022. Each time series is sampled at a 1-day interval, totaling 10,957 time slices.

Table 4: The statistics of four datasets.

| Datasets | Variates | Timesteps | Granularity |
|---|---|---|---|
| METR-LA | 207 | 34272 | 5 minutes |
| PEMS04 | 307 | 16992 | 5 minutes |
| China AQI | 1300 | 41506 | 1 hour |
| Global Wind | 2908 | 10957 | 1 day |

## A.2    BASELINES

The hyperparameter settings for the baselines are selected based on their original papers and codes. The search process of hyperparameters is mainly based on the grid search method. Specifically, we referred to the original papers and codes to set the search range for hyperparameters and introduced grid search to obtain the optimal hyperparameters. All baselines are introduced as follows:

- **Corrformer**: It uses autoregressive attention and cross attention to mine spatial-temporal correlations.

- **MegaCRN**: It uses utilizes the memory bank to enhance the adaptive graph convolution's ability to model spatial correlations and embeds the component into the recurrent neural network.

- **GPT4TS**: It uses a pretrained GPT2 to encode the context of time series, and then employs a linear decoder to obtain the forecasting results.

- **STID**: It uses spatial-temporal identity embedding to improve the ability of MLP to mine multivariate time series.

- **STID+SPIN**: SPIN effectively combines temporal attention, spatial attention, and cross attention to mine the spatial-temporal correlation of multivariate time series, thereby improving the effectiveness of data recovery.

---

[2]https://github.com/liyaguang/DCRNN

[3]https://github.com/guoshnBJTU/ASTGNN/tree/main/data

[4]https://quotsoft.net/air/

[5]https://www.ncei.noaa.gov/

- **STID+GPT2**: It first uses GPT2 to recover missing values, and then uses STID to model the processed data.

- **STID+MAE**: MAE adopts autoencoder structure to improve the effect of data recovery.

- **STID+GATGPT**: GATGPT combines GPT and graph attention mechanism to recover missing data by fully using spatial-temporal correlations.

- **iTransformer+S4**: iTransformer changes the function of the attention and feedforward layer to improve the time series forecasting results. S4 uses the fundamental state space model to mine temporal information of time series.

- **FourierGNN+SPIN**: FourierGNN uses Fourier Graph Operator to replace GCN and obtain better time series forecasting results.

- **DSformer+GATGPT**: DSformer uses uses double sampling block and temporal variable attention block to realize multivariate time series forecasting.

- **TSMixer+GPT2**: TSMixer uses residual connections and MLP to mine spatial-temporal correlations. Compared with complex models, this framework has the advantages of both performance and efficiency.

## B  Hyperparameter Analysis

Table 5 shows the main hyperparameters of the backbone (STID) and Merlin. We evaluate three hyperparameters that have the greatest impact on Merlin (The weight of the loss, batch size and temperature parameter) (Chen et al., 2020). Besides, we also evaluate three hyperparameters that have the greatest impact on the backbone (Embeding size, input length and number of layers).

The experimental results of hyperparameter analysis are shown in Figure 4 to Figure 7. Based on the hyperparameter analysis results, we can draw the following conclusions: (1) Appropriately increasing the batch size can improve the forecasting accuracy of STID. On the one hand, the increase of batch size can increase the number of negative data pairs, which can better enhance the model's robustness and uncover key semantic information. On the other hand, too large batch size can lead to premature convergence of STID, resulting in underfitting problems. (2) Proper balance of temperature parameter is important to improve the effect of contrastive learning. On the one hand, properly reducing the temperature parameter can improve the effect of the model and improve convergence. On the other hand, the value of temperature parameter being too small may lead to the problem of local optimality. (3) When the weight of the loss is set to 1, the proposed model can perform best, which fully demonstrates the importance of Merlin. Specifically, the proposed loss functions help STID realize semantic alignment effectively, reduce the interference of missing values, and thus guarantee the forecasting performance. (4) Properly balancing the size of the embedding dimension and the number of layers can effectively ensure the forecasting performance of STID. Specifically, too few parameters fail to sufficiently exploit the sparse observations, while too many parameters can lead to overfitting. (5) The input length has a significant impact on the forecasting results. The main reason is that the input length determines the amount of information that the model can capture. If the input length is too short, it fails to provide sufficient useful information, whereas an excessively long input length can lead to overfitting.

## C  Efficiency

In order to demonstrate the efficiency advantages of Merlin, this section compares the training times on the PEMS04 dataset for STID+Merlin, STID+GPT2, STID+GATGPT, iTransformer+S4, and FourierGNN+SPIN. Specifically, considering that STID+Merlin only needs to be trained once to adapt to different missing rates, whereas the other baselines require separate training sessions for each missing rate, we directly recorded the training time of STID+Merlin for a single epoch and summed up the training times for each missing rate for the other baselines. The experimental equipment is the Intel(R) Xeon(R) Gold 5217 CPU @ 3.00GHz, 128G RAM computing server with RTX 3090 graphics card.

Figure 8 displays the average training time per epoch for these models. Based on the experimental results, the following conclusions can be drawn: (1) Compared to two-stage models, STID+Merlin

Table 5: Values of the corresponding hyperparameters.

| Methods | Config | Values |
|---|---|---|
| Merlin | batch size | 32 |
| | $\beta$ | 1 |
| | temperature parameter | 0.1 |
| STID | optimizer | Adam (Kingma & Ba, 2014) |
| | learning rate | 0.002 |
| | embeding size | 64 |
| | node embedding size | 64 |
| | temporal embedding size (day) | 64 |
| | temporal embedding size (week) | 64 |
| | number of layers | 3 |
| | dropout | 0.15 |
| | learning rate schedule | MultiStepLR |
| | clip gradient normalization | 5 |
| | milestone | [1, 50, 80] |
| | gamme | 0.5 |
| | epoch | 100 |

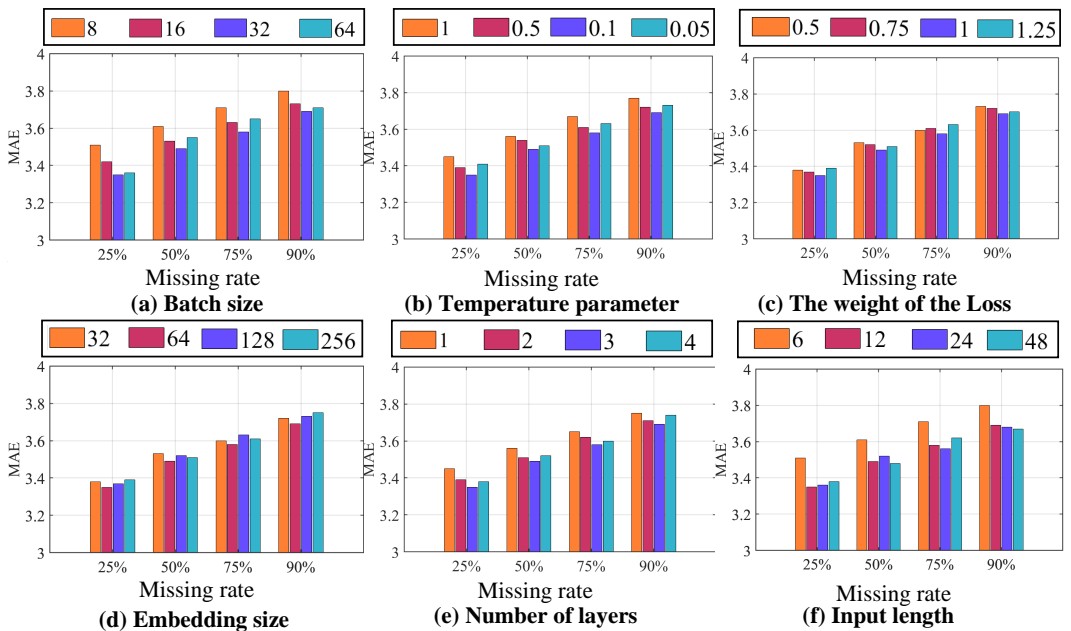

Figure 4: The results of hyperparameter experiment (METR-LA dataset).

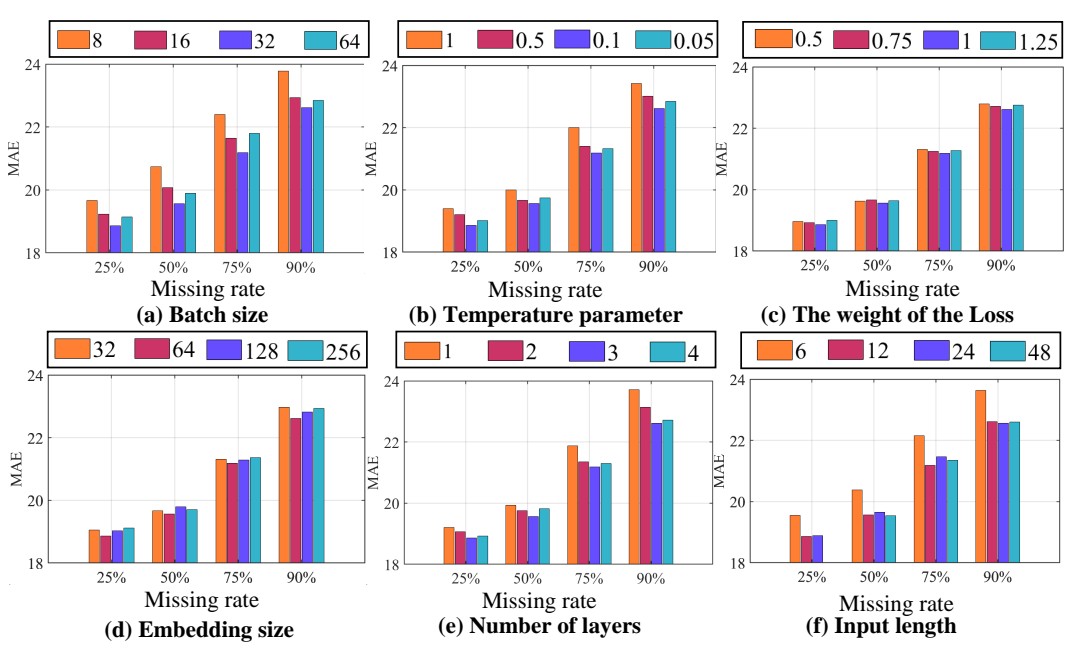

Figure 5: The results of hyperparameter experiment (PEMS04 dataset).

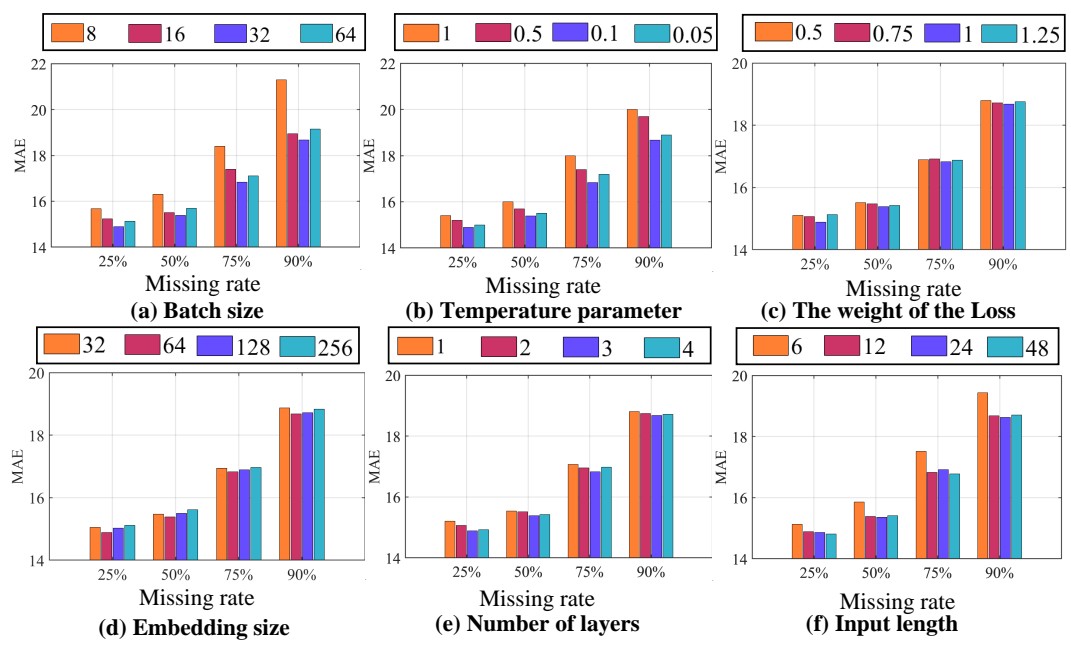

Figure 6: The results of hyperparameter experiment (China AQI dataset).

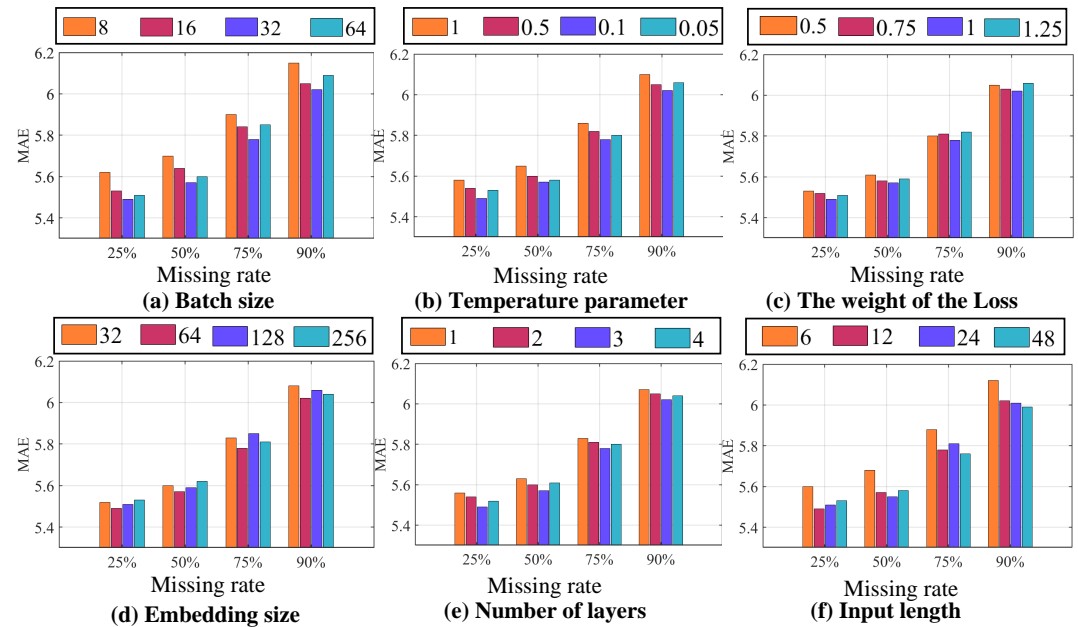

Figure 7: The results of hyperparameter experiment (Global Wind dataset).

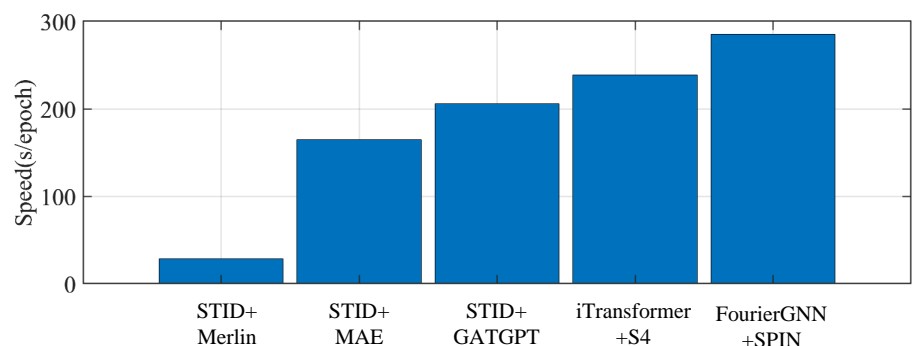

Figure 8: Training time for each epoch of different models. Compared to the two-stage models that require separate training for each missing rate, the proposed STID+Merlin significantly reduces training consumption.

requires less training time. The main reason is that STID+Merlin only needs to train one teacher model and one student model. (2) Since neither the imputation model nor the teacher model is needed during the inference phase, STID+Merlin offers greater efficiency advantages during inference. (3) Overall, despite incorporating components such as contrastive learning and knowledge distillation during the training process, STID+Merlin also achieves satisfactory results in terms of efficiency.

## D    VISUALIZATION

We demonstrate the input features and forecasting results of STID+Merlin under different missing rates on the Global wind dataset. Visualization results fully demonstrate the practical value of the proposed model. The visualization results are shown in Figure 9. It can be found that even if the input features are very sparse, the STID optimized by Merlin can still obtain satisfactory forecasting results. In addition, STID can obtain satisfactory forecasting results for input features with different missing rates. This fully proves the practical value of the proposed model in the task of multivariate time series forecasting with sparse observations.

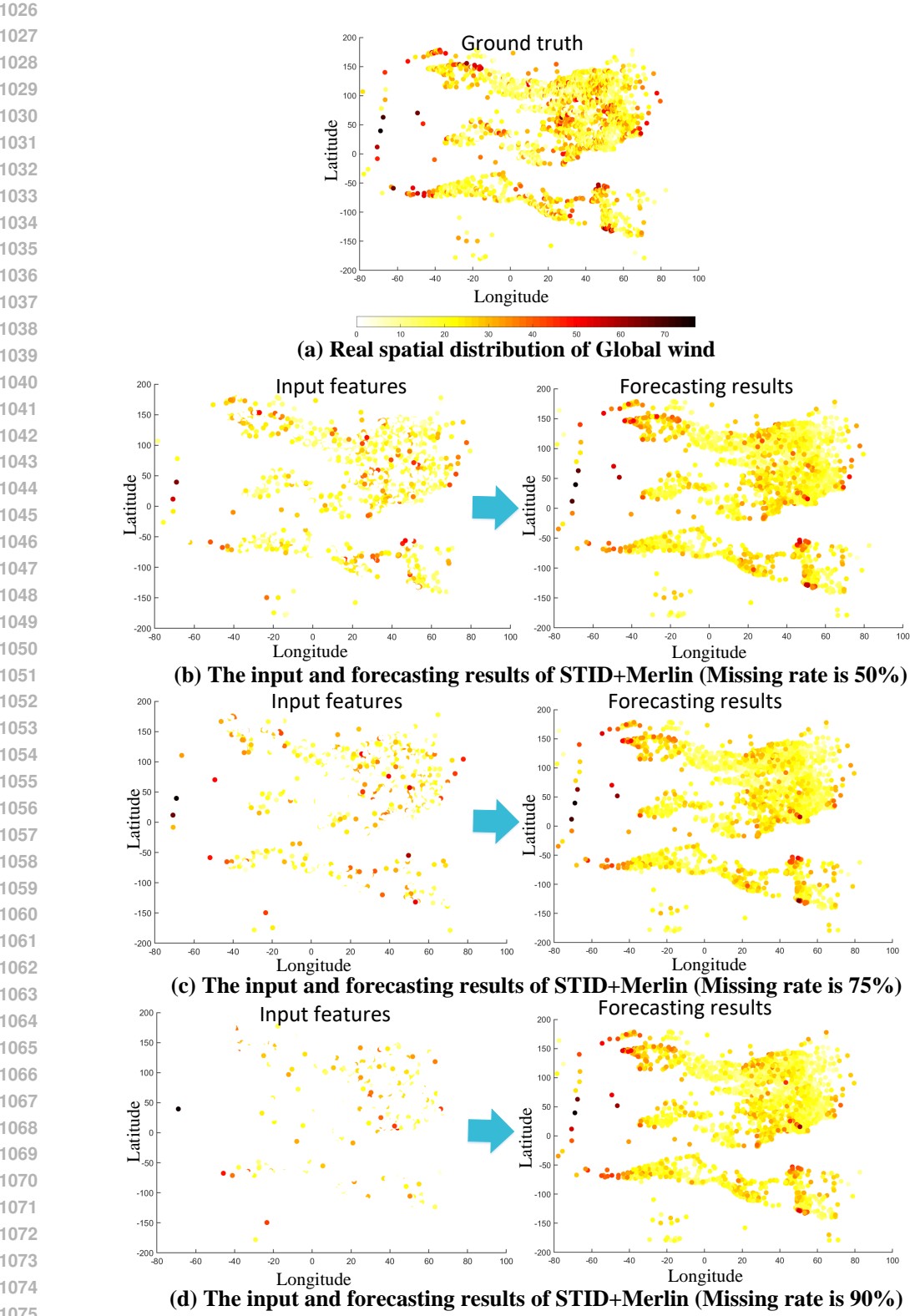

**(a) Real spatial distribution of Global wind**

**(b) The input and forecasting results of STID+Merlin (Missing rate is 50%)**

**(c) The input and forecasting results of STID+Merlin (Missing rate is 75%)**

**(d) The input and forecasting results of STID+Merlin (Missing rate is 90%)**

Figure 9: Visualization of input features and forecasting results of STID+Merlin under different missing rates (Global Wind dataset). Even with a significant increase in the missing rate, STID can still achieve good forecasting results.

Table 6: MAE values of the proposed method and other loss functions (The best results are shown in **bold**).

| Datasets | Methods | Missing rates | | | |
|---|---|---|---|---|---|
| | | 25% | 50% | 75% | 90% |
| METR-LA | Proposed | $\mathbf{3.35}_{\pm 0.01}$ | $\mathbf{3.49}_{\pm 0.01}$ | $\mathbf{3.58}_{\pm 0.02}$ | $\mathbf{3.69}_{\pm 0.02}$ |
| | L1 | $3.40_{\pm 0.01}$ | $3.51_{\pm 0.01}$ | $3.61_{\pm 0.02}$ | $3.72_{\pm 0.02}$ |
| | L2 | $3.39_{\pm 0.01}$ | $3.52_{\pm 0.01}$ | $3.63_{\pm 0.02}$ | $3.73_{\pm 0.02}$ |
| | KL-divergence | $3.42_{\pm 0.01}$ | $3.55_{\pm 0.01}$ | $3.68_{\pm 0.02}$ | $3.79_{\pm 0.02}$ |
| | Swapping | $3.37_{\pm 0.01}$ | $3.50_{\pm 0.01}$ | $3.60_{\pm 0.02}$ | $3.71_{\pm 0.02}$ |
| PEMS04 | Proposed | $\mathbf{18.86}_{\pm 0.10}$ | $\mathbf{19.56}_{\pm 0.11}$ | $\mathbf{21.19}_{\pm 0.13}$ | $\mathbf{22.62}_{\pm 0.13}$ |
| | L1 | $19.14_{\pm 0.11}$ | $19.95_{\pm 0.12}$ | $21.78_{\pm 0.14}$ | $23.05_{\pm 0.16}$ |
| | L2 | $19.22_{\pm 0.11}$ | $20.14_{\pm 0.12}$ | $22.12_{\pm 0.14}$ | $23.86_{\pm 0.16}$ |
| | KL-divergence | $19.45_{\pm 0.12}$ | $20.38_{\pm 0.14}$ | $22.53_{\pm 0.17}$ | $24.07_{\pm 0.19}$ |
| | Swapping | $19.36_{\pm 0.11}$ | $20.27_{\pm 0.13}$ | $22.09_{\pm 0.14}$ | $23.42_{\pm 0.16}$ |
| China AQI | Proposed | $\mathbf{14.89}_{\pm 0.08}$ | $\mathbf{15.39}_{\pm 0.10}$ | $\mathbf{16.83}_{\pm 0.11}$ | $\mathbf{18.68}_{\pm 0.13}$ |
| | L1 | $15.01_{\pm 0.09}$ | $15.68_{\pm 0.11}$ | $17.21_{\pm 0.12}$ | $19.06_{\pm 0.14}$ |
| | L2 | $14.98_{\pm 0.09}$ | $15.61_{\pm 0.11}$ | $17.13_{\pm 0.12}$ | $19.01_{\pm 0.14}$ |
| | KL-divergence | $15.05_{\pm 0.10}$ | $15.71_{\pm 0.12}$ | $17.24_{\pm 0.03}$ | $19.11_{\pm 0.15}$ |
| | Swapping | $14.93_{\pm 0.08}$ | $15.52_{\pm 0.10}$ | $17.02_{\pm 0.12}$ | $18.93_{\pm 0.13}$ |
| Global Wind | Proposed | $\mathbf{5.49}_{\pm 0.01}$ | $\mathbf{5.57}_{\pm 0.01}$ | $\mathbf{5.78}_{\pm 0.01}$ | $\mathbf{6.02}_{\pm 0.02}$ |
| | L1 | $5.54_{\pm 0.01}$ | $5.63_{\pm 0.01}$ | $5.84_{\pm 0.01}$ | $6.10_{\pm 0.02}$ |
| | L2 | $5.52_{\pm 0.01}$ | $5.61_{\pm 0.01}$ | $5.82_{\pm 0.01}$ | $6.07_{\pm 0.02}$ |
| | KL-divergence | $5.56_{\pm 0.01}$ | $5.65_{\pm 0.01}$ | $5.89_{\pm 0.01}$ | $6.15_{\pm 0.02}$ |
| | Swapping | $5.51_{\pm 0.01}$ | $5.59_{\pm 0.01}$ | $5.80_{\pm 0.01}$ | $6.05_{\pm 0.02}$ |

## E    COMPARED WITH DIFFERENT LOSS FUNCTIONS

In terms of constructing the loss function, this paper uses L1 Loss to evaluate the difference between the forecasting results of the student model and the ground truth. In addition, L2 Loss is used to evaluate the difference between the student model and the teacher model. To better analyze the impact of the loss function on the results, we consider using only one of the loss functions or swapping the use of the two loss functions. Besides, considering that KL divergence is also commonly used to evaluate the similarity between different distributions, we use KL divergence as a new Loss of the hidden representation distillation and carry out experiments.

Table 6 shows the MAE values of the proposed method and other loss functions (The best results are shown in boldface). The experimental results show that the proposed Loss function can get the best result. Additionally, compared to KL divergence, the MSE loss achieves better results. The main reason is that KL divergence focuses on improving the similarity between the distributions of representations, while MSE focuses on minimizing the numerical differences between representations. In summary, Multivariate time series forecasting is a regression task, where minimizing numerical differences is more important.

## F    COMPARED WITH MULTI-STAGE TRAINING

Considering that different training processes can affect the overall performance of the model, this section compares the effects of multi-stage training with adding all loss functions. The multi-stage training strategy used to construct the comparative experiment includes the following two aspects (Mukherjee & Awadallah, 2020): (1) Three-stage training: Firstly, train the model using the Loss function of knowledge distillation, then optimize the student model using the Loss function of contrastive learning, and finally optimize the student model using the Loss function of forecasting results. (2) Two-stage training: Firstly, train the model using the combination of knowledge distillation and contrastive learning. Then optimize the student model using the L1 Loss and the ground truth.

Table 7 shows the RMSE values of the proposed method and other multi-stage training methods (The best results are shown in boldface). Based on the experimental results, we can draw the following conclusions: (1) Compared with the multi-stage training strategy, the proposed method can achieve

Table 7: RMSE values of the proposed method and other multi-stage training methods (The best results are shown in **bold**).

| Datasets | Methods | Missing rates | | | |
|---|---|---|---|---|---|
| | | 25% | 50% | 75% | 90% |
| METR-LA | Proposed | **6.58**$_{\pm0.02}$ | **6.65**$_{\pm0.02}$ | **6.81**$_{\pm0.04}$ | **7.06**$_{\pm0.04}$ |
| | Two-stage | 6.62$_{\pm0.02}$ | 6.68$_{\pm0.02}$ | 6.84$_{\pm0.04}$ | 7.15$_{\pm0.04}$ |
| | Three-stage | 6.65$_{\pm0.02}$ | 6.70$_{\pm0.02}$ | 6.89$_{\pm0.04}$ | 7.22$_{\pm0.04}$ |
| PEMS04 | Proposed | **30.67**$_{\pm0.13}$ | **31.41**$_{\pm0.15}$ | **33.38**$_{\pm0.16}$ | **36.27**$_{\pm0.17}$ |
| | Two-stage | 30.89$_{\pm0.14}$ | 31.87$_{\pm0.16}$ | 33.94$_{\pm0.17}$ | 36.84$_{\pm0.19}$ |
| | Three-stage | 31.04$_{\pm0.15}$ | 31.94$_{\pm0.16}$ | 34.26$_{\pm0.18}$ | 37.15$_{\pm0.19}$ |
| China AQI | Proposed | **24.93**$_{\pm0.12}$ | **25.46**$_{\pm0.14}$ | **27.30**$_{\pm0.15}$ | **30.31**$_{\pm0.17}$ |
| | Two-stage | 25.06$_{\pm0.13}$ | 25.88$_{\pm0.15}$ | 27.95$_{\pm0.16}$ | 31.06$_{\pm0.18}$ |
| | Three-stage | 25.15$_{\pm0.15}$ | 26.03$_{\pm0.17}$ | 28.14$_{\pm0.18}$ | 31.47$_{\pm0.19}$ |
| Global Wind | Proposed | **7.85**$_{\pm0.02}$ | **8.01**$_{\pm0.02}$ | **8.49**$_{\pm0.02}$ | **8.84**$_{\pm0.04}$ |
| | Two-stage | 7.87$_{\pm0.02}$ | 8.05$_{\pm0.02}$ | 8.58$_{\pm0.02}$ | 8.97$_{\pm0.04}$ |
| | Three-stage | 7.89$_{\pm0.02}$ | 8.13$_{\pm0.02}$ | 8.61$_{\pm0.02}$ | 9.01$_{\pm0.04}$ |

Table 8: MAE values of different models on METR-LA datasets (The best results are shown in **bold**).

| Future Lengths | Methods | Missing rates | | | |
|---|---|---|---|---|---|
| | | 25% | 50% | 75% | 90% |
| 6 | STID+Merlin | **2.94**$_{\pm0.01}$ | **3.09**$_{\pm0.01}$ | **3.21**$_{\pm0.01}$ | **3.34**$_{\pm0.02}$ |
| | STID+GATGPT | 3.02$_{\pm0.01}$ | 3.16$_{\pm0.01}$ | 3.32$_{\pm0.02}$ | 3.43$_{\pm0.02}$ |
| | iTransformer+S4 | 3.14$_{\pm0.02}$ | 3.29$_{\pm0.02}$ | 3.44$_{\pm0.02}$ | 3.58$_{\pm0.02}$ |
| | TSMixer+GPT2 | 3.11$_{\pm0.01}$ | 3.25$_{\pm0.01}$ | 3.39$_{\pm0.02}$ | 3.54$_{\pm0.02}$ |
| 24 | STID+Merlin | **4.06**$_{\pm0.01}$ | **4.17**$_{\pm0.02}$ | **4.29**$_{\pm0.02}$ | **4.41**$_{\pm0.02}$ |
| | STID+GATGPT | 4.12$_{\pm0.01}$ | 4.23$_{\pm0.02}$ | 4.35$_{\pm0.02}$ | 4.52$_{\pm0.04}$ |
| | iTransformer+S4 | 4.42$_{\pm0.02}$ | 4.56$_{\pm0.04}$ | 4.60$_{\pm0.04}$ | 4.75$_{\pm0.06}$ |
| | TSMixer+GPT2 | 4.37$_{\pm0.01}$ | 4.52$_{\pm0.02}$ | 4.55$_{\pm0.04}$ | 4.71$_{\pm0.04}$ |
| 336 | STID+Merlin | **4.46**$_{\pm0.02}$ | **4.59**$_{\pm0.02}$ | **4.72**$_{\pm0.04}$ | **4.85**$_{\pm0.04}$ |
| | STID+GATGPT | 4.57$_{\pm0.02}$ | 4.71$_{\pm0.02}$ | 4.82$_{\pm0.04}$ | 4.95$_{\pm0.06}$ |
| | iTransformer+S4 | 5.06$_{\pm0.04}$ | 5.19$_{\pm0.04}$ | 5.32$_{\pm0.06}$ | 5.46$_{\pm0.06}$ |
| | TSMixer+GPT2 | 4.82$_{\pm0.02}$ | 4.95$_{\pm0.04}$ | 5.10$_{\pm0.04}$ | 5.23$_{\pm0.06}$ |

better forecasting results. The main reason is the problem of information forgetting in multi-stage training, which limits the performance of STID. (2) When the missing rate increases, the forecasting performance of the multi-stage training strategy decreases more significantly. The main reason is that information forgetting leads to the limited ability of STID to mine valuable semantics from sparse observations, which leads to the deterioration of forecasting performance.

# G    EXPERIMENT ON DIFFERENT FUTURE LENGTHS

Evaluating the performance of the proposed model under different future lengths can better show its application value. To this end, we additionally set three future lengths of 6, 24, and 336 on the METR-LA and PEMS04 datasets, and compare the forecasting performance of STID+Merlin with STID+GATGPT, DSformer+GATGPT, and TSMixer+GPT2. The setting of the input length is based on existing works (Zhou et al., 2023; Shao et al., 2023).

Table 8 and Table 9 shows the MAE values of different models. Based on the experimental results, it can be found that STID+Merlin can obtain the best forecasting results under different settings, which further proves its practicability. Specifically, the proposed model shows promising potential and value for applications in both short-term and long-term forecasting.

Table 9: MAE values of different models on PEMS04 datasets (The best results are shown in **bold**).

| Future Lengths | Methods | Missing rates | | | |
|---|---|---|---|---|---|
| | | 25% | 50% | 75% | 90% |
| 6 | STID+Merlin | $17.95_{\pm0.09}$ | $18.78_{\pm0.10}$ | $20.06_{\pm0.12}$ | $21.34_{\pm0.12}$ |
| | STID+GATGPT | $18.35_{\pm0.11}$ | $19.16_{\pm0.13}$ | $20.94_{\pm0.13}$ | $22.45_{\pm0.15}$ |
| | iTransformer+S4 | $19.54_{\pm0.15}$ | $20.63_{\pm0.17}$ | $22.06_{\pm0.18}$ | $24.04_{\pm0.20}$ |
| | TSMixer+GPT2 | $19.31_{\pm0.14}$ | $20.39_{\pm0.15}$ | $21.87_{\pm0.17}$ | $23.98_{\pm0.18}$ |
| 24 | STID+Merlin | $20.34_{\pm0.10}$ | $21.47_{\pm0.11}$ | $22.78_{\pm0.13}$ | $24.36_{\pm0.13}$ |
| | STID+GATGPT | $20.89_{\pm0.12}$ | $22.05_{\pm0.14}$ | $23.34_{\pm0.14}$ | $25.19_{\pm0.16}$ |
| | iTransformer+S4 | $21.97_{\pm0.16}$ | $23.86_{\pm0.18}$ | $25.88_{\pm0.19}$ | $28.04_{\pm0.21}$ |
| | TSMixer+GPT2 | $21.63_{\pm0.15}$ | $23.47_{\pm0.16}$ | $25.31_{\pm0.18}$ | $27.69_{\pm0.19}$ |
| 336 | STID+Merlin | $24.65_{\pm0.12}$ | $26.49_{\pm0.13}$ | $27.87_{\pm0.15}$ | $29.04_{\pm0.16}$ |
| | STID+GATGPT | $25.04_{\pm0.14}$ | $26.95_{\pm0.16}$ | $28.35_{\pm0.16}$ | $29.97_{\pm0.19}$ |
| | iTransformer+S4 | $27.58_{\pm0.18}$ | $28.78_{\pm0.20}$ | $30.06_{\pm0.21}$ | $31.57_{\pm0.23}$ |
| | TSMixer+GPT2 | $26.94_{\pm0.17}$ | $27.32_{\pm0.18}$ | $28.84_{\pm0.20}$ | $30.75_{\pm0.21}$ |

## H    EXPERIMENT ON TIME SERIES WITH UNFIXED MISSING RATES

To better simulate the unfixed missing rates in time series data under real-world scenarios, we conduct the following experiments in this section: (1) For the test data, we divided the time series into different segments based on time and applied masking to each segment with random missing rates of 25%, 50%, 75%, and 90%. (2) For the training and validation data, we additionally processed the data into four forms with missing rates of 25%, 50%, 75%, and 90%. (3) For Merlin+STID, we trained the models as described in this paper: the unmasked data is used to train the teacher model, while the masked data is used to train the student model. Only the student model is used on the test set. (4) For other baselines, we used three training strategies: the first strategy involve training separate models for each missing rate, with the corresponding model selected for forecasting on the test set based on the current data's missing rate. The second strategy uses a single model trained on data with all four missing rates, which is then directly evaluated on the test set. The final strategy is to train a model using only the raw data, which is then directly evaluated on the test set.

Table 10 shows the performance comparison results of several models under unfixed missing rates. Based on the experimental results, the following conclusions can be drawn: (1) With only be trained once, the proposed STID+Merlin achieves optimal results across all datasets. Experimental results demonstrate that STID+Merlin can effectively handle the real-world scenario of time series with unfixed missing rates. (2) For the other baselines, training models for each missing rates separately performs better than training a single model for all missing rates, which further demonstrates that existing methods are limited in both practical value and robustness in the real-world scenario of time series with unfixed missing rates. (3) If a forecasting model is trained using only complete data, its forecasting performance significantly declines when data missing occurs. This demonstrates the poor robustness of existing models in real-world scenarios.

## I    EXPERIMENT ON OTHER DATA MISSING SCENARIOS

Evaluating the proposed model's adaptability to different missing data scenarios can better demonstrate its practical value. Based on related works (Zerveas et al., 2021; Marisca et al., 2024), we conduct additional experiments under the following missing data scenarios: (1) **Data points whose mask exceeds a certain threshold**: we treat $m\%$ of the larger values and $m\%$ of the smaller values in the dataset as missing values. In other words, only the data points in the middle $(1 - 2m)\%$ of the value range are kept. (2) **Random point missing based on geometric distribution**: different from uniformly random missing situations, in this distribution, missing values appear in segments. In other words, multivariate time series exhibit a certain amount of consecutive missing values over different time periods.

Table 11 and Table 12 show the performance comparison results of several models under different data missing scenarios (The best results are shown in **bold**). Based on the experimental results, it can be found that STID+Merlin can still achieve the best experimental results under other data missing

Table 10: Performance comparison results of several models under unfixed missing rates (The best results are shown in **bold**).

| Datasets | Methods | MAE | MAPE | RMSE |
|---|---|---|---|---|
| METR-LA | Proposed | $\mathbf{3.54}_{\pm 0.01}$ | $\mathbf{9.41}_{\pm 0.05}$ | $\mathbf{6.72}_{\pm 0.02}$ |
| | STID+GATGPT (Separately) | $3.58_{\pm 0.01}$ | $9.52_{\pm 0.09}$ | $6.83_{\pm 0.02}$ |
| | STID+GATGPT (Together) | $3.67_{\pm 0.02}$ | $10.12_{\pm 0.10}$ | $6.98_{\pm 0.04}$ |
| | iTransformer+S4 (Separately) | $3.76_{\pm 0.02}$ | $10.78_{\pm 0.12}$ | $7.32_{\pm 0.04}$ |
| | iTransformer+S4 (Together) | $3.88_{\pm 0.04}$ | $11.12_{\pm 0.14}$ | $7.61_{\pm 0.07}$ |
| | STID (Separately) | $3.82_{\pm 0.04}$ | $10.87_{\pm 0.14}$ | $7.38_{\pm 0.07}$ |
| | STID (Together) | $3.95_{\pm 0.04}$ | $11.52_{\pm 0.15}$ | $7.62_{\pm 0.08}$ |
| | STID (Complete) | $4.06_{\pm 0.04}$ | $12.04_{\pm 0.16}$ | $8.01_{\pm 0.08}$ |
| | GPT4TS (Separately) | $3.89_{\pm 0.04}$ | $11.23_{\pm 0.15}$ | $7.64_{\pm 0.08}$ |
| | GPT4TS (Together) | $4.02_{\pm 0.04}$ | $12.06_{\pm 0.16}$ | $7.95_{\pm 0.08}$ |
| | GPT4TS (Complete) | $4.12_{\pm 0.04}$ | $12.34_{\pm 0.16}$ | $8.19_{\pm 0.08}$ |
| PEMS04 | Proposed | $\mathbf{20.37}_{\pm 0.12}$ | $\mathbf{13.91}_{\pm 0.10}$ | $\mathbf{32.33}_{\pm 0.16}$ |
| | STID+GATGPT (Separately) | $21.04_{\pm 0.14}$ | $14.06_{\pm 0.11}$ | $33.26_{\pm 0.18}$ |
| | STID+GATGPT (Together) | $22.76_{\pm 0.15}$ | $15.83_{\pm 0.13}$ | $34.68_{\pm 0.19}$ |
| | iTransformer+S4 (Separately) | $23.58_{\pm 0.18}$ | $16.32_{\pm 0.16}$ | $37.75_{\pm 0.22}$ |
| | iTransformer+S4 (Together) | $25.15_{\pm 0.19}$ | $17.68_{\pm 0.17}$ | $39.27_{\pm 0.24}$ |
| | STID (Separately) | $28.84_{\pm 0.21}$ | $20.15_{\pm 0.17}$ | $43.96_{\pm 0.25}$ |
| | STID (Together) | $30.06_{\pm 0.22}$ | $21.85_{\pm 0.18}$ | $45.28_{\pm 0.26}$ |
| | STID (Complete) | $31.45_{\pm 0.24}$ | $22.76_{\pm 0.21}$ | $47.89_{\pm 0.29}$ |
| | GPT4TS (Separately) | $26.57_{\pm 0.21}$ | $18.97_{\pm 0.15}$ | $42.06_{\pm 0.24}$ |
| | GPT4TS (Together) | $28.23_{\pm 0.23}$ | $19.52_{\pm 0.18}$ | $43.08_{\pm 0.26}$ |
| | GPT4TS (Complete) | $29.97_{\pm 0.25}$ | $20.84_{\pm 0.20}$ | $44.97_{\pm 0.30}$ |

scenarios. The experimental results show that Merlin can effectively guarantee the robustness of the prediction model under different data missing scenarios.

## J   EXPERIMENTS WHEN THE PERFORMANCE OF THE TEACHER MODEL IS DEGRADED

Existing imputation models typically assume access to complete training data and train models through reconstruction tasks (Ahn et al., 2022). Considering the possibility of incomplete data collection in real-world scenarios (i.e., missing data in the training set), the teacher model might be trained on multivariate time series with missing values, potentially leading to degraded performance. Therefore, it is crucial to evaluate the effectiveness of Merlin under such conditions. In this section, we simulate scenarios where the training data for the teacher model has missing rates of 5% and 10% (imputation models also face this challenge) and assess the improvement brought by Merlin and GATGPT to different backbone under these settings. Specifically, the original data is first processed to simulate missing rates of 5% and 10%. Subsequently, the data with these missing rates is further processed to simulate missing rates of 25%, 50%, 75%, and 90%. The proposed model and baselines are trained separately on datasets with 5% and 10% missing rates, as well as the datasets with subsequent missing rates of 25%, 50%, 75%, and 90%.

Table 13 and Table 14 show the MAE values of Merlin and other methods when the missing rates of the training sets are 5% and 10%, respectively. Based on the experimental results, the following conclusions can be drawn: (1) Even when the data quality of the training sets for the teacher model decreases, Merlin can still effectively enhance the forecasting performance of several backbone models. (2) Compared to GATGPT, Merlin demonstrates superior capability in recovering the forecasting performance of different backbone models, further highlighting its practical value in real-world scenarios.

Table 11: MAE values of several models (Data points whose mask exceeds a certain threshold).

| Datasets | Methods | Missing rates | | | |
|---|---|---|---|---|---|
| | | 10% | 20% | 30% | 40% |
| METR-LA | Proposed | $3.31_{\pm 0.01}$ | $3.37_{\pm 0.01}$ | $3.42_{\pm 0.01}$ | $3.51_{\pm 0.02}$ |
| | STID+GATGPT | $3.39_{\pm 0.01}$ | $3.44_{\pm 0.01}$ | $3.49_{\pm 0.02}$ | $3.56_{\pm 0.02}$ |
| | STID+MAE | $3.46_{\pm 0.01}$ | $3.53_{\pm 0.02}$ | $3.57_{\pm 0.02}$ | $3.63_{\pm 0.02}$ |
| | STID+GPT2 | $3.44_{\pm 0.01}$ | $3.50_{\pm 0.01}$ | $3.55_{\pm 0.02}$ | $3.61_{\pm 0.02}$ |
| | STID+SPIN | $3.40_{\pm 0.01}$ | $3.46_{\pm 0.01}$ | $3.52_{\pm 0.01}$ | $3.58_{\pm 0.02}$ |
| | FourierGNN+SPIN | $3.45_{\pm 0.01}$ | $3.51_{\pm 0.01}$ | $3.58_{\pm 0.02}$ | $3.65_{\pm 0.02}$ |
| | DSformer+GATGPT | $3.49_{\pm 0.01}$ | $3.54_{\pm 0.02}$ | $3.62_{\pm 0.02}$ | $3.70_{\pm 0.02}$ |
| | TSMixer+GPT2 | $3.43_{\pm 0.01}$ | $3.49_{\pm 0.01}$ | $3.55_{\pm 0.02}$ | $3.63_{\pm 0.02}$ |
| PEMS04 | Proposed | $18.56_{\pm 0.10}$ | $18.94_{\pm 0.10}$ | $19.32_{\pm 0.11}$ | $19.75_{\pm 0.12}$ |
| | STID+GATGPT | $19.21_{\pm 0.12}$ | $19.52_{\pm 0.12}$ | $20.34_{\pm 0.14}$ | $20.86_{\pm 0.15}$ |
| | STID+MAE | $19.67_{\pm 0.15}$ | $20.03_{\pm 0.16}$ | $20.79_{\pm 0.17}$ | $21.42_{\pm 0.17}$ |
| | STID+GPT2 | $19.53_{\pm 0.14}$ | $19.97_{\pm 0.15}$ | $20.87_{\pm 0.16}$ | $21.67_{\pm 0.16}$ |
| | STID+SPIN | $19.28_{\pm 0.13}$ | $19.61_{\pm 0.14}$ | $20.57_{\pm 0.14}$ | $21.15_{\pm 0.15}$ |
| | FourierGNN+SPIN | $19.98_{\pm 0.14}$ | $20.14_{\pm 0.15}$ | $21.06_{\pm 0.15}$ | $21.74_{\pm 0.16}$ |
| | DSformer+GATGPT | $20.15_{\pm 0.15}$ | $20.45_{\pm 0.16}$ | $21.58_{\pm 0.16}$ | $22.35_{\pm 0.18}$ |
| | TSMixer+GPT2 | $20.23_{\pm 0.16}$ | $20.57_{\pm 0.17}$ | $21.68_{\pm 0.17}$ | $22.73_{\pm 0.19}$ |
| China AQI | Proposed | $14.76_{\pm 0.08}$ | $14.92_{\pm 0.08}$ | $15.12_{\pm 0.09}$ | $15.45_{\pm 0.10}$ |
| | STID+GATGPT | $14.93_{\pm 0.09}$ | $15.10_{\pm 0.10}$ | $15.57_{\pm 0.11}$ | $15.83_{\pm 0.13}$ |
| | STID+MAE | $14.98_{\pm 0.12}$ | $15.29_{\pm 0.13}$ | $15.82_{\pm 0.15}$ | $16.19_{\pm 0.16}$ |
| | STID+GPT2 | $15.05_{\pm 0.12}$ | $15.21_{\pm 0.12}$ | $15.74_{\pm 0.14}$ | $16.08_{\pm 0.15}$ |
| | STID+SPIN | $14.87_{\pm 0.09}$ | $15.04_{\pm 0.09}$ | $15.61_{\pm 0.11}$ | $15.94_{\pm 0.12}$ |
| | FourierGNN+SPIN | $15.07_{\pm 0.12}$ | $15.32_{\pm 0.13}$ | $15.87_{\pm 0.15}$ | $16.25_{\pm 0.16}$ |
| | DSformer+GATGPT | $15.21_{\pm 0.12}$ | $15.45_{\pm 0.14}$ | $15.98_{\pm 0.15}$ | $16.53_{\pm 0.17}$ |
| | TSMixer+GPT2 | $15.25_{\pm 0.12}$ | $15.51_{\pm 0.14}$ | $16.04_{\pm 0.15}$ | $16.68_{\pm 0.17}$ |
| Global Wind | Proposed | $5.46_{\pm 0.01}$ | $5.52_{\pm 0.01}$ | $5.57_{\pm 0.01}$ | $5.60_{\pm 0.01}$ |
| | STID+GATGPT | $5.53_{\pm 0.01}$ | $5.58_{\pm 0.01}$ | $5.64_{\pm 0.01}$ | $5.71_{\pm 0.01}$ |
| | STID+MAE | $5.56_{\pm 0.01}$ | $5.61_{\pm 0.01}$ | $5.67_{\pm 0.02}$ | $5.74_{\pm 0.02}$ |
| | STID+GPT2 | $5.55_{\pm 0.01}$ | $5.60_{\pm 0.01}$ | $5.64_{\pm 0.01}$ | $5.72_{\pm 0.02}$ |
| | STID+SPIN | $5.51_{\pm 0.01}$ | $5.57_{\pm 0.01}$ | $5.61_{\pm 0.01}$ | $5.69_{\pm 0.01}$ |
| | FourierGNN+SPIN | $5.58_{\pm 0.01}$ | $5.62_{\pm 0.01}$ | $5.69_{\pm 0.01}$ | $5.76_{\pm 0.02}$ |
| | DSformer+GATGPT | $5.61_{\pm 0.01}$ | $5.67_{\pm 0.01}$ | $5.74_{\pm 0.01}$ | $5.82_{\pm 0.02}$ |
| | TSMixer+GPT2 | $5.59_{\pm 0.01}$ | $5.64_{\pm 0.01}$ | $5.75_{\pm 0.01}$ | $5.86_{\pm 0.02}$ |

Table 12: MAE values of several models (Random point missing based on geometric distribution).

| Datasets | Methods | Missing rates | | | |
|---|---|---|---|---|---|
| | | 25% | 50% | 75% | 90% |
| METR-LA | Proposed | $3.41_{\pm 0.01}$ | $3.55_{\pm 0.01}$ | $3.68_{\pm 0.02}$ | $3.81_{\pm 0.02}$ |
| | STID+GATGPT | $3.48_{\pm 0.01}$ | $3.63_{\pm 0.01}$ | $3.75_{\pm 0.02}$ | $3.93_{\pm 0.02}$ |
| | STID+MAE | $3.54_{\pm 0.02}$ | $3.67_{\pm 0.02}$ | $3.81_{\pm 0.02}$ | $3.95_{\pm 0.02}$ |
| | STID+GPT2 | $3.53_{\pm 0.01}$ | $3.66_{\pm 0.01}$ | $3.78_{\pm 0.02}$ | $3.96_{\pm 0.02}$ |
| | STID+SPIN | $3.49_{\pm 0.01}$ | $3.64_{\pm 0.01}$ | $3.77_{\pm 0.02}$ | $3.98_{\pm 0.02}$ |
| | FourierGNN+SPIN | $3.55_{\pm 0.01}$ | $3.71_{\pm 0.01}$ | $3.84_{\pm 0.02}$ | $4.01_{\pm 0.02}$ |
| | DSformer+GATGPT | $3.59_{\pm 0.01}$ | $3.74_{\pm 0.01}$ | $3.87_{\pm 0.02}$ | $4.05_{\pm 0.02}$ |
| | TSMixer+GPT2 | $3.53_{\pm 0.01}$ | $3.69_{\pm 0.01}$ | $3.81_{\pm 0.02}$ | $3.98_{\pm 0.02}$ |
| PEMS04 | Proposed | $19.03_{\pm 0.10}$ | $19.87_{\pm 0.11}$ | $21.45_{\pm 0.13}$ | $22.87_{\pm 0.13}$ |
| | STID+GATGPT | $19.63_{\pm 0.12}$ | $21.06_{\pm 0.14}$ | $22.57_{\pm 0.14}$ | $24.15_{\pm 0.16}$ |
| | STID+MAE | $20.16_{\pm 0.15}$ | $21.44_{\pm 0.17}$ | $22.63_{\pm 0.18}$ | $24.14_{\pm 0.20}$ |
| | STID+GPT2 | $20.04_{\pm 0.14}$ | $21.67_{\pm 0.16}$ | $22.87_{\pm 0.17}$ | $24.35_{\pm 0.19}$ |
| | STID+SPIN | $19.75_{\pm 0.13}$ | $21.13_{\pm 0.15}$ | $23.14_{\pm 0.15}$ | $24.49_{\pm 0.18}$ |
| | FourierGNN+SPIN | $20.85_{\pm 0.14}$ | $22.35_{\pm 0.15}$ | $23.76_{\pm 0.18}$ | $24.55_{\pm 0.19}$ |
| | DSformer+GATGPT | $21.03_{\pm 0.15}$ | $22.89_{\pm 0.16}$ | $24.32_{\pm 0.18}$ | $24.78_{\pm 0.23}$ |
| | TSMixer+GPT2 | $21.16_{\pm 0.15}$ | $23.07_{\pm 0.16}$ | $24.58_{\pm 0.18}$ | $25.19_{\pm 0.21}$ |
| China AQI | Proposed | $14.95_{\pm 0.08}$ | $15.48_{\pm 0.10}$ | $17.06_{\pm 0.11}$ | $18.87_{\pm 0.13}$ |
| | STID+GATGPT | $15.14_{\pm 0.10}$ | $15.89_{\pm 0.12}$ | $17.43_{\pm 0.13}$ | $19.31_{\pm 0.15}$ |
| | STID+MAE | $15.28_{\pm 0.12}$ | $16.13_{\pm 0.14}$ | $17.51_{\pm 0.14}$ | $19.42_{\pm 0.15}$ |
| | STID+GPT2 | $15.19_{\pm 0.12}$ | $16.06_{\pm 0.14}$ | $17.63_{\pm 0.15}$ | $19.78_{\pm 0.16}$ |
| | STID+SPIN | $15.06_{\pm 0.10}$ | $15.91_{\pm 0.10}$ | $17.75_{\pm 0.10}$ | $20.1_{\pm 0.10}5$ |
| | FourierGNN+SPIN | $15.37_{\pm 0.12}$ | $16.26_{\pm 0.14}$ | $18.59_{\pm 0.17}$ | $20.98_{\pm 0.19}$ |
| | DSformer+GATGPT | $15.49_{\pm 0.12}$ | $16.45_{\pm 0.14}$ | $18.67_{\pm 0.19}$ | $21.45_{\pm 0.20}$ |
| | TSMixer+GPT2 | $15.61_{\pm 0.12}$ | $16.53_{\pm 0.14}$ | $18.81_{\pm 0.18}$ | $21.97_{\pm 0.20}$ |
| Global Wind | Proposed | $5.52_{\pm 0.01}$ | $5.61_{\pm 0.01}$ | $5.82_{\pm 0.01}$ | $6.09_{\pm 0.02}$ |
| | STID+GATGPT | $5.59_{\pm 0.01}$ | $5.72_{\pm 0.01}$ | $5.91_{\pm 0.02}$ | $6.18_{\pm 0.02}$ |
| | STID+MAE | $5.62_{\pm 0.01}$ | $5.75_{\pm 0.02}$ | $5.93_{\pm 0.02}$ | $6.16_{\pm 0.02}$ |
| | STID+GPT2 | $5.60_{\pm 0.01}$ | $5.73_{\pm 0.02}$ | $5.97_{\pm 0.02}$ | $6.19_{\pm 0.03}$ |
| | STID+SPIN | $5.57_{\pm 0.01}$ | $5.69_{\pm 0.01}$ | $6.01_{\pm 0.02}$ | $6.25_{\pm 0.03}$ |
| | FourierGNN+SPIN | $5.63_{\pm 0.01}$ | $5.76_{\pm 0.02}$ | $6.02_{\pm 0.02}$ | $6.21_{\pm 0.03}$ |
| | DSformer+GATGPT | $5.66_{\pm 0.01}$ | $5.82_{\pm 0.02}$ | $6.07_{\pm 0.02}$ | $6.28_{\pm 0.03}$ |
| | TSMixer+GPT2 | $5.64_{\pm 0.01}$ | $5.85_{\pm 0.02}$ | $6.15_{\pm 0.02}$ | $6.37_{\pm 0.03}$ |

Table 13: MAE values of Merlin and other methods (The missing rate of the training set is 5%).

| Backbone | Methods | METR-LA | | | | PEMS04 | | | | China AQI | | | | Global Wind | | | |
|---|---|---|---|---|---|---|---|---|---|---|---|---|---|---|---|---|---|
| | | 25% | 50% | 75% | 90% | 25% | 50% | 75% | 90% | 25% | 50% | 75% | 90% | 25% | 50% | 75% | 90% |
| STID | +Merlin | $3.39_{\pm0.01}$ | $3.54_{\pm0.01}$ | $3.62_{\pm0.02}$ | $3.71_{\pm0.02}$ | $19.14_{\pm0.10}$ | $20.07_{\pm0.11}$ | $21.43_{\pm0.13}$ | $23.28_{\pm0.13}$ | $15.06_{\pm0.08}$ | $15.67_{\pm0.10}$ | $17.06_{\pm0.11}$ | $18.93_{\pm0.13}$ | $5.52_{\pm0.01}$ | $5.61_{\pm0.01}$ | $5.82_{\pm0.02}$ | $6.06_{\pm0.02}$ |
| | +GATGPT | $3.46_{\pm0.01}$ | $3.57_{\pm0.01}$ | $3.68_{\pm0.02}$ | $3.80_{\pm0.02}$ | $19.72_{\pm0.12}$ | $21.08_{\pm0.14}$ | $22.54_{\pm0.14}$ | $23.97_{\pm0.16}$ | $15.27_{\pm0.10}$ | $15.98_{\pm0.12}$ | $17.52_{\pm0.13}$ | $19.43_{\pm0.15}$ | $5.58_{\pm0.01}$ | $5.72_{\pm0.01}$ | $5.90_{\pm0.02}$ | $6.17_{\pm0.02}$ |
| | +GPT2 | $3.51_{\pm0.01}$ | $3.64_{\pm0.01}$ | $3.72_{\pm0.02}$ | $3.81_{\pm0.02}$ | $20.05_{\pm0.14}$ | $21.76_{\pm0.16}$ | $22.84_{\pm0.17}$ | $24.15_{\pm0.17}$ | $15.31_{\pm0.12}$ | $16.19_{\pm0.14}$ | $17.64_{\pm0.15}$ | $19.84_{\pm0.16}$ | $5.59_{\pm0.01}$ | $5.74_{\pm0.02}$ | $5.94_{\pm0.02}$ | $6.21_{\pm0.03}$ |
| | +MAE | $3.52_{\pm0.02}$ | $3.66_{\pm0.02}$ | $3.74_{\pm0.02}$ | $3.82_{\pm0.02}$ | $20.13_{\pm0.15}$ | $21.52_{\pm0.17}$ | $22.63_{\pm0.18}$ | $24.02_{\pm0.20}$ | $15.41_{\pm0.12}$ | $16.52_{\pm0.14}$ | $18.27_{\pm0.14}$ | $19.52_{\pm0.15}$ | $5.60_{\pm0.01}$ | $5.76_{\pm0.02}$ | $5.92_{\pm0.02}$ | $6.16_{\pm0.02}$ |
| | +SPIN | $3.47_{\pm0.01}$ | $3.62_{\pm0.02}$ | $3.75_{\pm0.02}$ | $3.84_{\pm0.02}$ | $19.80_{\pm0.13}$ | $21.34_{\pm0.15}$ | $23.02_{\pm0.15}$ | $24.33_{\pm0.17}$ | $15.25_{\pm0.12}$ | $16.39_{\pm0.13}$ | $18.41_{\pm0.15}$ | $20.09_{\pm0.15}$ | $5.57_{\pm0.01}$ | $5.73_{\pm0.02}$ | $6.01_{\pm0.02}$ | $6.26_{\pm0.03}$ |
| | raw | $3.54_{\pm0.02}$ | $3.77_{\pm0.02}$ | $3.93_{\pm0.04}$ | $4.07_{\pm0.04}$ | $20.67_{\pm0.19}$ | $28.36_{\pm0.21}$ | $30.11_{\pm0.22}$ | $33.65_{\pm0.25}$ | $15.53_{\pm0.14}$ | $18.56_{\pm0.16}$ | $20.36_{\pm0.19}$ | $23.24_{\pm0.21}$ | $5.63_{\pm0.01}$ | $6.05_{\pm0.02}$ | $6.34_{\pm0.04}$ | $6.68_{\pm0.04}$ |
| TSmixer | +Merlin | $3.48_{\pm0.01}$ | $3.59_{\pm0.01}$ | $3.70_{\pm0.02}$ | $3.82_{\pm0.02}$ | $19.84_{\pm0.12}$ | $21.95_{\pm0.13}$ | $22.78_{\pm0.14}$ | $24.43_{\pm0.15}$ | $15.43_{\pm0.09}$ | $16.31_{\pm0.11}$ | $18.24_{\pm0.12}$ | $20.81_{\pm0.14}$ | $5.58_{\pm0.01}$ | $5.81_{\pm0.01}$ | $6.01_{\pm0.02}$ | $6.21_{\pm0.02}$ |
| | +GATGPT | $3.51_{\pm0.01}$ | $3.64_{\pm0.01}$ | $3.75_{\pm0.02}$ | $3.87_{\pm0.02}$ | $20.45_{\pm0.13}$ | $22.38_{\pm0.14}$ | $23.47_{\pm0.16}$ | $24.89_{\pm0.17}$ | $15.51_{\pm0.10}$ | $16.74_{\pm0.12}$ | $18.55_{\pm0.14}$ | $21.17_{\pm0.16}$ | $5.62_{\pm0.01}$ | $5.84_{\pm0.01}$ | $6.07_{\pm0.02}$ | $6.26_{\pm0.02}$ |
| | +GPT2 | $3.55_{\pm0.01}$ | $3.66_{\pm0.01}$ | $3.78_{\pm0.02}$ | $3.90_{\pm0.02}$ | $20.72_{\pm0.15}$ | $22.85_{\pm0.16}$ | $24.41_{\pm0.18}$ | $25.84_{\pm0.19}$ | $15.69_{\pm0.12}$ | $16.82_{\pm0.14}$ | $18.67_{\pm0.16}$ | $21.49_{\pm0.18}$ | $5.65_{\pm0.01}$ | $5.89_{\pm0.02}$ | $6.13_{\pm0.02}$ | $6.37_{\pm0.03}$ |
| | +MAE | $3.57_{\pm0.02}$ | $3.67_{\pm0.02}$ | $3.79_{\pm0.02}$ | $3.89_{\pm0.02}$ | $20.94_{\pm0.16}$ | $23.01_{\pm0.16}$ | $24.54_{\pm0.16}$ | $25.78_{\pm0.20}$ | $15.78_{\pm0.14}$ | $16.96_{\pm0.15}$ | $18.59_{\pm0.16}$ | $21.30_{\pm0.17}$ | $5.67_{\pm0.01}$ | $5.97_{\pm0.02}$ | $6.15_{\pm0.02}$ | $6.34_{\pm0.02}$ |
| | +SPIN | $3.54_{\pm0.01}$ | $3.62_{\pm0.01}$ | $3.81_{\pm0.02}$ | $3.93_{\pm0.02}$ | $20.53_{\pm0.13}$ | $22.62_{\pm0.15}$ | $24.56_{\pm0.16}$ | $26.07_{\pm0.18}$ | $15.57_{\pm0.12}$ | $16.77_{\pm0.14}$ | $18.74_{\pm0.16}$ | $21.85_{\pm0.17}$ | $5.63_{\pm0.01}$ | $5.86_{\pm0.02}$ | $6.11_{\pm0.02}$ | $6.39_{\pm0.03}$ |
| | raw | $3.62_{\pm0.02}$ | $3.78_{\pm0.02}$ | $3.95_{\pm0.04}$ | $4.06_{\pm0.04}$ | $21.53_{\pm0.20}$ | $26.39_{\pm0.22}$ | $29.18_{\pm0.25}$ | $31.42_{\pm0.27}$ | $16.33_{\pm0.15}$ | $18.44_{\pm0.17}$ | $20.59_{\pm0.20}$ | $22.98_{\pm0.22}$ | $5.77_{\pm0.02}$ | $6.01_{\pm0.02}$ | $6.31_{\pm0.04}$ | $6.63_{\pm0.04}$ |
| DSformer | +Merlin | $3.54_{\pm0.01}$ | $3.66_{\pm0.01}$ | $3.74_{\pm0.02}$ | $3.88_{\pm0.02}$ | $20.54_{\pm0.13}$ | $22.18_{\pm0.14}$ | $22.74_{\pm0.16}$ | $24.47_{\pm0.17}$ | $15.50_{\pm0.10}$ | $16.39_{\pm0.12}$ | $18.45_{\pm0.13}$ | $21.16_{\pm0.15}$ | $5.57_{\pm0.01}$ | $5.76_{\pm0.01}$ | $5.92_{\pm0.02}$ | $6.18_{\pm0.02}$ |
| | +GATGPT | $3.58_{\pm0.01}$ | $3.70_{\pm0.01}$ | $3.84_{\pm0.02}$ | $3.96_{\pm0.02}$ | $20.86_{\pm0.15}$ | $22.54_{\pm0.16}$ | $23.26_{\pm0.18}$ | $24.68_{\pm0.20}$ | $15.71_{\pm0.12}$ | $17.04_{\pm0.14}$ | $19.15_{\pm0.16}$ | $21.71_{\pm0.18}$ | $5.63_{\pm0.01}$ | $5.82_{\pm0.02}$ | $6.01_{\pm0.02}$ | $6.24_{\pm0.03}$ |
| | +GPT2 | $3.61_{\pm0.01}$ | $3.75_{\pm0.01}$ | $3.88_{\pm0.02}$ | $4.01_{\pm0.02}$ | $21.04_{\pm0.16}$ | $22.92_{\pm0.18}$ | $24.06_{\pm0.19}$ | $25.40_{\pm0.21}$ | $15.82_{\pm0.13}$ | $17.15_{\pm0.15}$ | $19.28_{\pm0.17}$ | $21.89_{\pm0.19}$ | $5.65_{\pm0.01}$ | $5.85_{\pm0.02}$ | $6.07_{\pm0.02}$ | $6.29_{\pm0.04}$ |
| | +MAE | $3.63_{\pm0.02}$ | $3.77_{\pm0.02}$ | $3.90_{\pm0.02}$ | $3.99_{\pm0.02}$ | $21.27_{\pm0.16}$ | $23.01_{\pm0.17}$ | $24.11_{\pm0.19}$ | $25.26_{\pm0.21}$ | $15.87_{\pm0.14}$ | $17.24_{\pm0.16}$ | $19.31_{\pm0.18}$ | $21.77_{\pm0.20}$ | $5.70_{\pm0.01}$ | $5.92_{\pm0.02}$ | $6.09_{\pm0.03}$ | $6.26_{\pm0.03}$ |
| | +SPIN | $3.59_{\pm0.01}$ | $3.72_{\pm0.01}$ | $3.87_{\pm0.02}$ | $4.03_{\pm0.02}$ | $20.97_{\pm0.15}$ | $22.87_{\pm0.17}$ | $24.23_{\pm0.19}$ | $25.73_{\pm0.22}$ | $15.76_{\pm0.13}$ | $17.08_{\pm0.14}$ | $19.23_{\pm0.16}$ | $21.98_{\pm0.19}$ | $5.67_{\pm0.01}$ | $5.83_{\pm0.02}$ | $6.05_{\pm0.04}$ | $6.31_{\pm0.04}$ |
| | raw | $3.72_{\pm0.02}$ | $3.87_{\pm0.02}$ | $3.95_{\pm0.04}$ | $4.11_{\pm0.04}$ | $23.24_{\pm0.21}$ | $27.85_{\pm0.23}$ | $30.47_{\pm0.23}$ | $33.25_{\pm0.25}$ | $16.52_{\pm0.15}$ | $18.75_{\pm0.16}$ | $20.96_{\pm0.18}$ | $23.47_{\pm0.21}$ | $5.75_{\pm0.02}$ | $5.98_{\pm0.02}$ | $6.25_{\pm0.04}$ | $6.57_{\pm0.04}$ |
| FourierGNN | +Merlin | $3.50_{\pm0.01}$ | $3.57_{\pm0.01}$ | $3.68_{\pm0.02}$ | $3.80_{\pm0.02}$ | $19.67_{\pm0.13}$ | $20.79_{\pm0.14}$ | $22.25_{\pm0.14}$ | $23.92_{\pm0.15}$ | $15.32_{\pm0.09}$ | $16.22_{\pm0.11}$ | $17.92_{\pm0.13}$ | $20.38_{\pm0.14}$ | $5.55_{\pm0.01}$ | $5.70_{\pm0.01}$ | $5.91_{\pm0.02}$ | $6.10_{\pm0.02}$ |
| | +GATGPT | $3.52_{\pm0.01}$ | $3.61_{\pm0.01}$ | $3.73_{\pm0.02}$ | $3.85_{\pm0.02}$ | $20.08_{\pm0.13}$ | $21.19_{\pm0.15}$ | $22.68_{\pm0.15}$ | $24.05_{\pm0.17}$ | $15.47_{\pm0.11}$ | $16.42_{\pm0.12}$ | $18.27_{\pm0.14}$ | $20.84_{\pm0.15}$ | $5.60_{\pm0.01}$ | $5.73_{\pm0.01}$ | $5.95_{\pm0.02}$ | $6.15_{\pm0.02}$ |
| | +GPT2 | $3.55_{\pm0.01}$ | $3.65_{\pm0.01}$ | $3.76_{\pm0.02}$ | $3.89_{\pm0.02}$ | $20.35_{\pm0.15}$ | $21.94_{\pm0.16}$ | $22.84_{\pm0.18}$ | $24.24_{\pm0.20}$ | $15.63_{\pm0.12}$ | $16.54_{\pm0.14}$ | $18.35_{\pm0.16}$ | $20.97_{\pm0.18}$ | $5.61_{\pm0.01}$ | $5.76_{\pm0.02}$ | $5.98_{\pm0.02}$ | $6.18_{\pm0.03}$ |
| | +MAE | $3.57_{\pm0.02}$ | $3.69_{\pm0.02}$ | $3.80_{\pm0.02}$ | $3.87_{\pm0.02}$ | $20.32_{\pm0.15}$ | $22.05_{\pm0.17}$ | $23.02_{\pm0.18}$ | $24.19_{\pm0.19}$ | $15.69_{\pm0.13}$ | $16.58_{\pm0.15}$ | $18.39_{\pm0.16}$ | $20.89_{\pm0.17}$ | $5.64_{\pm0.01}$ | $5.78_{\pm0.02}$ | $5.96_{\pm0.02}$ | $6.17_{\pm0.03}$ |
| | +SPIN | $3.54_{\pm0.01}$ | $3.63_{\pm0.01}$ | $3.74_{\pm0.02}$ | $3.91_{\pm0.02}$ | $20.14_{\pm0.14}$ | $21.87_{\pm0.15}$ | $22.97_{\pm0.18}$ | $24.35_{\pm0.19}$ | $15.56_{\pm0.12}$ | $16.46_{\pm0.14}$ | $18.31_{\pm0.15}$ | $21.02_{\pm0.17}$ | $5.63_{\pm0.01}$ | $5.75_{\pm0.02}$ | $6.01_{\pm0.02}$ | $6.21_{\pm0.03}$ |
| | raw | $3.61_{\pm0.02}$ | $3.77_{\pm0.02}$ | $3.92_{\pm0.04}$ | $4.05_{\pm0.04}$ | $21.34_{\pm0.18}$ | $24.58_{\pm0.20}$ | $27.05_{\pm0.22}$ | $29.71_{\pm0.24}$ | $15.98_{\pm0.15}$ | $17.69_{\pm0.16}$ | $19.13_{\pm0.18}$ | $21.57_{\pm0.21}$ | $5.73_{\pm0.02}$ | $5.93_{\pm0.02}$ | $6.15_{\pm0.04}$ | $6.39_{\pm0.04}$ |

Table 14: MAE values of Merlin and other methods (The missing rate of the training set is 10%).

| Backbone | Methods | METR-LA | | | | PEMS04 | | | | China AQI | | | | Global Wind | | | |
|---|---|---|---|---|---|---|---|---|---|---|---|---|---|---|---|---|---|
| | | 25% | 50% | 75% | 90% | 25% | 50% | 75% | 90% | 25% | 50% | 75% | 90% | 25% | 50% | 75% | 90% |
| STID | +Merlin | $3.42_{\pm0.01}$ | $3.57_{\pm0.01}$ | $3.66_{\pm0.02}$ | $3.75_{\pm0.02}$ | $19.41_{\pm0.10}$ | $20.39_{\pm0.11}$ | $21.81_{\pm0.13}$ | $23.64_{\pm0.13}$ | $15.23_{\pm0.08}$ | $15.94_{\pm0.10}$ | $17.35_{\pm0.11}$ | $19.25_{\pm0.13}$ | $5.55_{\pm0.01}$ | $5.64_{\pm0.01}$ | $5.85_{\pm0.02}$ | $6.10_{\pm0.02}$ |
| | +GATGPT | $3.49_{\pm0.01}$ | $3.62_{\pm0.01}$ | $3.73_{\pm0.02}$ | $3.85_{\pm0.02}$ | $20.05_{\pm0.12}$ | $21.43_{\pm0.14}$ | $22.88_{\pm0.14}$ | $24.31_{\pm0.16}$ | $15.41_{\pm0.10}$ | $16.24_{\pm0.12}$ | $17.79_{\pm0.13}$ | $19.78_{\pm0.15}$ | $5.60_{\pm0.01}$ | $5.78_{\pm0.01}$ | $5.96_{\pm0.02}$ | $6.21_{\pm0.02}$ |
| | +GPT2 | $3.52_{\pm0.01}$ | $3.67_{\pm0.01}$ | $3.76_{\pm0.02}$ | $3.90_{\pm0.02}$ | $20.33_{\pm0.14}$ | $22.08_{\pm0.16}$ | $23.14_{\pm0.17}$ | $24.48_{\pm0.17}$ | $15.46_{\pm0.12}$ | $16.38_{\pm0.14}$ | $17.93_{\pm0.15}$ | $20.03_{\pm0.16}$ | $5.61_{\pm0.01}$ | $5.80_{\pm0.02}$ | $6.01_{\pm0.02}$ | $6.27_{\pm0.03}$ |
| | +MAE | $3.53_{\pm0.02}$ | $3.68_{\pm0.02}$ | $3.77_{\pm0.02}$ | $3.88_{\pm0.02}$ | $20.51_{\pm0.15}$ | $21.84_{\pm0.17}$ | $22.95_{\pm0.18}$ | $24.37_{\pm0.20}$ | $15.51_{\pm0.12}$ | $16.78_{\pm0.14}$ | $18.47_{\pm0.14}$ | $19.87_{\pm0.15}$ | $5.62_{\pm0.01}$ | $5.83_{\pm0.02}$ | $5.99_{\pm0.02}$ | $6.20_{\pm0.02}$ |
| | +SPIN | $3.49_{\pm0.01}$ | $3.65_{\pm0.01}$ | $3.79_{\pm0.02}$ | $3.91_{\pm0.02}$ | $20.14_{\pm0.13}$ | $21.71_{\pm0.15}$ | $23.31_{\pm0.15}$ | $24.64_{\pm0.17}$ | $15.39_{\pm0.12}$ | $16.57_{\pm0.13}$ | $18.56_{\pm0.15}$ | $20.27_{\pm0.15}$ | $5.59_{\pm0.01}$ | $5.79_{\pm0.02}$ | $6.07_{\pm0.02}$ | $6.29_{\pm0.03}$ |
| | raw | $3.54_{\pm0.02}$ | $3.77_{\pm0.02}$ | $3.93_{\pm0.04}$ | $4.07_{\pm0.04}$ | $20.67_{\pm0.19}$ | $28.36_{\pm0.21}$ | $30.11_{\pm0.22}$ | $33.65_{\pm0.25}$ | $15.53_{\pm0.14}$ | $18.56_{\pm0.16}$ | $20.36_{\pm0.19}$ | $23.24_{\pm0.21}$ | $5.63_{\pm0.01}$ | $6.05_{\pm0.02}$ | $6.34_{\pm0.04}$ | $6.68_{\pm0.04}$ |
| TSmixer | +Merlin | $3.51_{\pm0.01}$ | $3.64_{\pm0.01}$ | $3.75_{\pm0.02}$ | $3.88_{\pm0.02}$ | $20.11_{\pm0.12}$ | $22.29_{\pm0.13}$ | $23.15_{\pm0.14}$ | $24.82_{\pm0.15}$ | $15.77_{\pm0.09}$ | $16.58_{\pm0.11}$ | $18.48_{\pm0.12}$ | $21.03_{\pm0.14}$ | $5.62_{\pm0.01}$ | $5.86_{\pm0.01}$ | $6.07_{\pm0.02}$ | $6.27_{\pm0.02}$ |
| | +GATGPT | $3.55_{\pm0.01}$ | $3.68_{\pm0.01}$ | $3.80_{\pm0.02}$ | $3.93_{\pm0.02}$ | $20.79_{\pm0.13}$ | $22.75_{\pm0.14}$ | $23.81_{\pm0.16}$ | $25.13_{\pm0.17}$ | $15.86_{\pm0.10}$ | $16.92_{\pm0.12}$ | $18.79_{\pm0.14}$ | $21.42_{\pm0.16}$ | $5.67_{\pm0.01}$ | $5.89_{\pm0.01}$ | $6.14_{\pm0.02}$ | $6.31_{\pm0.02}$ |
| | +GPT2 | $3.58_{\pm0.01}$ | $3.71_{\pm0.01}$ | $3.84_{\pm0.02}$ | $3.96_{\pm0.02}$ | $21.03_{\pm0.15}$ | $23.12_{\pm0.16}$ | $24.73_{\pm0.18}$ | $26.16_{\pm0.19}$ | $15.97_{\pm0.12}$ | $17.05_{\pm0.14}$ | $18.81_{\pm0.16}$ | $21.65_{\pm0.18}$ | $5.65_{\pm0.01}$ | $5.94_{\pm0.02}$ | $6.18_{\pm0.02}$ | $6.41_{\pm0.03}$ |
| | +MAE | $3.60_{\pm0.02}$ | $3.72_{\pm0.02}$ | $3.83_{\pm0.02}$ | $3.95_{\pm0.02}$ | $20.87_{\pm0.15}$ | $22.95_{\pm0.15}$ | $24.94_{\pm0.16}$ | $26.29_{\pm0.18}$ | $15.92_{\pm0.12}$ | $16.99_{\pm0.14}$ | $18.92_{\pm0.16}$ | $22.03_{\pm0.17}$ | $5.63_{\pm0.01}$ | $5.91_{\pm0.02}$ | $6.22_{\pm0.02}$ | $6.45_{\pm0.03}$ |
| | +SPIN | $3.57_{\pm0.01}$ | $3.67_{\pm0.01}$ | $3.86_{\pm0.02}$ | $3.98_{\pm0.02}$ | $20.87_{\pm0.13}$ | $22.95_{\pm0.15}$ | $24.94_{\pm0.16}$ | $26.29_{\pm0.18}$ | $16.33_{\pm0.13}$ | $16.99_{\pm0.14}$ | $18.92_{\pm0.16}$ | $22.03_{\pm0.17}$ | $5.77_{\pm0.02}$ | $5.91_{\pm0.02}$ | $6.22_{\pm0.02}$ | $6.45_{\pm0.03}$ |
| | raw | $3.62_{\pm0.02}$ | $3.78_{\pm0.02}$ | $3.95_{\pm0.04}$ | $4.06_{\pm0.04}$ | $21.53_{\pm0.20}$ | $26.39_{\pm0.22}$ | $29.18_{\pm0.25}$ | $31.42_{\pm0.27}$ | $16.33_{\pm0.15}$ | $18.44_{\pm0.17}$ | $20.59_{\pm0.20}$ | $22.98_{\pm0.22}$ | $5.77_{\pm0.02}$ | $6.01_{\pm0.02}$ | $6.31_{\pm0.04}$ | $6.63_{\pm0.04}$ |
| DSformer | +Merlin | $3.58_{\pm0.01}$ | $3.71_{\pm0.01}$ | $3.78_{\pm0.02}$ | $3.94_{\pm0.02}$ | $20.82_{\pm0.13}$ | $22.50_{\pm0.14}$ | $23.07_{\pm0.16}$ | $24.84_{\pm0.17}$ | $15.73_{\pm0.10}$ | $16.62_{\pm0.12}$ | $18.73_{\pm0.13}$ | $21.38_{\pm0.15}$ | $5.63_{\pm0.01}$ | $5.83_{\pm0.01}$ | $5.99_{\pm0.02}$ | $6.25_{\pm0.02}$ |
| | +GATGPT | $3.62_{\pm0.01}$ | $3.76_{\pm0.01}$ | $3.89_{\pm0.02}$ | $4.01_{\pm0.02}$ | $21.14_{\pm0.15}$ | $22.90_{\pm0.16}$ | $23.59_{\pm0.18}$ | $25.06_{\pm0.20}$ | $15.96_{\pm0.12}$ | $17.31_{\pm0.14}$ | $19.39_{\pm0.16}$ | $21.95_{\pm0.18}$ | $5.66_{\pm0.01}$ | $5.87_{\pm0.02}$ | $6.08_{\pm0.02}$ | $6.30_{\pm0.03}$ |
| | +GPT2 | $3.65_{\pm0.01}$ | $3.79_{\pm0.01}$ | $3.92_{\pm0.02}$ | $4.05_{\pm0.02}$ | $21.36_{\pm0.16}$ | $23.35_{\pm0.18}$ | $24.34_{\pm0.19}$ | $25.71_{\pm0.21}$ | $16.09_{\pm0.13}$ | $17.33_{\pm0.15}$ | $19.54_{\pm0.17}$ | $22.14_{\pm0.19}$ | $5.68_{\pm0.01}$ | $5.90_{\pm0.02}$ | $6.11_{\pm0.03}$ | $6.35_{\pm0.04}$ |
| | +MAE | $3.66_{\pm0.02}$ | $3.80_{\pm0.02}$ | $3.93_{\pm0.02}$ | $4.04_{\pm0.02}$ | $21.52_{\pm0.16}$ | $23.35_{\pm0.17}$ | $24.42_{\pm0.19}$ | $25.58_{\pm0.21}$ | $16.15_{\pm0.14}$ | $17.41_{\pm0.16}$ | $19.61_{\pm0.18}$ | $22.03_{\pm0.20}$ | $5.72_{\pm0.01}$ | $5.96_{\pm0.02}$ | $6.14_{\pm0.03}$ | $6.31_{\pm0.03}$ |
| | +SPIN | $3.63_{\pm0.01}$ | $3.78_{\pm0.01}$ | $3.90_{\pm0.02}$ | $4.07_{\pm0.02}$ | $21.19_{\pm0.15}$ | $23.19_{\pm0.17}$ | $24.56_{\pm0.19}$ | $26.01_{\pm0.22}$ | $16.04_{\pm0.13}$ | $17.24_{\pm0.14}$ | $19.49_{\pm0.16}$ | $22.26_{\pm0.19}$ | $5.69_{\pm0.01}$ | $5.8_{\pm0.02}$ | $6.12_{\pm0.04}$ | $6.38_{\pm0.04}$ |
| | raw | $3.72_{\pm0.02}$ | $3.87_{\pm0.02}$ | $3.95_{\pm0.04}$ | $4.11_{\pm0.04}$ | $23.24_{\pm0.21}$ | $27.85_{\pm0.23}$ | $30.47_{\pm0.23}$ | $33.25_{\pm0.25}$ | $16.52_{\pm0.15}$ | $18.75_{\pm0.16}$ | $20.96_{\pm0.18}$ | $23.47_{\pm0.21}$ | $5.75_{\pm0.02}$ | $5.98_{\pm0.02}$ | $6.25_{\pm0.04}$ | $6.57_{\pm0.04}$ |
| FourierGNN | +Merlin | $3.54_{\pm0.01}$ | $3.62_{\pm0.01}$ | $3.72_{\pm0.02}$ | $3.85_{\pm0.02}$ | $19.97_{\pm0.13}$ | $21.05_{\pm0.12}$ | $22.54_{\pm0.13}$ | $24.26_{\pm0.15}$ | $15.58_{\pm0.09}$ | $16.47_{\pm0.11}$ | $18.14_{\pm0.13}$ | $20.63_{\pm0.14}$ | $5.59_{\pm0.01}$ | $5.75_{\pm0.01}$ | $5.97_{\pm0.02}$ | $6.16_{\pm0.02}$ |
| | +GATGPT | $3.56_{\pm0.01}$ | $3.65_{\pm0.01}$ | $3.79_{\pm0.02}$ | $3.91_{\pm0.02}$ | $20.37_{\pm0.13}$ | $21.55_{\pm0.15}$ | $23.03_{\pm0.15}$ | $24.43_{\pm0.17}$ | $15.71_{\pm0.11}$ | $16.68_{\pm0.12}$ | $18.51_{\pm0.14}$ | $21.06_{\pm0.15}$ | $5.65_{\pm0.01}$ | $5.79_{\pm0.01}$ | $6.01_{\pm0.02}$ | $6.21_{\pm0.02}$ |
| | +GPT2 | $3.58_{\pm0.01}$ | $3.68_{\pm0.01}$ | $3.80_{\pm0.02}$ | $3.96_{\pm0.02}$ | $20.54_{\pm0.14}$ | $22.27_{\pm0.16}$ | $23.25_{\pm0.18}$ | $24.63_{\pm0.20}$ | $15.81_{\pm0.12}$ | $16.79_{\pm0.14}$ | $18.64_{\pm0.16}$ | $21.28_{\pm0.18}$ | $5.67_{\pm0.01}$ | $5.82_{\pm0.02}$ | $6.04_{\pm0.02}$ | $6.23_{\pm0.03}$ |
| | +MAE | $3.59_{\pm0.02}$ | $3.72_{\pm0.02}$ | $3.83_{\pm0.02}$ | $3.92_{\pm0.02}$ | $20.63_{\pm0.15}$ | $22.34_{\pm0.17}$ | $23.38_{\pm0.18}$ | $24.52_{\pm0.19}$ | $15.86_{\pm0.13}$ | $16.89_{\pm0.15}$ | $18.62_{\pm0.16}$ | $21.15_{\pm0.17}$ | $5.68_{\pm0.01}$ | $5.84_{\pm0.02}$ | $6.03_{\pm0.02}$ | $6.23_{\pm0.03}$ |
| | +SPIN | $3.57_{\pm0.01}$ | $3.66_{\pm0.01}$ | $3.79_{\pm0.02}$ | $3.97_{\pm0.02}$ | $20.44_{\pm0.14}$ | $22.15_{\pm0.15}$ | $23.34_{\pm0.18}$ | $24.75_{\pm0.19}$ | $15.75_{\pm0.12}$ | $16.71_{\pm0.14}$ | $18.58_{\pm0.15}$ | $21.34_{\pm0.17}$ | $5.66_{\pm0.01}$ | $5.81_{\pm0.02}$ | $6.07_{\pm0.02}$ | $6.27_{\pm0.03}$ |
| | raw | $3.61_{\pm0.02}$ | $3.77_{\pm0.02}$ | $3.92_{\pm0.04}$ | $4.05_{\pm0.04}$ | $21.34_{\pm0.18}$ | $24.58_{\pm0.20}$ | $27.05_{\pm0.22}$ | $29.71_{\pm0.24}$ | $15.98_{\pm0.15}$ | $17.69_{\pm0.16}$ | $19.13_{\pm0.18}$ | $21.57_{\pm0.21}$ | $5.73_{\pm0.02}$ | $5.93_{\pm0.02}$ | $6.15_{\pm0.04}$ | $6.39_{\pm0.04}$ |

# K COMPARED WITH END-TO-END MODELS THAT CAN HANDLE MISSING DATA

The experimental results in Section 4.2 (Main Results) and Section 4.3 (Transferability of Merlin) demonstrate that Merlin achieves superior predictive performance compared to two-stage models. To further validate the model's performance, we compared Merlin with several existing end-to-end models that can handle missing data. All models are introduced as follows:

- **MGSFformer** (Yu et al., 2024b): This model introduces residual redundancy reduction blocks, spatiotemporal attention blocks, and dynamic fusion blocks to achieve multivariate time series forecasting (MTSF).
- **S4** (Gu et al., 2022): It proposes a fundamental state space model to achieve accurate MTSF.
- **GinAR** (Yu et al., 2024a): This model incorporates interpolation attention and adaptive graph learning to enhance its performance in MTSF with missing data.

Table 15 show the RMSE values of several models. Based on the experimental results, it can be observed that compared with end-to-end models that can handle missing data, the proposed model still achieves better forecasting performance.

Table 15: RMSE values of several models (The best results are shown in **bold**).

| Datasets | Methods | Missing rates | | | |
|---|---|---|---|---|---|
| | | 25% | 50% | 75% | 90% |
| METR-LA | Proposed | $\mathbf{6.58}_{\pm 0.02}$ | $\mathbf{6.65}_{\pm 0.02}$ | $\mathbf{6.81}_{\pm 0.04}$ | $\mathbf{7.06}_{\pm 0.04}$ |
| | GinAR | $6.72_{\pm 0.02}$ | $6.91_{\pm 0.04}$ | $7.38_{\pm 0.04}$ | $7.67_{\pm 0.04}$ |
| | MGSFformer | $6.78_{\pm 0.04}$ | $6.98_{\pm 0.04}$ | $7.45_{\pm 0.04}$ | $7.84_{\pm 0.06}$ |
| | S4 | $7.13_{\pm 0.04}$ | $7.54_{\pm 0.04}$ | $7.82_{\pm 0.06}$ | $8.16_{\pm 0.08}$ |
| PEMS04 | Proposed | $\mathbf{30.67}_{\pm 0.13}$ | $\mathbf{31.41}_{\pm 0.15}$ | $\mathbf{33.38}_{\pm 0.16}$ | $\mathbf{36.27}_{\pm 0.17}$ |
| | GinAR | $32.15_{\pm 0.16}$ | $34.27_{\pm 0.18}$ | $35.86_{\pm 0.21}$ | $38.19_{\pm 0.22}$ |
| | MGSFformer | $32.78_{\pm 0.19}$ | $36.43_{\pm 0.21}$ | $39.21_{\pm 0.23}$ | $40.16_{\pm 0.24}$ |
| | S4 | $35.23_{\pm 0.24}$ | $40.17_{\pm 0.25}$ | $43.06_{\pm 0.27}$ | $45.58_{\pm 0.29}$ |

