# OpenReview forum: "Merlin: Multi-View Representation Learning for Robust Multivariate Time Series Forecasting with Unfixed Missing Rates"
_ICLR.cc/2025/Conference — Submitted to ICLR 2025_

### Official Review · Reviewer_odRh · 2024-10-30

**Soundness:** 2
**Presentation:** 1
**Contribution:** 2
**Rating:** 3
**Confidence:** 4

**Summary:**

This paper deals with multivariate time series forecasting (MTSF) with missing values and different missing rates. The paper proposed an interesting method called Merlin, which incorporates offline knowledge distillation and multi-view contrastive learning in the spatio-temporal identity embedding (STID) architecture. The idea of Merlin is not dependent on the STID architecture and can be transferred to any other models. Merlin learns better semantics from sparse observations using offline knowledge distillation. Merlin comprises a teacher and a student network, both based on STID architecture, where the teacher is trained with complete observations and the student is trained with sparse ones. The distillation loss between the teacher and student network helps the student network learn better semantics from the sparse observations. Merlin is further aided by multi-view contrastive learning, which helps the student network align semantics across sparse observations with different missing rates.

**Strengths:**

1. Merlin is an interesting approach to remove the need for imputation in MTSF.
2. The authors, to some extent, demonstrate the problem with imputation models as the missing rate increases in the data. This is an important insight to present experimentally, which can benefit the field of MTSF.
3. The writing of the paper is straightforward and easy to follow.

**Weaknesses:**

**Major Comments:**
1. The paper argues that the performance of baseline models declines substantially as the missing rate increases, as illustrated in Figure 1. However,
    1. It is difficult to compare the models in Figure 1(a) in terms of the decline value. For example, what is the decline in performance for STID+SPIN compared to MERLIN from 50%$ to 75% and 75% to 100%? It looks very close in Figure 1(a). It seems that the percentage decline of STID+SPIN is smaller compared to MERLIN from 75% to 100%. Please provide the exact decline percentage somewhere in the paper, even if it is in the supplementary.
    2. Is there any specific reason for the chosen baseline models presented in Figure 1(a)? As shown in Table 1, STID+GATGPT and STID+MAE are better baselines in terms of performance. Why weren’t they considered for the comparison in Figure 1(a)?
    3. This observation is only shown in one dataset, PEMS04. Can the authors also present a similar trend in at least one more dataset? Showing a conclusion in only one dataset is not convincing.
2. The author mentions about two recent baseline methods in related works [1, 2] in Lines 147 and 155. However, Merlin is not compared against them. Why?
3. In the last line of section 2.2, the author mentions, “As a result, the student model’s ability to mine key semantics from sparse observations is effectively enhanced.” To support this statement, the author cites some papers. Among them, the author argues that [3] and [4] show satisfactory results using knowledge distillation even with 50% and 25% observations, respectively. If I am not wrong, [3] and [4] show their efficacy by using fewer complete observations. Spare and fewer observations are two completely different terms and concepts. It feels like the related work here is on fewer observations, however, the conclusion is made for sparse observations. The authors should justify this.
4. One of the main contributions of the paper is stated in Line 123, “we refine that robustness is the main factor contributing to the poor performance of forecasting models in MTSF with sparse observations.” I miss a justification for this contribution, either experimentally or theoretically. The authors argue that due to unfixed missing rates, the existing models give poor robustness. However, this is not a strong justification. The authors need to show this with some kind of experiment. An experiment can be done to show the performance of the baseline models, trained only once, with all unfixed missing rates. The authors can think of other appropriate experiments, too.
5. The paper uses the word semantic in multiple places. To give a few examples, semantic alignment, extract semantics, align semantics, semantic mining capabilities, key semantics, disrupt the semantics, incorrect semantics, capture the semantics, error semantics, local semantics, enhance the semantic differences, semantic similarities, high-quality semantics, additional semantic information, etc. I think a reader will benefit if the meaning of the semantics is properly defined in the paper, expanding or limiting its scope to the field of MTSF.
6. The paper uses knowledge distillation between sparse and complete observations and contrastive learning among sparse observations. It would be interesting to get insights on using complete observations for contrastive learning. The complete observations can also be an input to the student model and subsequently used in contrastive learning.
7. In step II, section 3.2, the authors mention about spatial-temporal identity embeddings, namely, $S_E, T^D_E,$ and $T^W_E$. However, there is no information about how these embeddings are generated. This is an integral part of the Merlin architecture. Therefore, the authors should discuss them in detail.
8. It is mentioned in section 3.3 that the sparse input observations are passed to the fully connected layers to generate embeddings. How are the missing values handled there? Are they just considered as zeros?
9. For offline knowledge distillation, especially the $L_{HD}$ loss, what is the rationale behind using mean squared error (MSE) loss? The most common type of loss used in this scenario is the KL-Divergence loss. How does KL-Divergence fare in comparison to MSE in Merlin?
10. The authors mention in lines 351 and 352 that “After testing, it has been found that the performance is optimal when the weight of each loss is set to 1”. However, I could not find any experiment in the rest of the paper which shows the mentioned finding.
11. The paper does not report any appropriate information about the statistical significance of the experiments. At the very least, authors should report the standard deviation along with the mean.
12. The training difference between the Merlin and baseline models is unclear. The paper in lines 422-424 states, “To demonstrate the robustness of the proposed model, we only train it once, using samples with different missing rates. For other baselines, we separately train a model for each missing rate.”  Do the authors mean that in Merlin, they incorporate the data with each missing rate in one model? That is, the authors make 3 more versions of the data with 50%, 75%, and 90% missing rates. Merlin is trained on all these 4 datasets combined. Meanwhile, the baseline model is only trained on one of the missing rates datasets individually. If that is the case, then the comparison between Merlin and baseline models is not fair since Merlin sees four times more data compared to the baseline models.
13. Section C in the Appendix presents hyperparameter analysis with the hyperparameter values used in STID+Merlin. However, there is no discussion of the process of hyperparameter searching. Additionally, the authors should also provide the hyperparameters and their search process for the baseline models. This will help in the reproducibility of the paper.
14. Figure 4 in the Appendix presents an experiment on determining two hyperparameters of Merlin on the China AQI dataset. These hyperparameter values are then also reported as the best values in Table 5. Does this mean that similar conclusions are seen in other datasets? Please provide this information in the manuscript.

**Minor Comments:**
1. The citation type is not appropriate. The citations should be written within brackets in most of the cases in this article. Therefore, as a reader, it became very difficult to read this paper.
2. The paper has several grammatical mistakes/typos. Here are some of them:
    1. Line 87: “…usually require reconstructing reconstruct both missing…”
    2. Line 116: “…adapt to unfixes missing…”
    3. Line 119 and 309: “…different miss rates…”
    4. Line 143: “gated recursive unit (GRU).” It should be recurrent.
    5. Line 164: “Knowledge distillation can transfer valuable Knowledge from the teacher model to the student model.” Shouldn’t the second knowledge in this line be in small letters?
    6. Line 267: “…abbove…”
    7. Line 348: “Finally, we need to effectively integrate above all Loss functions.”
    8. Line 350: “…we adopts the..”
3. What does “sudden change” mean in line 73? Do the authors mean the line drop in Figure 1(b)? The author should clarify this because there can be other reasons for sudden change, like concept drift.
4. In the paper, the authors are inconsistent with their use of capital or small letters for the full form of abbreviations. Please be consistent. Some of the examples are:
    1. graph convolutional network (GCN)
    2. gated recursive unit (GRU)
    3. Graph WaveNet (GWNet)
    4. Temporal Convolutional Network (TCN)
    5. Multi-Layer Perceptron (MLP)
5. The paper introduces a notation $N_H$ in line 210. However, the meaning of $N_H$ is not present in the subsequent text. The authors provide its meaning in Appendix A. However, for ease of reading, the author should provide the meaning in line 210. The same practice should be followed for other notations, too.

**References:**
1. Chengqing Yu, Fei Wang, Zezhi Shao, Tangwen Qian, Zhao Zhang, Wei Wei, and Yongjun Xu. Ginar: An end-to-end multivariate time series forecasting model suitable for variable missing. arXiv preprint arXiv:2405.11333, 2024a.
2. Chengqing Yu, FeiWang, YilunWang, Zezhi Shao, Tao Sun, Di Yao, and Yongjun Xu. Mgsfformer: A multi-granularity spatiotemporal fusion transformer for air quality prediction. Information Fusion, pp. 102607, 2024b. ISSN 1566-2535. doi: https://doi.org/10.1016/j.inffus.2024.102607.
3. Muhammad Ali Chattha, Ludger van Elst, Muhammad Imran Malik, Andreas Dengel, and Sheraz Ahmed. Kenn: Enhancing deep neural networks by leveraging knowledge for time series forecasting. arXiv preprint arXiv:2202.03903, 2022.
4. Alessio Monti, Angelo Porrello, Simone Calderara, Pasquale Coscia, Lamberto Ballan, and Rita Cucchiara. How many observations are enough? knowledge distillation for trajectory forecasting. In Proceedings of the IEEE/CVF Conference on Computer Vision and Pattern Recognition, pp. 6553–6562, 2022.

**Questions:**

All the questions are listed in the weakness section.

---

> ### Author Response · Authors · 2024-11-21
> **Rebuttal by authors**
>
> Dear Reviewer odRh:
>
> Thank you very much for your constructive review comments. Below are our responses. We hope to resolve all your concerns.
>
> 1. **W1:** The paper argues that the performance of baseline models declines substantially as the missing rate increases, as illustrated in Figure 1. However,
> （1）It is difficult to compare the models in Figure 1(a) in terms of the decline value. For example, what is the decline in performance for STID+SPIN compared to MERLIN from 50%$ to 75% and 75% to 100%? It looks very close in Figure 1(a). It seems that the percentage decline of STID+SPIN is smaller compared to MERLIN from 75% to 100%. Please provide the exact decline percentage somewhere in the paper, even if it is in the supplementary.
> （2）Is there any specific reason for the chosen baseline models presented in Figure 1(a)? As shown in Table 1, STID+GATGPT and STID+MAE are better baselines in terms of performance. Why weren’t they considered for the comparison in Figure 1(a)?
> （3）This observation is only shown in one dataset, PEMS04. Can the authors also present a similar trend in at least one more dataset? Showing a conclusion in only one dataset is not convincing.
>
> **Reply:** Thank you for your valuable review comments. Apologies for the lack of rigor in Figure 1, which has caused some confusion for the reviewers. Specifically, the main purpose of Figure 1(a) is to demonstrate that the performance of existing prediction models consistently declines as the data missing rate increases. Regarding the choice of models, we randomly selected some of the currently well-performing works for demonstration. Based on the reviewers’ suggestions, we believe a more intuitive approach would help readers understand this better.
> To address this, we have revised Figure 1(a) to present the prediction error (Mean Absolute Error) curves of Merlin and other models. Additionally, we have provided an extra plot showing the prediction error curves of Merlin and other models on METR-LA (Figure 1(b)). From these two curves, it can be observed that, compared to other baselines, Merlin’s prediction error remains at a relatively low level. The specific changes in MAE can be found in Section 4.2 (Main Results).
> The revised Figure 1 has been submitted to the revised paper
>
>
> 2. **W2:** The author mentions about two recent baseline methods in related works [1, 2] in Lines 147 and 155. However, Merlin is not compared against them. Why?
>
> **Reply:** Thank you for your valuable review comments. When selecting baselines, the following two issues prevented us from including these works for comparison:
> (1) MGSFformer: This model relies on multi-granularity input, which is not available in the datasets we used.
> (2) GinAR: This model requires the predefined graph, which is absent in the China AQI and Global Wind datasets. Additionally, its high model complexity limits its ability to handle datasets like Global Wind effectively.
> Besides, our primary goal is to improve the robustness of forecasting models by proposing Merlin. Therefore, when designing the experiments, we focused on comparing imputation models, such as Merlin and other imputation models, in terms of their ability to enhance the performance of different forecasting models. However, we believe the reviewers' comments are very important. To further validate the model's performance, we compared Merlin with several existing end-to-end models that can handle missing data. The following are partial experimental results. More detailed experimental results can be found in the revised version of Appendix K.
>
> METR-LA
>
> Methods | RMSE values under different Missing rates (25%, 50%, 75%, 90%)
>
> Proposed | 6.58 | 6.65 | 6.81 | 7.06
>
> GinAR | 6.72 | 6.91 | 7.38| 7.67
>
> MGSFformer | 6.78 | 6.98 | 7.45 | 7.84
>
> S4 | 7.13 | 7.54 | 7.82 | 8.16
>
> PEMS04
>
> Methods | RMSE values under different Missing rates (25%, 50%, 75%, 90%)
>
> Proposed | 30.67 | 31.41 | 33.38 | 36.27
>
> GinAR | 32.15 | 34.27 | 35.86 | 38.19
>
> MGSFformer | 32.78 | 36.43 | 39.21 | 40.16
>
> S4 | 35.23 | 40.17 | 43.06 | 45.58

---

> ### Author Response · Authors · 2024-11-21
> **Rebuttal by authors**
>
> 3. **W3:**  In the last line of section 2.2, the author mentions, “As a result, the student model’s ability to mine key semantics from sparse observations is effectively enhanced.” To support this statement, the author cites some papers. Among them, the author argues that [3] and [4] show satisfactory results using knowledge distillation even with 50% and 25% observations, respectively. If I am not wrong, [3] and [4] show their efficacy by using fewer complete observations. Spare and fewer observations are two completely different terms and concepts. It feels like the related work here is on fewer observations, however, the conclusion is made for sparse observations. The authors should justify this.
>
> **Reply:**  Thank you for your valuable review comments. Apologies for the lack of rigor in our content. What we intended to convey is that knowledge distillation can help the student model maintain its performance when the information content in the input features is reduced. Sparse observations indicate the presence of missing values in the input features, leading to a reduction in effective information. Similarly, fewer observations imply a decrease in the size of input features, which also reduces the information content. To avoid this issues, we have revised the content in the paper.
>
> 4. **W4:**  One of the main contributions of the paper is stated in Line 123, “we refine that robustness is the main factor contributing to the poor performance of forecasting models in MTSF with sparse observations.” I miss a justification for this contribution, either experimentally or theoretically. The authors argue that due to unfixed missing rates, the existing models give poor robustness. However, this is not a strong justification. The authors need to show this with some kind of experiment. An experiment can be done to show the performance of the baseline models, trained only once, with all unfixed missing rates. The authors can think of other appropriate experiments, too.
>
> **Reply:** Thank you for your valuable review comments. In the revised version, we have added the corresponding experiment in Appendix H (Experiment on Time Series with Unfixed Missing Rates). To be specific, to better simulate the unfixed missing rates in time series data under real-world scenarios, we conduct the following experiments:
> (1) For the test data, we divided the time series into different segments based on time and applied masking to each segment with random missing rates of 25\%, 50\%, 75\%, and 90\%.
> (2) For the training and validation data, we additionally processed the data into four forms with missing rates of 25\%, 50\%, 75\%, and 90\%.
> (3) For Merlin+STID, we trained the models as described in this paper: the unmasked data is used to train the teacher model, while the masked data is used to train the student model. Only the student model is used on the test set.
> (4) For other baselines, we used three training strategies: the first strategy involve training separate models for each missing rate, with the corresponding model selected for forecasting on the test set based on the current data's missing rate. The second strategy uses a single model trained on data with all four missing rates, which is then directly evaluated on the test set. The final strategy is to train a model using only the raw data, which is then directly evaluated on the test set.
> The experimental results show that, when trained only once (either using only complete data or data with multiple missing rates simultaneously), the performance of the baselines is inferior to training a separate model for each missing rate individually.
>
> METR-LA
>
> Methods | MAE | MAPE | RMSE
>
> Proposed | 3.54 | 9.41 | 6.72
>
> STID+GATGPT(Separately)| 3.58| 9.52 | 6.83
>
> STID+GATGPT(Together)| 3.67| 10.12| 6.98
>
> iTransformer+S4(Separately) | 3.76 | 10.78 | 7.32
>
> iTransformer+S4(Together) | 3.88| 11.12| 7.61
>
> STID(Separately)| 3.82 | 10.87| 7.38
>
> STID(Together) | 3.95 | 11.52 | 7.62
>
> STID(Complete) | 4.06 | 12.04| 8.01
>
> GPT4TS(Separately)| 3.89 | 11.23| 7.64
>
> GPT4TS(Together) | 4.02| 12.06 | 7.95
>
> GPT4TS(Complete) | 4.12| 12.34 | 8.19

---

> ### Author Response · Authors · 2024-11-21
> **Rebuttal by authors**
>
> 5. **W5:**  The paper uses the word semantic in multiple places. To give a few examples, semantic alignment, extract semantics, align semantics, semantic mining capabilities, key semantics, disrupt the semantics, incorrect semantics, capture the semantics, error semantics, local semantics, enhance the semantic differences, semantic similarities, high-quality semantics, additional semantic information, etc. I think a reader will benefit if the meaning of the semantics is properly defined in the paper, expanding or limiting its scope to the field of MTSF.
>
> **Reply:**  Thank you for your valuable review comments. In the revised version, we have clarified the definition of "semantics" in the introduction. Specifically, we define the semantics of time series as its global and local information. Missing values disrupt the global information (such as periodicity) of time series and introduce erroneous local information, such as sudden changes (from normal to zero) and abnormal straight lines. These phenomena damage the semantics of the time series, leading to poor performance in existing models. Moreover, we have made every effort to refine the wording of the paper to minimize the excessive use of the term "semantics."
>
> 6. **W6:**  The paper uses knowledge distillation between sparse and complete observations and contrastive learning among sparse observations. It would be interesting to get insights on using complete observations for contrastive learning. The complete observations can also be an input to the student model and subsequently used in contrastive learning.
>
> **Reply:**  Thank you for your valuable review comments. This is a very interesting suggestion. In fact, we also considered this experimental setup. However, we found that knowledge distillation already effectively helps the student model achieve semantic alignment between complete and sparse observations. In the ablation experiment (section 4.4), we removed knowledge distillation and incorporated complete observations into contrastive learning, but the experimental results were not satisfactory.
>
> 7. **W7:**  In step II, section 3.2, the authors mention about spatial-temporal identity embeddings, namely, $S_E, T^D_E,$ and $T^W_E$. However, there is no information about how these embeddings are generated. This is an integral part of the Merlin architecture. Therefore, the authors should discuss them in detail.
>
> **Reply:**  Thank you for your valuable review comments. Since STID is not a model we proposed, we simplified its introduction and provided the corresponding reference. In the revised version, we have included a detailed explanation of these embeddings. The following is an introduction to these embeddings：
>
> Assuming $N_v$ time series and $N_H$ time slots in a day and $N_w$ = 7 days in a week.
>
> $S_E \in R^{N_v*D}$ is the spatial identity embedding.
>
>  $T_{E}^{D} \in R^{N_H*D}$ is the temporal embedding.
>
>  $T_{E}^{W} \in R^{N_w*D}$ is the temporal embedding.
>
> $D$ is the embedding size.
>
> 8. **W8:**  It is mentioned in section 3.3 that the sparse input observations are passed to the fully connected layers to generate embeddings. How are the missing values handled there? Are they just considered as zeros?
>
> **Reply:**  Thank you for your valuable review comments. To ensure a fair comparison with most existing works, we adopted a unified approach to handling missing values (filling them with zero), which is the common practice in the majority of current related works [1] [2].
>
> 9. **W9:** For offline knowledge distillation, especially the $L_{HD}$ loss, what is the rationale behind using mean squared error (MSE) loss? The most common type of loss used in this scenario is the KL-Divergence loss. How does KL-Divergence fare in comparison to MSE in Merlin?
>
> **Reply:** Thank you for your valuable review comments. We referred to existing related works [3]. Specifically, KL divergence focuses on improving the similarity between the distributions of representations, while MSE focuses on minimizing the numerical differences between representations. Considering that time series forecasting is a regression task involving numerical mapping, we believe that using MSE loss is more effective. We have added Appendix E (Compared with Different Loss Functions), where experimental results demonstrate that MSE loss performs better.
>
> METR-LA
>
> Methods | MAE values under different Missing rates (25%, 50%, 75%, 90%)
>
>  Proposed | 3.35 | 3.49 | 3.58 | 3.69
>
>  KL-divergence | 3.42 | 3.55 | 3.68 | 3.79
>
>  PEMS04
>
> Methods | MAE values under different Missing rates (25%, 50%, 75%, 90%)
>
>  Proposed | 18.86 | 19.56 |  21.19 | 22.62
>
>  KL-divergence | 19.45 | 20.38 | 22.53 | 24.07
>
> [1] Learning to Reconstruct Missing Data from Spatiotemporal Graphs with Sparse Observations
>
> [2] Ginar: An end-to-end multivariate time series forecasting model suitable for variable missing
>
> [3] Comparing Kullback-Leibler Divergence and Mean Squared Error Loss in Knowledge Distillation

---

> ### Author Response · Authors · 2024-11-21
> **Rebuttal by authors**
>
> 10. **W10:** The authors mention in lines 351 and 352 that “After testing, it has been found that the performance is optimal when the weight of each loss is set to 1”. However, I could not find any experiment in the rest of the paper which shows the mentioned finding.
>
> **Reply:** Thank you for your valuable review comments. In the revised version, we included the experimental results of the weight of loss functions in Appendix B (Hyperparameter Analysis).
>
> 11. **W11:*** The paper does not report any appropriate information about the statistical significance of the experiments. At the very least, authors should report the standard deviation along with the mean.
>
> **Reply:**  Thank you for your valuable review comments. In the revised version, we have added the standard deviation of the forecasting results for all models in Section 4.2 (Main Results).
>
> 12. **W12:**  The training difference between the Merlin and baseline models is unclear. The paper in lines 422-424 states, “To demonstrate the robustness of the proposed model, we only train it once, using samples with different missing rates. For other baselines, we separately train a model for each missing rate.” Do the authors mean that in Merlin, they incorporate the data with each missing rate in one model? That is, the authors make 3 more versions of the data with 50%, 75%, and 90% missing rates. Merlin is trained on all these 4 datasets combined. Meanwhile, the baseline model is only trained on one of the missing rates datasets individually. If that is the case, then the comparison between Merlin and baseline models is not fair since Merlin sees four times more data compared to the baseline models.
>
> **Reply:** Thank you for your valuable review comments. In fact, when designing the experiments, we considered whether to train a separate model for each missing rate or to train a single model using data with varying missing rates for the baselines. Our experiments revealed that the baselines perform better when a separate model is trained for each missing rate. The experimental results in Appendix H (Experiment on Time Series with Unfixed Missing Rates) also support this conclusion. Therefore, to better demonstrate Merlin's advantages, we adopted this training approach.
>
> METR-LA
>
> Methods | MAE | MAPE | RMSE
>
> Proposed | 3.54 | 9.41 | 6.72
>
> STID+GATGPT(Separately)| 3.58| 9.52 | 6.83
>
> STID+GATGPT(Together)| 3.67| 10.12| 6.98
>
> iTransformer+S4(Separately) | 3.76 | 10.78 | 7.32
>
> iTransformer+S4(Together) | 3.88| 11.12| 7.61
>
> STID(Separately)| 3.82 | 10.87| 7.38
>
> STID(Together) | 3.95 | 11.52 | 7.62
>
> STID(Complete) | 4.06 | 12.04| 8.01
>
> GPT4TS(Separately)| 3.89 | 11.23| 7.64
>
> GPT4TS(Together) | 4.02| 12.06 | 7.95
>
> GPT4TS(Complete) | 4.12| 12.34 | 8.19
>
> 13. **W13:** Section C in the presents hyperparameter analysis with the hyperparameter values used in STID+Merlin. However, there is no discussion of the process of hyperparameter searching. Additionally, the authors should also provide the hyperparameters and their search process for the baseline models. This will help in the reproducibility of the paper.
>
> **Reply:** Thank you for your valuable review comments. Regarding the hyperparameter search process, we determined it based on the hyperparameter settings from the raw papers and source codes, using the grid search method.
>
> 14. **W14:** Figure 4 in the Appendix presents an experiment on determining two hyperparameters of Merlin on the China AQI dataset. These hyperparameter values are then also reported as the best values in Table 5. Does this mean that similar conclusions are seen in other datasets? Please provide this information in the manuscript.
>
> **Reply:** Thank you for your valuable review comments. In the revised version, we included the experimental results of the proposed model on four datasets in Appendix B (Hyperparameter Analysis). The hyperparameter search process was primarily determined based on these experiments.
>
> 15. **W15:** The citation type is not appropriate. The citations should be written within brackets in most of the cases in this article. Therefore, as a reader, it became very difficult to read this paper.
>
> **Reply:**  Thank you for your valuable review comments. We sincerely apologize for overlooking this issue. Initially, we used the officially provided "iclr2025_conference" citation format, which did not include brackets. In the revised version, we have added brackets to each citation.
>
> 16. **W16:** The paper has several grammatical mistakes/typos.
>
> **Reply:** Thank you for your valuable review comments. We carefully checked and corrected all grammatical errors.

---

> ### Author Response · Authors · 2024-11-21
> **Rebuttal by authors**
>
> 17. **W17:**	What does “sudden change” mean in line 73? Do the authors mean the line drop in Figure 1(b)? The author should clarify this because there can be other reasons for sudden change, like concept drift.
>
> **Replay:** Thank you for your valuable review comments. We have added the corresponding explanation in the introduction. Specifically, "sudden changes" refer to anomalies where the data abruptly shifts from normal values to zero values.
>
> 18. **W18:**	In the paper, the authors are inconsistent with their use of capital or small letters for the full form of abbreviations. Please be consistent.
>
> **Replay:** Thank you for your valuable review comments. We have unified the format according to the reviewer's suggestions
>
> 19. **W19:** The paper introduces a notation $N_H$ in line 210. However, the meaning of $N_H$ is not present in the subsequent text. The authors provide its meaning in Appendix A. However, for ease of reading, the author should provide the meaning in line 210. The same practice should be followed for other notations, too.
>
> **Replay:**  Thank you for your valuable review comments. In the revised version, we have included the Notations in the main context. In addition, we have also modified the corresponding content

---

> ### Author Response · Authors · 2024-11-25
>
> Dear Reviewer
>
> Sorry to bother you again.
>
> As the rebuttal phase nears the end, we would like to know if we have addressed your concerns. If you have any remaining concerns, please let us know. We look forward to your valuable reply. Thank you for your efforts in reviewing our paper.
>
> Best regards,
>
> Authors

---

> > ### Comment · Reviewer_odRh · 2024-11-25
> > **Reply to authors rebuttal**
> >
> > I thank the authors for their rebuttal.
> >
> > **Re W1**: The authors mention Figure 1 is “to demonstrate that the performance of existing prediction models consistently declines as the data missing rate increases”. I assume that this is the motivation for this work. However, the performance of MERLIN too declines as the data missing rate increases. I don’t understand the relationship of this experiment (Figure 1) with the story in the Introduction.
> >
> > Moreover, it is clear that the performance decline in MERLIN is not the best compared to baselines (e.g., STID+GATGPT). Specifically, the MAE increases by 2.62%, 3.41%, and 3.02% for STID+GATGPT when the missing rate changes from 25% to 50%, 50% to 75%, and 75% to 90%, respectively. Meanwhile, the MAE increases by 4.18%, 2.58%, and 3.07% for MERLIN. Therefore, MERLIN does not perform well as the missing rate increases.
> >
> > **Re W2**: Thank you for the experiments.
> >
> > **Re W3**: I still stand by my comment that fewer complete observations and sparse observations are different concepts. Moreover, the author mentions, “fewer observations imply a decrease in the size of input features.” How is that? From my understanding, the size/number of input features is fixed, and only the number of instances is reduced. I think implying that since knowledge distillation performs well in fewer complete observations, it will also perform well in sparse observations may not be correct.
> >
> > **Re W4**: Thank you for the experiment. I have a general comment here. It is good that the authors considered multiple training ways of baselines for comparison. However, it is still not a fair comparison. MERLIN is trained with 5 sets of data (complete observation, 25% sparse observations, 50% sparse observations, 75% sparse observations, and 90% sparse observations). Compared to this, the maximum data considered for baselines is 4 sets of data (25% sparse observations, 50% sparse observations, 75% sparse observations, and 90% sparse observations).
> >
> > **Re W5**: Thank you for partially clarifying the meaning of semantics as global and local information. However, please consider the meaning provided with the adjectives used with the word semantic. For example, “enhance the semantic differences,” what does this mean now, and similarly, what about other adjectives? I don’t think this paper is clear.
> >
> > **Re W6**: Thank you for the experiment.
> >
> > **Re W7**: Thank you for additional information. I understand the STID is not the model the author proposed. But, as a reader, it is important to have information regarding the STID in this manuscript, even if it is supplementary.
> >
> > **Re W8, W9, and W10**: Thank you for clarification.
> >
> > **Re W11**: Thank you for this. I would highly suggest including standard deviation (or other statistical significance) for all kinds of comparisons (benchmarking, ablation, and other comparisons).
> >
> > **Re W12**: Please refer to Re W4.
> >
> > **Re W13**: Have the authors mentioned this anywhere in the paper? Additionally, I don’t understand this comment: “Regarding the hyperparameter search process, we determined it based on the hyperparameter settings from the raw papers and source codes, using the grid search method.” Did the authors use grid search themselves or consider the hyperparameters provided in the baseline papers? If the baseline paper's hyperparameter values are considered, then does that mean the training and testing setting, along with the dataset considered and its division and preparation, are exactly the same as the original baseline paper?
> >
> > **Re W14**: Thank you for the clarification.
> >
> > **Re W15*: I still think the citation is not appropriate. I may be wrong, but I don’t think the authors followed the citation format provided by the ICLR 2025. The ICLR 2025 formatting instructions in section 4.1 states
> >
> > > Citations within the text should be based on the natbib package and include the authors’ last names and year (with the “et al.” construct for more than two authors). When the authors or the publication are included in the sentence, the citation should not be in parenthesis using \citet{} (as in “See Hinton et al. (2006) for more information.”). Otherwise, the citation should be in parenthesis using \citep{} (as in “Deep learning shows promise to make progress towards AI (Bengio & LeCun,
> > 2007).”).
> >
> > Please follow the above guidelines. Note that this is a minor comment and does not affect the score provided by me. However, it is important to adhere to ICLR formatting instructions.
> >
> > **Re W16, W17, W18, and W19**: Thank you for all the corrections made to the paper.

---

> > > ### Comment · Reviewer_odRh · 2024-11-25
> > > **Futher comment**
> > >
> > > **Other question**: Is there any specific reason why RMSE is considered in some ablation studies, and MAE is considered in others? Why isn’t there any consistency in metrics for comparison?
> > >
> > > It would have been really helpful if, in the revised manuscript, the authors would have marked all the changes with some color. There have been many changes in the manuscript, with even the section number changed. For example, section 3.2 has now become section 3.3. It was difficult and very time-consuming to understand all the changes made to the paper.
> > >
> > > I would like to maintain my score since the paper is still not clear, the motivation is not clear, the performance (MAE) increase of MERLIN is higher compared to the baseline, the benchmarking is not fair, comparison metrics are inconsistent, etc.

---

> > > > ### Author Response · Authors · 2024-11-25
> > > >
> > > > 10. I would like to maintain my score since the paper is still not clear, the motivation is not clear, the performance (MAE) increase of MERLIN is higher compared to the baseline, the benchmarking is not fair, comparison metrics are inconsistent, etc.
> > > >
> > > > **Reply:** Thank you for your valuable review comments. Below is our overall response to the reviewer's most critical questions:
> > > >
> > > > **(1) The paper is still not clear**
> > > >
> > > > **Reply:** We have further added details about the experimental setup, including the selection and search of hyperparameters. Additionally, we have refined the use and explanation of "semantics," as well as the formatting of citations. We hope these improvements will help the reviewers better understand our paper.
> > > >
> > > > **(2)  the motivation is not clear**
> > > >
> > > > **Reply:** In our paper, we are addressing the classical task of multivariate time series forecasting with sparse observations. First, we observed that the prediction error of forecasting models increases due to the influence of missing values. By examining the data, we found that missing values disrupt the semantics of time series. Moreover, missing rates are often unfixed. By analyzing this task and above two problems, we identify the poor robustness of existing models as the critical issue that needs to be resolved. To tackle this issue, we propose the idea that semantic alignment is the key to improving the robustness of the forecasting model. Based on this idea, we propose the multi-view representation learning. In summary, within the context of an existing task, we believe that the key challenge to address is the poor robustness of existing models. To solve this challenge, we propose the core idea of semantic alignment and design Merlin accordingly. Therefore, we believe that our insights and motivation are complete and well-founded.
> > > >
> > > > **(3) the performance (MAE) increase of MERLIN is higher compared to the baseline**
> > > >
> > > > **Reply:** Whether it is Merlin or other imputation models, their role is to minimize the forecasting error of the forecasting model (STID) as much as possible. In other words, what we need to compare is the performance recovery effect of Merlin and other imputation models on STID under different missing rates. From the curves in the figure 1 (a) and figure 1 (b), it can also be observed that, compared to GATGPT, Merlin achieves a better reduction in the prediction error of STID under all four missing rates. To demonstrate Merlin's performance with data, take Figure 1(a) as an example. When the missing rates are 25%, 50%, 75%, and 90%, Merlin reduces the prediction error of STID by 8.7%, 31.0%, 29.6%, and 32.77%, respectively. In comparison, GATGPT reduces the prediction error of STID by 5.7%, 26.9%, 27.0%, and 30.4%, respectively. This clearly shows that Merlin's improvement over STID is significantly greater than that of GATGPT. Furthermore, based on the results in Section 4.3 (Transferability of Merlin), it can be seen that compared to existing imputation models, Merlin can better reduce the prediction errors of different backbone models.
> > > >
> > > > **(4) the benchmarking is not fair, comparison metrics are inconsistent**
> > > >
> > > > **Reply:**  (a) Both Merlin and the baselines were evaluated using 5 sets of data (complete observations, 25% sparse observations, 50% sparse observations, 75% sparse observations, and 90% sparse observations). However, there are differences in how these datasets are used. For Merlin, the complete observations are used to train the teacher model, while the data with 4 different missing rates are used to train the student model. During the inference phase, the teacher model is not used. For the other baselines, the complete observations are used to train the imputation model or the reconstruction loss of end-to-end models, while the data with 4 different missing rates are used to train the forecasting model. Therefore, the amount of data used is equivalent for both Merlin and the baselines. As a result, our experimental setup is fair.
> > > >
> > > > (b) RMSE and MAE are essentially not significantly different; their primary purpose is to measure the numerical difference between the predicted values and the true values in regression tasks. Although we used different evaluation metrics across different tasks, the metrics used within the same task are consistent (either using a single fixed metric or applying all metrics simultaneously). In this case, there is no issue of inconsistent metrics.

---

> > > > ### Author Response · Authors · 2024-11-25
> > > > **Thank you again for your valuable comments.**
> > > >
> > > > We sincerely appreciate the reviewers for taking the time to provide constructive feedback. We also hope that our responses can address the reviewers' misunderstandings regarding the motivation, Merlin's performance, and the fairness of our experiments.
> > > >
> > > > If you have any further questions, please do not hesitate to let us know. We look forward to your response.

---

> > > ### Author Response · Authors · 2024-11-25
> > >
> > > 7. **Re W15:**  I still think the citation is not appropriate. I may be wrong, but I don’t think the authors followed the citation format provided by the ICLR 2025. The ICLR 2025 formatting instructions in section 4.1 states.
> > >
> > > **Reply:** Thank you for your valuable review comments. We have revised the citations according to the reviewer's suggestions. The current version complies with the required standards.
> > >
> > > 8. **Other question:** Is there any specific reason why RMSE is considered in some ablation studies, and MAE is considered in others? Why isn’t there any consistency in metrics for comparison?
> > >
> > > **Reply:** Thank you for your valuable review comments. RMSE and MAE are essentially not significantly different; their primary purpose is to measure the numerical difference between the predicted values and the true values in regression tasks. Although we used different evaluation metrics across different tasks, the metrics used within the same task are consistent (either using a single fixed metric or applying all metrics simultaneously). In this case, there is no issue of inconsistent metrics.
> > >
> > > 9. It would have been really helpful if, in the revised manuscript, the authors would have marked all the changes with some color. There have been many changes in the manuscript, with even the section number changed. For example, section 3.2 has now become section 3.3. It was difficult and very time-consuming to understand all the changes made to the paper.
> > >
> > > **Reply:** Thank you for your valuable review comments. In our latest revised version, all modifications are highlighted in red.

---

> ### Author Response · Authors · 2024-11-25
> **Rebuttal by authors**
>
> Dear Reviewer odRh:
>
> Thank you for your new suggestions. We have carefully read the new comments you provided. We apologize for any misunderstandings caused by our previous responses. Below are our new responses. We hope these replies will help you gain a comprehensive understanding of our paper.
>
> 1. **Re W1:** The authors mention Figure 1 is “to demonstrate that the performance of existing prediction models consistently declines as the data missing rate increases”. I assume that this is the motivation for this work. However, the performance of MERLIN too declines as the data missing rate increases. I don’t understand the relationship of this experiment (Figure 1) with the story in the Introduction. Moreover, it is clear that the performance decline in MERLIN is not the best compared to baselines (e.g., STID+GATGPT). Specifically, the MAE increases by 2.62%, 3.41%, and 3.02% for STID+GATGPT when the missing rate changes from 25% to 50%, 50% to 75%, and 75% to 90%, respectively. Meanwhile, the MAE increases by 4.18%, 2.58%, and 3.07% for MERLIN. Therefore, MERLIN does not perform well as the missing rate increases.
>
> **Reply:** Thank you for your valuable review comments. There are several misunderstandings that need clarification:
>
> (1) Figure 1 consists of three parts. The purpose of Figures 1(a) and 1(b) is to demonstrate that the performance of existing forecasting models (e.g., STID and GPT4TS) deteriorates in scenarios with missing data, and that the higher the missing rate, the greater the forecasting error (It just shows that missing data affects the forecasting model's ability, not our core motivation.). The purpose of this observation is to prompt us to think: What causes the decline in model performance? How can we restore the performance of the forecasting model?
>
> (2) Based on the observation of the specific data in Figure 1(c), we found that missing values disrupt the semantics of time series. Moreover, missing rates are often unfixed. We believe that addressing these two issues is key to reducing the forecasting error of forecasting models.
>
> (3) Whether it is Merlin or GATGPT, their role is to minimize the forecasting error of the forecasting model (STID) as much as possible. In other words, what we need to compare is the performance recovery effect of Merlin and GATGPT on STID under different missing rates. From the curves in the figure, it can also be observed that, compared to GATGPT, Merlin achieves a better reduction in the prediction error of STID under all four missing rates. To demonstrate Merlin's performance with data, take Figure 1(a) as an example. When the missing rates are 25%, 50%, 75%, and 90%, Merlin reduces the prediction error of STID by 8.7%, 31.0%, 29.6%, and 32.77%, respectively. In comparison, GATGPT reduces the prediction error of STID by 5.7%, 26.9%, 27.0%, and 30.4%, respectively. This clearly shows that Merlin's improvement over STID is significantly greater than that of GATGPT.
>
> 2. **Re W3:** I still stand by my comment that fewer complete observations and sparse observations are different concepts. Moreover, the author mentions, “fewer observations imply a decrease in the size of input features.” How is that? From my understanding, the size/number of input features is fixed, and only the number of instances is reduced. I think implying that since knowledge distillation performs well in fewer complete observations, it will also perform well in sparse observations may not be correct.
>
> **Reply:** Thank you for your valuable review comments. In time series forecasting tasks, historical observations do not represent the number of samples; instead, they represent the input features of the model, denoted as $X \in R^{N_v, N_H, N_c}$ [1] [2].
>
> Here, $N_v$ is the number of sequences.
>
> $N_H$ is the number of time slices.
>
> $N_c$ is the number of features.
>
>  In the revised version of Section 3.1 (Preliminaries), we clarified this point. In other words, "fewer observations" refers to a limitation on the size of $N_H$ in $X\in R^{N_v, N_H, N_c}$, which reduces the size of the model's input features. When the size of the input features decreases, the amount of information available to the model also diminishes.
>
> [1] Pre-training enhanced spatial-temporal graph neural network for multivariate time series forecasting
>
> [2] Ginar: An end-to-end multivariate time series forecasting model suitable for variable missing

---

> ### Author Response · Authors · 2024-11-25
> **Rebuttal by authors**
>
> 3. **Re W4 and Re W12:**  It is good that the authors considered multiple training ways of baselines for comparison. However, it is still not a fair comparison. MERLIN is trained with 5 sets of data (complete observation, 25% sparse observations, 50% sparse observations, 75% sparse observations, and 90% sparse observations). Compared to this, the maximum data considered for baselines is 4 sets of data (25% sparse observations, 50% sparse observations, 75% sparse observations, and 90% sparse observations).
>
> **Reply:** Thank you for your valuable review comments. We sincerely apologize for any misunderstanding regarding the fairness of the experiments. In fact, both Merlin and the baselines were evaluated using 5 sets of data (complete observations, 25% sparse observations, 50% sparse observations, 75% sparse observations, and 90% sparse observations). However, there are differences in how these datasets are used. For Merlin, the complete observations are used to train the teacher model, while the data with 4 different missing rates are used to train the student model. During the inference phase, the teacher model is not used. For the other baselines, the complete observations are used to train the imputation model or the reconstruction loss of end-to-end models, while the data with 4 different missing rates are used to train the forecasting model. Therefore, the amount of data used is equivalent for both Merlin and the baselines. As a result, our experimental setup is fair.
>
> 4. **Re W5:** Thank you for partially clarifying the meaning of semantics as global and local information. However, please consider the meaning provided with the adjectives used with the word semantic. For example, “enhance the semantic differences,” what does this mean now, and similarly, what about other adjectives? I don’t think this paper is clear.
>
> **Reply:** Thank you for your valuable review comments. We have reviewed the entire text and standardized the use of "semantic." Specifically, we removed redundant modifiers, retaining primarily "semantic alignment" and "incorrect semantics." Their explanations are as follows:
>
> (1) **Semantic alignment:** (a) enabling forecasting models to align the semantics between sparse observations and complete observations. (b) enabling forecasting models to align the semantics among sparse observations with different missing rates.
>
> (2) **incorrect semantics:** Missing values disrupt the global information (such as periodicity) of time series and introduce error local information such as sudden changes (From normal to zero) and abnormal straight lines.
>
> 5. **Re W11:**   I would highly suggest including standard deviation (or other statistical significance) for all kinds of comparisons (benchmarking, ablation, and other comparisons).
>
> **Reply:** Thank you for your valuable review comments.  Considering page limitations, we have added standard deviations to most of the experimental results. For details, please refer to Appendix E, Appendix F, Appendix G, Appendix H, Appendix I, and Appendix K.
>
> 6. **Re W13:** Have the authors mentioned this anywhere in the paper? Additionally, I don’t understand this comment: “Regarding the hyperparameter search process, we determined it based on the hyperparameter settings from the raw papers and source codes, using the grid search method.” Did the authors use grid search themselves or consider the hyperparameters provided in the baseline papers? If the baseline paper's hyperparameter values are considered, then does that mean the training and testing setting, along with the dataset considered and its division and preparation, are exactly the same as the original baseline paper?
>
> **Reply:** Thank you for your valuable review comments.   We have explained this in Appendix A.2 (Baselines). Specifically, we referred to the original papers and codes to set the search range for hyperparameters and introduced grid search to obtain the optimal hyperparameters. In addition, the training, validation, and test set splits for these datasets are fixed in most works [1], and we followed the most commonly used proportions for the splits.
>
> [1] Exploring progress in multivariate time series forecasting: Comprehensive benchmarking and heterogeneity analysis

---

### Official Review · Reviewer_PvdY · 2024-11-01

**Soundness:** 3
**Presentation:** 3
**Contribution:** 3
**Rating:** 6
**Confidence:** 4

**Summary:**

The paper introduces Merlin, a robust Multi-View Representation Learning approach for multivariate time series forecasting with unfixed missing rates. This method leverages offline knowledge distillation and multi-view contrastive learning to enhance the robustness of forecasting models, particularly when sparse observations with varying missing rates are present. By aligning the semantic interpretation between sparse and complete observations and across different missing rates, Merlin significantly improves forecasting accuracy and robustness.

**Strengths:**

1.  The introduction of Merlin, which combines offline knowledge distillation with multi-view contrastive learning, is innovative. This framework efficiently addresses the challenge of unfixed missing rates in multivariate time series data, which is a common real-world issue.
2. The paper thoroughly validates the approach using four real-world datasets, demonstrating Merlin's superiority over existing models and imputation methods.
3. The results show that Merlin consistently outperforms traditional forecasting and imputation methods across different datasets and missing rates. This improvement highlights the ability of Merlin to handle the complexity and variability of real-world data effectively.

**Weaknesses:**

1. The experiment only includes 12 steps of forecasting results. It can be convincing to include different step settings such as 6 or 24.
2. I have some concerns about the two-stage settings. To ensure fairness, should we keep the teacher and student models with the same architecture? I'm also interested in whether switching backbones and applying Merlin could improve forecasting with missing data.
3. The paper could benefit from a deeper exploration of scenarios where Merlin might underperform, such as extremely high missing rates or particular types of time series data that do not conform to the assumptions made by the model.

**Questions:**

Minor typo:
1. Line 86 require reconstructing reconstruct

---

> ### Author Response · Authors · 2024-11-21
> **Rebuttal by authors**
>
> Dear Reviewer PvdY:
>
> Thank you very much for your constructive review comments. Below are our responses. We hope to resolve all your concerns.
>
> 1. **W1:** The experiment only includes 12 steps of forecasting results. It can be convincing to include different step settings such as 6 or 24.
>
> **Reply:** Thank you for your valuable review comments. When initially designing the experiments, we aimed to demonstrate that Merlin improves the performance of mainstream architectures (STGNN, Transformer, and MLP). Considering the higher complexity of STGNNs and their use of a fixed prediction horizon of 12 steps, we chose a prediction horizon of 12 steps for consistency. However, we also recognize the importance of evaluating model performance under different prediction horizons. To address this, we have added the experiment Experiment on Different Future Lengths (Appendix G) as a supplement.  The following are partial experimental results. More detailed experimental results can be found in the revised version.
>
> PEMS04
>
> Future Lengths  |  Methods  |  MAE values under different Missing rates (25%, 50%, 75%, 90%)
>
>  6 |  STID+Merlin  |  17.95  |  18.78  |  20.06  |  21.34
>
>  |  STID+GATGPT |  18.35  |  19.16  |  20.94 |  22.45
>
>  |  iTransformer+S4  |  19.54  |  20.63  |  22.06  |  24.04
>
>   | TSMixer+GPT2  |  19.31  | 20.39   | 21.87  |  23.98
>
>  24  |  STID+Merlin  |  20.34  |  21.47   | 22.78   |  24.36
>
>   | STID+GATGPT |  20.89 |  22.05 |  23.34  | 25.19
>
>  |  iTransformer+S4  | 21.97 |  23.86 |  25.88  | 28.04
>
>  |  TSMixer+GPT2 |  21.63 |  23.47  | 25.31 |  27.69
>
>  336 |  STID+Merlin |  24.65 |  26.49 |  27.87 |  29.04
>
>   | STID+GATGPT  | 25.04 |  26.95 |  28.35 |  29.97
>
>  |  iTransformer+S4  | 27.58 |  28.78 |  30.06 |  31.57
>
>  |  TSMixer+GPT2 |  26.94  | 27.32  | 28.84 |  30.75
>
>
> 2. **W2:** I have some concerns about the two-stage settings. To ensure fairness, should we keep the teacher and student models with the same architecture? I'm also interested in whether switching backbones and applying Merlin could improve forecasting with missing data.
>
> **Reply:** Thank you for your valuable review comments. The main reason we use the same structure for the teacher model and the student model is to realize semantic alignment more effectively.   In fact, this is not a two-stage process;   the teacher model is only used during the training phase, while during the testing phase, only the student model is utilized to obtain predictions.   Additionally, in Section 4.3 (Transferability of Merlin), we evaluated Merlin's improvement effect on different backbone models.   The experimental results show that Merlin achieves better performance compared to the interpolation model.
>
> 3. **W3:** The paper could benefit from a deeper exploration of scenarios where Merlin might underperform, such as extremely high missing rates or particular types of time series data that do not conform to the assumptions made by the model.
>
> **Reply:** Thank you for your valuable review comments. In fact, we have considered scenarios where Merlin might perform poorly. Like most imputation methods, we assume that the collected training data is complete and generate missing data by applying a masking operation to train the model. However, in reality, the training data itself may contain missing values, which could degrade the performance of the teacher model. To this end, we conducted an additional experiment by introducing a certain missing rate in the training data and then evaluating Merlin's performance. As shown in Appendix J, even when the training data is subjected to additional missing rates, Merlin still improves the performance of the forecasting model and outperforms imputation models. The following are partial experimental results. More detailed experimental results can be found in the revised version.
>
> The missing rate of the training set is 5%
>
> METR-LA
>
> Backbone | Methods | MAE values under different Missing rates (25%, 50%, 75%, 90%)
>
> STID | +Merlin | 3.39 | 3.54 | 3.62 | 3.71
>
> | +GATGPT | 3.46 | 3.57 | 3.68 | 3.80
>
> | raw | 3.54 | 3.77 | 3.93 | 4.07
>
> DSformer | +Merlin | 3.54| 3.66 | 3.74 | 3.88
>
> | +GATGPT | 3.58 | 3.70 | 3.84| 3.96
>
> | raw | 3.72 | 3.87| 3.95 | 4.11
>
> The missing rate of the training set is 10%
>
> PEMS04
>
> Backbone | Methods | MAE values under different Missing rates (25%, 50%, 75%, 90%)
>
> STID | +Merlin | 19.41 | 20.39 | 21.81 | 23.64
>
> | +GATGPT | 20.05 | 21.43 | 22.88 | 24.31
>
> | raw | 20.67 | 28.36 | 30.11 | 33.65
>
> TSmixer |+Merlin | 20.11 | 22.29 | 23.15 | 24.82
>
> |+GATGPT |20.79 |22.75 | 23.81 | 25.13
>
> | raw | 21.53 | 26.39 | 29.18 | 31.42
>
> 4. **Q1:** Minor typo: Line 86 require reconstructing reconstruct
>
> **Reply:**  Thank you for your valuable review comments. We carefully checked and corrected all grammatical errors.

---

> ### Author Response · Authors · 2024-11-25
>
> Dear Reviewer
>
> Sorry to bother you again.
>
> As the rebuttal phase nears the end, we would like to know if we have addressed your concerns. If you have any remaining concerns, please let us know. We look forward to your valuable reply. Thank you for your efforts in reviewing our paper.
>
> Best regards,
>
> Authors

---

> > ### Comment · Reviewer_PvdY · 2024-11-25
> > **Thank you for the reply**
> >
> > Thanks authors for the response. All my concerns are addressed and I will keep my rating.

---

> > > ### Author Response · Authors · 2024-11-25
> > >
> > > Thank you for your valuable comments, which are very helpful to improve the quality of our paper

---

### Official Review · Reviewer_VJBz · 2024-11-02

**Soundness:** 3
**Presentation:** 2
**Contribution:** 3
**Rating:** 6
**Confidence:** 4

**Summary:**

This paper addresses time series forecasting with missing data by proposing a knowledge distillation approach to align hidden representations and forecasting results derived from both complete and incomplete data. To enhance robustness across varying missing rates, a contrastive loss is introduced, constructed using positive pairs with differing missing rates and negative pairs at different time point. Empirical experiments demonstrate the effectiveness of the proposed method.

**Strengths:**

- **Clear logic of the method** The paper has a clear presentation of the logic behind the proposed method.
- **Intuitive method**: The proposed approach is intuitive and easy to follow.
- **Clear ablation study**: The ablation study is thorough, clearly demonstrating the effectiveness of each component of the proposed method.

**Weaknesses:**

- **Vague description of key notations and definitions**: Two crucial components of the loss function are vaguely defined, making it challenging for others to reimplement the method based solely on the method description.

  - Knowledge distillation: The teacher model is trained on complete observations, while the student model is trained on missing observations. However, the authors should clarify what constitutes a "complete observation" versus a "missing observation." Given that the focus is on time series with missing data, it is unclear if or how truly complete observations are available.

  - Contrastive learning: The definitions of positive and negative pairs are too vague and should be explicitly defined using the established notations. This would enhance clarity and replicability.

- **Vague description of method details** The missing rates for the input in student model is not clearly described and hence is diffiult to see how robust the proposed method extrapolation to other unfixed missing rates.

**Questions:**

- **Method**
  - It appears that the loss functions may be defined using sliding windows on the time series. If so, it would be helpful to discuss how the size of the sliding window might impact the results.
  - The state space model [1] can effectively handle missing data. Why not consider using this model directly for forecasting in this setting?

- **Experiment**
  - Several related works mentioned in the literature review are not included in the experimental comparisons. Is there a reason for this omission?
  - How significant is the improvement of the proposed method compared to STID+GATGPT?
  - The experiments assume a "missing completely at random" pattern, but other patterns—such as "missing at random" or "missing not at random"—could yield different results. For example, what would happen if only observations above a certain threshold were missing?
  - The selected missing rates in the experiments are relatively high. How does the proposed method perform with lower missing rates? Additionally, perform forecasting when the majority of the data seems unreasonable in practice, it would be insightful to discuss how realistic this setting is.

- **Writing improvement**
  - Please refer to comments in the "Weaknesses" section for clarity improvements.
  - The LaTeX formatting in the equations could be improved for consistency and readability; for instance, in (8), use \log for log, and apply \left(...\right) for brackets within the log term.
  - Key definitions and notations should ideally be included in the main text rather than the appendix. If space is constrained, consider compressing the related work section.


References

[1] Efficiently Modeling Long Sequences with Structured State Spaces

---

> ### Author Response · Authors · 2024-11-21
> **Rebuttal by authors**
>
> Dear Reviewer VJBz:
>
> Thank you very much for your constructive review comments. Below are our responses. We hope to resolve all your concerns.
>
> 1.**W1**: Knowledge distillation: The teacher model is trained on complete observations, while the student model is trained on missing observations. However, the authors should clarify what constitutes a "complete observation" versus a "missing observation." Given that the focus is on time series with missing data, it is unclear if or how truly complete observations are available.
>
> **Reply:** Thank you for your valuable review comments. Like most imputation methods, we assume that the collected training data is complete and generate missing data by applying a masking operation to train the model. In this case, we assume that the input to the teacher model during the training phase is complete, without any missing values, just as the imputation model assumes that the model's labels are complete and without missing values. However, in reality, the training data itself may contain missing values, which could degrade the performance of the teacher model. To this end, we conducted an additional experiment by introducing a certain missing rate in the training data and then evaluating Merlin's performance. As shown in Appendix J, even when the training data is subjected to additional missing rates, Merlin still improves the performance of the forecasting model and outperforms imputation models.
>
> **The missing rate of the training set is 5%**
>
> METR-LA
>
> Backbone | Methods | MAE values under different Missing rates (25%, 50%, 75%, 90%)
>
> STID  |  +Merlin  | 3.39 |  3.54 | 3.62 | 3.71
>
> | +GATGPT | 3.46 | 3.57 | 3.68 | 3.80
>
> | raw   |   3.54 | 3.77 | 3.93 | 4.07
>
> DSformer | +Merlin | 3.54|  3.66 | 3.74 | 3.88
>
>  | +GATGPT | 3.58 | 3.70 | 3.84|  3.96
>
> |  raw | 3.72 | 3.87|  3.95 | 4.11
>
> **The missing rate of the training set is 10%**
>
>  PEMS04
>
> Backbone | Methods | MAE values under different Missing rates (25%, 50%, 75%, 90%)
>
>
> STID  | +Merlin  | 19.41  | 20.39  | 21.81  | 23.64
>
>   | +GATGPT  | 20.05  | 21.43  | 22.88  | 24.31
>
>   | raw  | 20.67  | 28.36  | 30.11  | 33.65
>
>  TSmixer |+Merlin | 20.11 | 22.29 | 23.15 | 24.82
>
>  |+GATGPT |20.79 |22.75 | 23.81 | 25.13
>
> | raw | 21.53 | 26.39 | 29.18 | 31.42
>
> 2.**W2**: Contrastive learning: The definitions of positive and negative pairs are too vague and should be explicitly defined using the established notations. This would enhance clarity and replicability.
>
> **Reply:** Thank you for your valuable review comments. To further enhance the robustness of the student model and realize the semantic alignment of data with different missing rates, we proposes a multi-view contrastive learning method. We use historical observations with different missing rates at the same time point as positive data pairs, and use historical observations at different time point (other samples within a batch) as negative data pairs. Specifically, for representations$ Z_{E,1} $ to $ Z_{E,m} $ encoded from historical observations with different missing rates, we employ a pairwise contrastive learning approach to achieve multi-view contrastive learning. For example, $ Z_{E,1} $ and $ Z_{E,2}$ represent two sets of samples with different missing rates, each containing $ N_s $ samples. In $Z_{E,1}$ and $ Z_{E,2} $, the corresponding two samples form a positive data pair, while all other samples are treated as negative data pairs.

---

> > ### Comment · Reviewer_VJBz · 2024-11-24
> >
> > Thank you for your response. In your experiment, you also comparied with GPT2, MAE, SPIN methods. Could you please provide the experiment results for these methods in the new setting? Also, could you provide more specific details on how the missing values in the training set were handled in your new experiment?

---

> ### Author Response · Authors · 2024-11-21
> **Rebuttal by authors**
>
> 3. **W3**: The missing rates for the input in student model is not clearly described and hence is diffiult to see how robust the proposed method extrapolation to other unfixed missing rates.
>
> **Reply:** Thank you for your valuable review comments. In main results, to prove the robustness of our model, we train it once, using samples with multiple missing rates. In other words, the student model is trained simultaneously using data with missing rates of 25\%, 50\%, 75\%, and 90\%. For other baselines, we train them using two ways and report the best: one is training a separate model for each missing rate, and the other is training a single model using samples with multiple missing rates.
>
> Besides, to better simulate the unfixed missing rates in time series data under real-world scenarios, we conduct new experiments in Appendix H (Experiment on Time Series with Unfixed Missing Rates):
> (1) For the test data, we divided the time series into different segments based on time and applied masking to each segment with random missing rates of 25\%, 50\%, 75\%, and 90\%.
> (2) For the training and validation data, we additionally processed the data into four forms with missing rates of 25\%, 50\%, 75\%, and 90\%.
> (3) For Merlin+STID, we trained the models as described in this paper: the unmasked data is used to train the teacher model, while the masked data is used to train the student model. Only the student model is used on the test set.
> (4) For other baselines, we used three training strategies: the first strategy involve training separate models for each missing rate, with the corresponding model selected for forecasting on the test set based on the current data's missing rate. The second strategy uses a single model trained on data with all four missing rates, which is then directly evaluated on the test set. The final strategy is to train a model using only the raw data, which is then directly evaluated on the test set.
>
>  The following are partial experimental results. More detailed experimental results can be found in the revised version of Appendix H.
>
>  METR-LA
>
> Methods |  MAE |  MAPE |  RMSE
>
>  Proposed |  3.54 | 9.41 | 6.72
>
>  STID+GATGPT(Separately)|  3.58|  9.52 | 6.83
>
>  STID+GATGPT(Together)|  3.67|  10.12|  6.98
>
>  iTransformer+S4(Separately) | 3.76 | 10.78 | 7.32
>
>  iTransformer+S4(Together) | 3.88|  11.12|  7.61
>
>  STID(Separately)|  3.82 | 10.87|  7.38
>
>  STID(Together) | 3.95 | 11.52 | 7.62
>
>  STID(Complete) | 4.06 | 12.04|  8.01
>
>  GPT4TS(Separately)|  3.89 | 11.23|  7.64
>
>  GPT4TS(Together) | 4.02|  12.06 | 7.95
>
>  GPT4TS(Complete) | 4.12|  12.34 | 8.19
>
> 4. **Q1**: It appears that the loss functions may be defined using sliding windows on the time series. If so, it would be helpful to discuss how the size of the sliding window might impact the results.
>
> **Reply:** Thank you for your valuable review comments. We added the influence of sliding window on experimental results in Appendix B (Hyperparameter analysis). The experimental results show that the size of sliding window should be properly balanced to ensure the performance of the model. When the size of the sliding window is too large, it will result in overfitting. When the size of the sliding window is too small, the input information of the model will be insufficient.
>
> 5. **Q2**: The state space model can effectively handle missing data. Why not consider using this model directly for forecasting in this setting?
>
> **Reply:** Thank you for your valuable review comments. Our primary goal is to improve the robustness of forecasting models by proposing Merlin. Therefore, when designing the experiments, we focused on comparing imputation models, such as Merlin and other imputation models, in terms of their ability to enhance the performance of different forecasting models. However, we believe the reviewers' comments are very important. To further validate the model's performance, we compared Merlin with several existing end-to-end models that can handle missing data. The following are partial experimental results. More detailed experimental results can be found in the revised version of Appendix K.
>
>  METR-LA
>
> Methods |  RMSE values under different Missing rates (25%, 50%, 75%, 90%)
>
>  Proposed |  6.58 |   6.65 |   6.81 |   7.06
>
>  GinAR |   6.72 |   6.91 |   7.38|   7.67
>
>  MGSFformer |   6.78 |   6.98 |  7.45 |   7.84
>
>  S4 |   7.13 |   7.54 |   7.82 |   8.16
>
>  PEMS04
>
> Methods |  RMSE values under different Missing rates (25%, 50%, 75%, 90%)
>
>  Proposed  |  30.67 |   31.41 |   33.38 |   36.27
>
>  GinAR |   32.15 |   34.27 |   35.86 |   38.19
>
>  MGSFformer |   32.78 |   36.43 |   39.21 |   40.16
>
>  S4 |   35.23 |   40.17  |  43.06 |   45.58

---

> ### Author Response · Authors · 2024-11-21
> **Rebuttal by authors**
>
> 6. **Q3:** Several related works mentioned in the literature review are not included in the experimental comparisons. Is there a reason for this omission?
>
> **Reply:** Thank you for your valuable review comments. When selecting baselines, we considered both the practical performance of the models and their level of open-sourcing. Some related works were not used because they often had issues such as excessive complexity, lack of open-source code, or poor performance on this task.
>
> 7. **Q4:**  How significant is the improvement of the proposed method compared to STID+GATGPT?
>
> **Reply:** Thank you for your valuable review comments. Compared to STID+GATGPT, the proposed model achieves an overall improvement of approximately 1.5\% to 5\%. Note that STID+GATGPT trains a separate model for each missing rate. If only a single STID+GATGPT model is trained, the proposed model achieves an overall improvement of approximately 4\% to 11\%.
>
> 8. **Q5:** The experiments assume a "missing completely at random" pattern, but other patterns—such as "missing at random" or "missing not at random"—could yield different results. For example, what would happen if only observations above a certain threshold were missing?
>
> **Reply:** Thank you for your valuable review comments. In Appendix I (Experiment on Other Data Missing Scenarios), we add two additional missing scenarios:
> (1)Data points whose mask exceeds a certain threshold: we treat $m$\% of the larger values and $m$\% of the smaller values in the dataset as missing values. In other words, only the data points in the middle $(1-2m)$\% of the value range are kept.
> (2) Random point missing based on geometric distribution: different from uniformly random missing situations, in this distribution, missing values appear in segments. In other words, multivariate time series exhibit a certain amount of consecutive missing values over different time periods.
> The following are partial experimental results. More detailed experimental results can be found in the revised version. The experimental results show that the proposed model can still achieve the best experimental results.
>
> **Data points whose mask exceeds a certain threshold**
>
> METR-LA
>
> Methods   |  MAE values under different  Missing rates  (10%, 20%, 30%, 40%)
>
>  Proposed  |   3.31  |   3.37   |  3.42   |  3.51
>
>  STID+GATGPT  |  3.39 |   3.44  |  3.49 |   3.56
>
>  FourierGNN+SPIN   |  3.45   |  3.51   |  3.58   |  3.65
>
>  DSformer+GATGPT  |   3.49  |   3.54   |  3.62   |  3.70
>
>  TSMixer+GPT2  |   3.43  |   3.49   |  3.55  |   3.63
>
> **Random point missing based on geometric distribution**
>
> PEMS04
>
> Methods   |   MAE values under different  Missing rates (25%, 50%, 75%, 90%)
>
>  Proposed  |  19.03  |  19.87  |  21.45  |  22.87
>
>  STID+GATGPT |   19.63  |  21.06  |  22.57  |  24.15
>
>  FourierGNN+SPIN  |  20.85 |   22.35  |  23.76  |  24.55
>
>  DSformer+GATGPT |   21.03  |  22.89  |  24.32 |   24.78
>
>  TSMixer+GPT2 |   21.16  |  23.07  |  24.58  |  25.19
>
>
> 9. **Q6:** The selected missing rates in the experiments are relatively high. How does the proposed method perform with lower missing rates? Additionally, perform forecasting when the majority of the data seems unreasonable in practice, it would be insightful to discuss how realistic this setting is.
>
> **Reply:** Thank you for your valuable review comments. In fact, during the process of designing our experiments, we referred to a considerable amount of related works (especially imputation models) [1] [2]. To demonstrate model performance, most studies evaluate their models under high missing rates. Furthermore, during severe natural disasters, the missing rate can indeed reach very high levels. However, we also agree with the reviewers' comments that evaluating the model's performance under less extreme missing rates is equally important. Therefore, in the revised version, we have added experiments with a missing rate of 25%. Fortunately, our model remains optimal even with a missing rate of 25%. The following are partial experimental results. More detailed experimental results can be found in the revised version.
>
> PEMS04
>
> Method | MAE | MAPE | RMSE
>
> STID+Merlin | 18.86 | 12.97 | 30.67
>
> STID+GATGPT | 19.48 | 13.15 | 31.28
>
> DSformer+GATGPT | 20.38 | 13.87 | 32.35
>
> FourierGNN+SPIN | 20.06 | 13.75 | 32.13
>
> TSMixer+GPT2 | 23.42 | 13.94 | 32.47
>
> METR-LA
>
> Method | MAE | MAPE | RMSE
>
> STID+Merlin | 3.35 | 9.21 | 6.58
>
> STID+GATGPT | 3.43 | 9.25 | 6.64
>
> DSformer+GATGPT | 3.52 | 9.37 | 6.73
>
> FourierGNN+SPIN | 3.50 | 9.32 | 6.71
>
> TSMixer+GPT2 | 3.48 | 9.29 | 6.69
>
> [1] Learning to Reconstruct Missing Data from Spatiotemporal Graphs with Sparse Observations
>
> [2] Ginar: An end-to-end multivariate time series forecasting model suitable for variable missing

---

> ### Author Response · Authors · 2024-11-21
> **Rebuttal by authors**
>
> 10. Please refer to comments in the "Weaknesses" section for clarity improvements.
>
> 11. The LaTeX formatting in the equations could be improved for consistency and readability; for instance, in (8), use \log for log, and apply \left(...\right) for brackets within the log term.
>
> 12. Key definitions and notations should ideally be included in the main text rather than the appendix. If space is constrained, consider compressing the related work section.
>
> **Reply:** Thank you for your valuable review comments. In the revised version, we have included the Notations in the main context. In addition, we uniformly correct the grammatical errors and formatting of the papers

---

> ### Author Response · Authors · 2024-11-24
> **Rebuttal by authors**
>
> Dear Reviewer VJBz:
>
> Thank you for your new suggestions.  Below are our responses. We hope to resolve your concerns.
>
> 1. **Q1:** In your experiment, you also comparied with GPT2, MAE, SPIN methods. Could you please provide the experiment results for these methods in the new setting?
>
> **Reply:** Thank you for your valuable review comments. Following the reviewers' suggestions, we have added three new baselines (GPT2, MAE, and SPIN) to Appendix I (Experiment on Other Data Missing Scenarios) and Appendix J (Experiments When the Performance of the Teacher Model is Degraded). Partial experimental results are shown below, and the complete results can be found in the revised Appendix I and Appendix J.
>
> **Data points whose mask exceeds a certain threshold**
>
> METR-LA
>
> Methods | MAE values under different Missing rates (10%, 20%, 30%, 40%)
>
> Proposed | 3.31 | 3.37 | 3.42 | 3.51
>
> STID+GATGPT | 3.39 | 3.44 | 3.49 | 3.56
>
> STID+MAE |  3.46 |  3.53 |  3.57  | 3.63
>
> STID+GPT2 |  3.44 |  3.50 |  3.55 |  3.61
>
> STID+SPIN |  3.40 |  3.46  | 3.52 |  3.58
>
> **Random point missing based on geometric distribution**
>
> PEMS04
>
> Methods | MAE values under different Missing rates (25%, 50%, 75%, 90%)
>
> Proposed | 19.03 | 19.87 | 21.45 | 22.87
>
> STID+GATGPT | 19.63 | 21.06 | 22.57 | 24.15
>
> STID+MAE | 20.16 | 21.44|  22.63|  24.14
>
> STID+GPT2|  20.04|  21.67|  22.87 | 24.35
>
> STID+SPIN|  19.75|  21.13|  23.14 | 24.49
>
> **The missing rate of the training set is 5%**
>
> METR-LA
>
> Backbone | Methods | MAE values under different Missing rates (25%, 50%, 75%, 90%)
>
> STID | +Merlin | 3.39 | 3.54 | 3.62 | 3.71
>
> | +GATGPT | 3.46 | 3.57 | 3.68 | 3.80
>
> | +GPT2|  3.51|  3.64|  3.72|  3.81
>
> | +MAE|  3.52|  3.66 | 3.74 | 3.82
>
> | +SPIN|  3.47|  3.62 | 3.75 | 3.84
>
> | raw | 3.54 | 3.77 | 3.93 | 4.07
>
> DSformer | +Merlin | 3.54| 3.66 | 3.74 | 3.88
>
> | +GATGPT | 3.58 | 3.70 | 3.84| 3.96
>
> | +GPT2|  3.61| 3.75|  3.88 | 4.01
>
> | +MAE|  3.63|  3.77 | 3.90 | 3.99
>
> | +SPIN|  3.59 | 3.72|  3.87|  4.03
>
> | raw | 3.72 | 3.87| 3.95 | 4.11
>
> **The missing rate of the training set is 10%**
>
> PEMS04
>
> Backbone | Methods | MAE values under different Missing rates (25%, 50%, 75%, 90%)
>
> STID | +Merlin | 19.41 | 20.39 | 21.81 | 23.64
>
> | +GATGPT | 20.05 | 21.43 | 22.88 | 24.31
>
> | +GPT2  | 20.33 | 22.08 | 23.14|  24.48
>
> | +MAE | 20.51 |  21.84 |  22.95 |  24.37
>
> | +SPIN |  20.14 |  21.71 |  23.31 |  24.64
>
> | raw | 20.67 | 28.36 | 30.11 | 33.65
>
> TSmixer |+Merlin | 20.11 | 22.29 | 23.15 | 24.82
>
> |+GATGPT |20.79 |22.75 | 23.81 | 25.13
>
> | +GPT2  | 21.03 |  23.12 |  24.73 |  26.16
>
> | +MAE  | 21.25  | 23.34 24.82 |  26.03
>
> | +SPIN  | 20.87  | 22.95 |  24.94  | 26.29
>
> | raw | 21.53 | 26.39 | 29.18 | 31.42
>
> 2. **Q2:**  Could you provide more specific details on how the missing values in the training set were handled in your new experiment?
>
> **Reply:** Thank you for your valuable review comments. Considering the characteristics of the imputation models [1] and the proposed model, we process the training data as follows:
>
> (1) Missing values are uniformly handled as zeros, following the approach commonly adopted in existing works [2] [3].
>
> (2) For the experiments in Appendix I (Experiment on Other Data Missing Scenarios), the original data is processed to simulate missing rates of 10%, 20%, 30%, and 40% (Data points whose mask exceeds a certain threshold).  For another missing scene (Random point missing based on geometric distribution), the original data is processed to simulate missing rates of 25%, 50%, 75%, and 90%. The proposed model and baselines are trained on the original data as well as the datasets with these four missing rates.
>
> (3) For the experiments in Appendix J (Experiments When the Performance of the Teacher Model is Degraded), the original data is first processed to simulate missing rates of 5% and 10%. Subsequently, the data with these missing rates is further processed to simulate missing rates of 25%, 50%, 75%, and 90%. The proposed model and baselines are trained separately on datasets with 5% and 10% missing rates, as well as the datasets with subsequent missing rates of 25%, 50%, 75%, and 90%.
>
> [1] Learning to Reconstruct Missing Data from Spatiotemporal Graphs with Sparse Observations
>
> [2] Ginar: An end-to-end multivariate time series forecasting model suitable for variable missing
>
> [3] Simmtm: A simple pre-training framework for masked time-series modeling
>
> We sincerely appreciate the reviewers' valuable feedback. We hope that our new responses address the reviewers' concerns. If there are any further questions or suggestions, please do not hesitate to let us know.

---

> > ### Comment · Reviewer_VJBz · 2024-11-25
> >
> > Thanks for the response. Most of my concerns are addressed, I have adjusted my rating.

---

> > > ### Author Response · Authors · 2024-11-25
> > > **Thank you for your valuable comments**
> > >
> > > Thank you for your valuable comments, which are very helpful to improve the quality of our paper

---

### Official Review · Reviewer_wG4z · 2024-11-11

**Soundness:** 3
**Presentation:** 3
**Contribution:** 2
**Rating:** 3
**Confidence:** 4

**Summary:**

This paper presents multi-view representation learning based on offline knowledge distillation and multi-view contrastive learning.  The key idea is to achieve semantic alignment between sparse observations with different missing rates and complete observations and enhance their robustness. The experimental results demonstrate the effectiveness of the proposed method.

**Strengths:**

* The paper is organized.
* Time series forecasting with missing values is an important problem to study and the proposed method is technically sound.
* The experiment results showed the effectiveness of the proposed method.

**Weaknesses:**

* The technical novelty is limited as this paper simply combines knowledge distillation and multi-view-based contrastive learning based on STID framework (which is also existing). From the application perspective, the proposed combination is somewhat reasonable.
* The motivation for this study is not clear (see questions below).
* Certain details about the proposed method are not provided  (see below questions).

**Questions:**

1. It seems the authors focus on studying high missing rates (more than 50%) for time series forecasting. However, it is not clear whether this is practical in real-world applications. Can you give a few examples of this setting? What’s the typical reason for such high missing rates?
2. It seems like the paper focuses on uniformly random missing situations. I wonder whether the proposed method also works for other types of missing, e.g., missing not at random /structural missing?
3. How do you set up the model training? Did you put 50%, 75%, and 90% missing rates in the student model to train one unified student model for different missing rates?
4. In Eq (4), I wonder whether the current weights are optimal ones

Minor suggestions:
1. Notations should be provided in the main context to make it self-contained.
2. Standard deviations are not provided in the experiment results.

---

> ### Author Response · Authors · 2024-11-21
> **Rebuttal by authors**
>
> Dear Reviewer wG4z:
>
> Thank you very much for your constructive review comments. Below are our responses. We hope to resolve all your concerns.
>
> 1. **W1**: The technical novelty is limited as this paper simply combines knowledge distillation and multi-view-based contrastive learning based on STID framework (which is also existing). From the application perspective, the proposed combination is somewhat reasonable.
>
> **Reply:**  Thank you for your valuable review comments. We sincerely apologize for causing the reviewer to mistakenly believe that we are simply combining knowledge distillation, contrastive learning, and STID. Currently, many related works [1] [2] [3] [4] focus on improving knowledge distillation and contrastive learning, treating them as conceptual frameworks rather than standalone methods. The core technical contribution of knowledge distillation lies in designing the process of knowledge transfer, while that of contrastive learning centers on the design of positive and negative data pairs.
> The primary goal of this paper is to enhance the robustness of existing models when mining data with unfixed missing rates. We identify semantic alignment between complete observations and sparse observations with different missing rates as the key to achieving this goal. To this end, we integrate the ideas of knowledge distillation and contrastive learning.
> In terms of knowledge distillation, we propose using both representations and forecasting results as the transferred knowledge to achieve semantic alignment between sparse and complete observations. For contrastive learning, we design positive and negative data pairs from samples with different missing rates and propose a multi-view contrastive loss to strengthen semantic alignment among sparse observations with different missing rates. Through the above approach, we designed the multi-view representation learning (Merlin), where each component is designed with a core purpose in mind.
>
>
> 2. **W2**: The motivation for this study is not clear (see questions below).
>
> **Reply:**  Thank you for your valuable review comments. We sincerely apologize for not clearly conveying our motivation to the reviewers. In fact, we are addressing the classical task of multivariate time series forecasting with sparse observations. Within this task, we identify the poor robustness of existing models as the critical issue that needs to be resolved. To tackle this issue, we propose the idea that semantic alignment is the key to improving the robustness of the forecasting model. Based on this idea, we propose the multi-view representation learning. In summary, within the context of an existing task, we believe that the key challenge to address is the poor robustness of existing models. To solve this challenge, we propose the core idea of semantic alignment and design Merlin accordingly. Therefore, we believe that our insights and motivation are complete and well-founded.
>
> 3. **W3**: Certain details about the proposed method are not provided (see below questions).
>
>  **Reply:** Thank you for your valuable review comments. We have modified the corresponding content of our paper to help reviewers better understand the key details and technical processes of the model.
>
>
> [1] Online Knowledge Distillation for Financial Timeseries Forecasting
>
> [2] How many observations are enough? knowledge distillation for trajectory forecasting
>
> [3] CoST: Contrastive Learning of Disentangled Seasonal-Trend Representations for Time Series Forecasting
>
> [4] Time series contrastive learning with information-aware augmentations

---

> ### Author Response · Authors · 2024-11-21
> **Rebuttal by authors**
>
> 4. **Q1**: It seems the authors focus on studying high missing rates (more than 50%) for time series forecasting. However, it is not clear whether this is practical in real-world applications. Can you give a few examples of this setting? What’s the typical reason for such high missing rates?
>
> **Reply:** Thank you for your valuable review comments. In fact, during the process of designing our experiments, we referred to a considerable amount of related works (especially imputation models) [1] [2]. To demonstrate model performance, most studies evaluate their models under high missing rates. Furthermore, during severe natural disasters, the missing rate can indeed reach very high levels. However, we also agree with the reviewers' comments that evaluating the model's performance under less extreme missing rates is equally important. Therefore, in the revised version, we have added experiments with the missing rate of 25%. Fortunately, our model remains optimal even with the missing rate of 25%. The following are partial experimental results. More detailed experimental results can be found in the revised version.
>
> PEMS04
>
> Method | MAE  | MAPE | RMSE
>
> STID+Merlin  |  18.86  |  12.97  |  30.67
>
> STID+GATGPT  |  19.48 |   13.15 |  31.28
>
> DSformer+GATGPT  |  20.38  |  13.87  |  32.35
>
>  FourierGNN+SPIN |  20.06 | 13.75 | 32.13
>
> TSMixer+GPT2  |  23.42  |  13.94  | 32.47
>
> METR-LA
>
> Method | MAE  | MAPE | RMSE
>
> STID+Merlin  |  3.35  |   9.21 |  6.58
>
> STID+GATGPT  | 3.43 |  9.25 | 6.64
>
> DSformer+GATGPT  |  3.52  |  9.37  | 6.73
>
>  FourierGNN+SPIN |  3.50 | 9.32 | 6.71
>
> TSMixer+GPT2  |  3.48  |  9.29  | 6.69
>
> 5. **Q2**: It seems like the paper focuses on uniformly random missing situations. I wonder whether the proposed method also works for other types of missing, e.g., missing not at random /structural missing?
>
> **Reply:** Thank you for your valuable review comments. In Appendix I (Experiment on Other Data Missing Scenarios), we add two additional missing scenarios:
> (1) **Data points whose mask exceeds a certain threshold**: we treat $m$\% of the larger values and $m$\% of the smaller values in the dataset as missing values. In other words, only the data points in the middle $(1-2m)$\% of the value range are kept.
> (2) **Random point missing based on geometric distribution**: different from uniformly random missing situations, in this distribution, missing values appear in segments. In other words, multivariate time series exhibit a certain amount of consecutive missing values over different time periods.
> The following are partial experimental results. More detailed experimental results can be found in the revised version of Appendix I. The experimental results show that the proposed model can still achieve the best experimental results.
>
> **Data points whose mask exceeds a certain threshold**
>
> METR-LA
>
> Methods   |  MAE values under different  Missing rates  (10%, 20%, 30%, 40%)
>
>  Proposed  |   3.31  |   3.37   |  3.42   |  3.51
>
>  STID+GATGPT  |  3.39 |   3.44  |  3.49 |   3.56
>
>  FourierGNN+SPIN   |  3.45   |  3.51   |  3.58   |  3.65
>
>  DSformer+GATGPT  |   3.49  |   3.54   |  3.62   |  3.70
>
>  TSMixer+GPT2  |   3.43  |   3.49   |  3.55  |   3.63
>
> **Random point missing based on geometric distribution**
>
> Methods   |   MAE values under different  Missing rates (25%, 50%, 75%, 90%)
>
>  Proposed  |  19.03  |  19.87  |  21.45  |  22.87
>
>  STID+GATGPT |   19.63  |  21.06  |  22.57  |  24.15
>
>  FourierGNN+SPIN  |  20.85 |   22.35  |  23.76  |  24.55
>
>  DSformer+GATGPT |   21.03  |  22.89  |  24.32 |   24.78
>
>  TSMixer+GPT2 |   21.16  |  23.07  |  24.58  |  25.19
>
> 6. **Q3**: How do you set up the model training? Did you put 50%, 75%, and 90% missing rates in the student model to train one unified student model for different missing rates?
>
> **Reply:** Thank you for your valuable review comments. To prove the robustness of our model, we train it once, using samples with multiple missing rates. In other words, the student model is trained simultaneously using data with missing rates of 25\%, 50\%, 75\%, and 90\%. For other baselines, we train them using two ways and report the best: one is training a separate model for each missing rate, and the other is training a single model using samples with multiple missing rates.
>
> 7. **Q4**: In Eq (4), I wonder whether the current weights are optimal ones.
>
> **Reply:** We consistently use the grid search method to ensure that the model's weights and hyperparameter settings are optimal. For detailed experimental results on hyperparameters, please refer to Appendix B.
>
> [1] Learning to Reconstruct Missing Data from Spatiotemporal Graphs with Sparse Observations
>
> [2] Ginar: An end-to-end multivariate time series forecasting model suitable for variable missing

---

> ### Author Response · Authors · 2024-11-21
> **Rebuttal by authors**
>
> 8.**Q5**:Notations should be provided in the main context to make it self-contained.
>
> **Reply:**  Thank you for your valuable review comments. In the revised version, we have included the Notations in the main context.
>
> 9.**Q6**: Standard deviations are not provided in the experiment results.
>
> **Reply:** Thank you for your valuable review comments. In the revised version, we have added the standard deviation of the forecasting results for all models in Section 4.2 (Main Results).

---

> ### Author Response · Authors · 2024-11-25
>
> Dear Reviewer
>
> Sorry to bother you again.
>
> As the rebuttal phase nears the end, we would like to know if we have addressed your concerns. If you have any remaining concerns, please let us know. We look forward to your valuable reply. Thank you for your efforts in reviewing our paper.
>
> Best regards,
>
> Authors

---

### Author Response · Authors · 2024-11-21
**Rebuttal for all Reviewers**

Dear Reviewers,

We sincerely appreciate the time and effort you have taken to thoroughly read our paper and provide constructive review comments. We have now revised the paper according to all the reviewers' comments and uploaded the updated version. We hope that our responses address all of your concerns. Below is a summary of the modifications made to the paper:

1. We have added experiments with a missing rate of 25% across all experiments to verify Merlin's performance under low missing rates.

2. In the hyperparameter experiments (Appendix B), we have provided additional analysis on the hyperparameters of the backbone model and the weight of the loss, conducting hyperparameter analysis on all four datasets.

3. We have also included additional experiments analyzing the model's performance under different future lengths (Appendix G).

4. processed the datasets into a format with varying missing rates and conducted experiments (Appendix H).

5. We evaluated the performance of the proposed model under two additional data missing scenarios (Appendix I).

6. Considering the possibility of incomplete data collection in real-world scenarios (i.e., missing data in the training set), we designed experiments to evaluate performance when the training set contains missing data and when the teacher model's performance degrades (Appendix J).

7. Additionally, we selected several end-to-end models designed for MTSF tasks with missing data as baselines and conducted experiments (Appendix K).

8. We have added the standard deviation of the forecasting results for all models in Section 4.2 (Main Results).

9. We have revised Figure 1. Specifically, we used MAE to illustrate the performance changes of the baselines, enabling readers to more intuitively understand the variations in the model‘s’ performance.

---

### Author Response · Authors · 2024-11-25
**Official Comment by Authors**

Dear reviewers, given that the Author/Reviewer Interaction Stage is approaching its end, we are wondering if our response has satisfactorily addressed your concerns. In case you have any further questions or comments, we will be glad to answer and discuss.

---

### Author Response · Authors · 2024-11-25

Dear Reviewers,

In the latest revised version, to help reviewers better understand our paper and the modifications made, we have made the following efforts:

(1) All modifications in our revision are highlighted in red.

(2) Standard deviations have been added to as many experiments as possible (Appendix E, Appendix F, Appendix G, Appendix H, Appendix I, and Appendix K).

(3) We have corrected the incorrect citation format.

---

### Author Response · Authors · 2024-11-27
**Summary of the Revised Version of our Paper**

Dear Reviewers,

We greatly appreciate each reviewer for taking the time to carefully read our paper and provide valuable feedback, which has been crucial in improving the quality of our work. Considering that the revision phase of the paper is nearing completion, we have submitted another revised version. To facilitate the reviewers' comprehensive and accurate understanding of our paper, we provide the following summary:

1. **Motivation:** The main motivation of our paper is to improve the robustness of  in existing forecasting models in Multivariate Time Series Forecasting with Unfixed Missing Rates. Specifically, in the real world, the presence of missing values can harm the performance of prediction models in the Multivariate Time Series Forecasting (MTSF) task (the higher the missing rate, the greater the model's prediction error), and we aim to mitigate this phenomenon. To this end, by observing MTS data with missing values, we believe existing models face two key challenges: (1) Missing values disrupt the semantics of the MTS (damaging global information and introducing anomalous local information). (2) The missing rate varies over time, requiring existing models to be trained separately for different missing rates. These two challenges demonstrate that prediction models lack robustness when facing MTSF with unfixed missing rates, limiting their performance. To address these challenges, we believe the most important task is to achieve semantic alignment between fully observed and sparsely observed data under different missing rates. To achieve this goal, we designed two methods: offline knowledge distillation and multi-view contrastive learning, and proposed Merlin. As a result, compared to imputation models, Merlin improves the prediction performance of various backbones (STID, DSformer, TSmixer, FourierGNN) by 2% to 10%. Overall, our analysis and summary of the key challenges in the task, the design of the core technical approach, and the proposal of the method are all based on specific reasoning.

2. **The scenario of the task:** In the initial design of our experiments, we referred to the design of most existing works [1] [2]. Specifically, the most common experimental scenario typically involves the random missing scenario, and most works evaluate the model's performance under high missing rates to demonstrate the model's capability. However, we also acknowledge the reviewers' concerns, so we have also evaluated other missing scenarios and the model's performance under low missing rates.

3. **Experimental setup:** In terms of experimental setup, we mask the original data to obtain four different datasets with varying missing rates. For Merlin, we train a teacher model on the original data and train a student model on the four datasets with different missing rates, using only the student model during the inference phase. For the baselines, we considered two training approaches: the first involves training a model using both the original data and the four datasets with different missing rates. The second approach trains separate models for each missing rate. We report the best prediction results from these two methods. Overall, whether for Merlin or the baselines, the amount of data used for training is equivalent.

4. More experimental details (such as hyperparameters) are provided in the revised version. Additionally, we have included the standard deviations for all the experimental results in the tables.

We would like to express our sincere gratitude to all reviewers for taking their valuable time to provide constructive comments. Considering that no further revisions can be submitted in the subsequent stages, if the reviewers have additional suggestions, we will provide a detailed response in the rebuttal. Moreover, we will continue to refine our paper in future revisions.

Best regards,

Authors

[1] Learning to Reconstruct Missing Data from Spatiotemporal Graphs with Sparse Observations

[2] Biased Temporal Convolution Graph Network for Time Series Forecasting with Missing Values

---

### Meta-Review · Area_Chair_3naE · 2024-12-21

**Metareview:**

This paper proposes Merlin, a multi-view representation learning approach that addresses multivariate time series forecasting (MTSF) with missing values using offline knowledge distillation and multi-view contrastive learning. The topic is relevant and potentially impactful across various domains. The experimental evaluation is also quite comprehensive after the rebuttal. However, there is a key concern about the novelty. The method leverages the existing backbone STID and combines existing techniques, such as knowledge distillation and contrastive learning. For example, the feature distillation, label distillation and the multi-view contrastive learning are all standard. Therefore, the novelty of this paper is insufficient for a top-tier conference like ICLR. For this reason, I would like to recommend rejecting this paper.

**Additional Comments On Reviewer Discussion:**

During the rebuttal period, 3 out of 4 reviewers responded to the authors’ replies. Both reviewers VJBz and PvdY were satisfied with the author responses and thus gave positive scores. Reviewer odRh still had concerns about the presentation, motivation, benchmarking, etc., and thus kept the negative score. In addition, I would think the authors did not address the concerns about the novelty part.

---

### Decision · Program_Chairs · 2025-01-22

Reject